# Political expression of academics on Twitter

Prashant Garg [1] ✉ & Thiemo Fetzer[2,3]

Academics play a vital role in the generation and dissemination of knowledge, ideas and narratives. Social media provide new, more direct ways of science communication. Yet, since not all academics engage with social media, the sample that does so may have an outsized influence on shaping public perceptions of academia through the set topics they engage with and their style and tone of communication. We describe patterns in academics' expression online using an international dataset covering nearly 100,000 scholars linking their Twitter content to academic records. We document large and systematic variation in politically salient academic expression concerning climate action, cultural and economic concepts. We show that US academics often diverge from the US Twitter population at large in topic focus and style, although academics are not necessarily more extreme in their beliefs. Future work should examine potential impacts on public trust and the reasons why academics express themselves politically on social media.

Social media platforms have become key venues for sharing, debating and spreading ideas. Academics play a vital role in these platforms as knowledge producers and disseminators. Policymakers may seek ideas, and journalists often look for content on these platforms for traditional media[1,2]. According to Muck Rack's 'State of Journalism 2019' report (https://info.muckrack.com/stateofjournalism), 60% of journalists consult digital sources first, with 22% prioritizing Twitter, highlighting its influence on shaping news.

How academics engage with social media probably influences public interaction with science. Do political stances vary by field or region? Do academics represent diverse views, or are certain perspectives over-represented? Academic authority, inequalities in 'success', and selection biases in social media participation may impact language and tone, potentially impacting healthy discourse.

Concerns about declining trust in science have been raised[3–6], pointing to issues such as lack of reproducibility[7], partisan polarization[8–10] and misinformation[11]. However, the potential role of academics' social media expressions in influencing public perceptions has been underexplored. The backlash[12] against Nature's endorsement of Joe Biden in the 2020 US election[13] shows risks associated with political expression in scientific discourse.

This paper focuses on Twitter, which may play a role in amplifying topics such as climate change and socio-economic policy, making it a potentially useful platform for analysing academic expression. We constructed a dataset linking Twitter profiles of 99,274 academics to their academic records, covering 12,675 institutions across 174 countries and 19 disciplines, capturing all Twitter activity from 2016 to 2022. Previous studies on Twitter data of academics have explored the composition of academics online or focused on specific fields[14,15]. We employ scalable large-language model (LLM) classification techniques to categorize and label the content, focusing on both the substantive content ('what') and the language and tone of expression ('how'). For content, we classify topics and use stance-detection techniques to assess academics' stances on topics such as climate crisis, economic policy and cultural issues. For tone, we measure egocentrism, toxicity and the balance between reason and emotion. This results in a dataset of academics' revealed political stances, allowing for detailed analysis of both 'what' they express and 'how' they communicate.

Our data reveal key observations. US academics' stances on politicized topics such as climate change, immigration and redistribution often diverge significantly from those of the general US population. They display a higher net stance on climate action than the general US Twitter population, and they express significantly greater support for cultural liberalism and economic collectivism. These differences may reflect greater expressiveness rather than more extreme beliefs, potentially skewing public perceptions.

[1]Department of Economics and Public Policy, Imperial College London, London, UK. [2]Department of Economics, University of Warwick, Coventry, UK. [3]Department of Economics, University of Bonn, Bonn, Germany. ✉e-mail: prashant.garg@imperial.ac.uk

We also find that a small minority of highly active academics—often those with less traditional academic recognition—generate much of the content. Political expression differs by field, institution rank, gender and country. US-based academics show lower support for climate action than their non-US counterparts but display stronger cultural liberalism. Among institutions, scholars at top-100 universities have slightly lower climate action support than those at institutions ranked 101–500—yet higher than those at institutions ranked 501–1,500—and they express more pronounced cultural liberalism than their peers at rank 101–500 institutions. These differences could indicate a division of labour in knowledge dissemination but may also risk shaping public perceptions based on a small, vocal subset. This divergence between academics and other social media users could contribute to misconceptions about academic consensus, possibly exacerbating representation gaps[16–20].

Not only do we observe differences in political expression—the 'what'—but we also observe differences in language and tone—the 'how'. Self-referential language is linked to psychological traits such as vulnerable narcissism and distress[21–25]. We document a significant increase in such egocentric language over time, with US academics exceeding their non-US counterparts in both egocentrism and toxicity. In addition, academics with high Twitter reach and lower academic credibility display slightly greater emotionality than those with low reach and high credibility.

Although our study does not examine the potential effects of academics' social media activity on public trust, understanding how academics express themselves online is the first step for grasping impacts on public perceptions of science. Our study provides a large-scale descriptive analysis of academics' political expression on Twitter across subgroups, laying the groundwork for future research to explore its impacts on public trust and underlying causal factors, such as academics' motivations for political expression—whether driven by name recognition, ideological beliefs, or desire to disseminate knowledge.

## Results

We analyse communication patterns of 99,274 academics on Twitter from January 2016 to December 2022 (see Table 1), using AI-driven classifiers linked to OpenAlex for academic metrics. Our study examines variations across demographics (country, field, gender), focusing on political discourse around climate change and socio-economic issues. We explore inequalities in content creation, engagement, followers and citations to motivate our findings. Behavioural traits such as egocentrism, toxicity and emotionality are examined, along with temporal trends in academic expression.

### Distribution of academic influence on Twitter

Our dataset includes all forms of Twitter activity—original tweets, retweets, quoted retweets and replies—capturing full academic engagement. Figure 1 shows that distributions of content creation, engagement (likes), followers and citations follow a power law, highlighting significant inequality, with a small subset of academics dominating these metrics. For instance, the top 5% of accounts receive 30% of likes and 40% of followers, while the bottom 50% receive only 10% in both metrics. Disparities in content creation are less severe: the top 5% generate 25% of posts versus 15% by the bottom 50%. Citations show the most inequality: the top 5% account for over 50% of citations, while the bottom 50% get less than 5% (Extended Data Fig. 1).

Power laws are common in citation metrics[26,27] and social media mentions of scientific articles[28]. However, leading academics are not typically social media influencers. Despite lower barriers to entry on Twitter, inequalities persist. We found weak correlations between citation counts and Twitter metrics: citations and likes ($r = -0.024$, 95% CI [−0.030, −0.018], $P < 0.001$, $n = 99,274$), citations and followers ($r = 0.090$, 95% CI [0.084, 0.096], $P < 0.001$), and citations and content creation ($r = -0.021$, 95% CI [−0.027, −0.014], $P < 0.001$). This

suggests that academic recognition does not directly translate to Twitter engagement, although Twitter metrics still follow a power law. Similar patterns are observed across subgroups (gender, field, university, country), as shown in Extended Data Fig. 1. This raises concerns that public engagement with academic research may be skewed, potentially leading to misrepresentation or reflecting a division of labour between influencers and knowledge producers. The weak correlation underscores that many prominent public intellectuals online gain visibility through public engagement rather than scholarly achievements, often holding lower academic credentials while commanding significant public attention, thus widening the gap between social media influencers and established academic experts.

To substantiate the power-law distribution, we conducted a statistical analysis using the methodology of ref. 29. This confirmed that distributions of content creation, engagement (likes), followers and citations among academics align with power-law behaviour. As detailed in Supplementary Table 1, power-law exponents ($\alpha$) ranged from 2.636 (95% CI [2.625, 2.648]) for likes to 3.337 (95% CI [3.323, 3.352]) for posts, indicating significant inequality. Low Kolmogorov–Smirnov (KS) statistics and corresponding $P$ values (for example, KS = 0.007, $P = 0.399$ for posts) confirm the goodness of fit, consistent with Fig. 1.

### How academics express political topics online

We examined academic expression online using GPT-3.5 turbo and GPT-4 models (summary in Table 2), classifying over 4.9 million tweets into categories such as climate action, immigration and welfare (Fig. 2). We first explored the stance on 'climate action', measured as the net proportion of supportive tweets over all climate-related tweets, ranging from −1 (all anti) to 1 (all supportive). The average academic stance is supportive, with a mean of 0.085 (95% CI [0.083, 0.086]), indicating generally positive sentiment on climate issues.

We define two narratives for addressing climate change: techno-optimism and behavioural adjustment; individuals can support both or neither. On average, academics exhibit significantly greater support for behavioural adjustment ($M = 0.0811$, s.d. = 0.2739, 95% CI [0.0798, 0.0825]) compared with techno-optimism ($M = 0.0498$, s.d. = 0.2176, 95% CI [0.0487, 0.0509]); a paired Welch's $t$-test ($t(151,791) = 33.844$, $P < 0.001$, mean diff = 0.0313, $d = 0.309$) indicates a small but meaningful difference.

No significant gender differences in climate action stance emerged ($t(145,063) = -0.14$, $P = 0.889$, $d = 0.0004$), but males are more techno-optimistic than females (diff = 0.0266, $t(151,697) = 24.45$, $P < 0.001$, $d = 0.12$), while females show stronger support for behavioural adjustments (diff = −0.0107, $t(138,990) = -7.51$, $P < 0.001$, $d = -0.04$). Similarly, STEM sciencies demonstrate greater climate action support than social sciencies (diff = 0.0271, $t(116,536.6) = 16.65$, $P < 0.001$, $d = 0.091$) and humanities (diff = 0.0217, $t(1,142.5) = 2.53$, $P = 0.012$, $d = 0.069$), and are more techno-optimistic than social sciences (diff = 0.0383, $t(134,760.5) = 36.83$, $P < 0.001$, $d = 0.185$) and humanities (diff = 0.0264, $t(1,158.1) = 4.84$, $P < 0.001$, $d = 0.112$). No significant difference appears between social sciences and humanities on climate action (diff = −0.0051, $t(1,153.2) = -0.597$, $P = 0.551$, $d = -0.019$), but social sciences are slightly less techno-optimistic than humanities (diff = −0.0119, $t(1,139.8) = -2.187$, $P = 0.029$, $d = -0.082$). No significant differences in behavioural adjustments were observed across fields, as indicated by the small and non-significant mean differences between STEM and social sciences (diff = 0.0024, $t(103,486) = 1.52$, $P = 0.129$, $d = 0.009$), STEM and humanities (diff = 0.0077, $t(1,138.9) = 0.97$, $P = 0.333$, $d = 0.028$), and social sciences and humanities (diff = 0.0054, $t(1,160.9) = 0.67$, $P = 0.502$, $d = 0.020$).

Academics with high Twitter reach but lacking subject-matter expertise exhibit lower climate action support ($M = 0.0675$) relative to high-reach experts ($M = 0.1297$; $t(13,504) = -18.17$, $P < 0.001$, 95% CI [−0.0689, −0.0554], $d = -0.22$). Similarly, high-reach non-expert academics exhibit significantly lower support for the techno-optimism

**Table 1 | Author-level summary statistics**

| | Mean | Median | s.d. | Min. | Max. | *N* |
|---|---|---|---|---|---|---|
| **Climate: stance and narratives** | | | | | | |
| Climate action | 0.09 | 0 | 0.18 | −1 | 1 | 26,555 |
| Techno-optimism | 0.07 | 0 | 0.17 | 0 | 1 | 26,555 |
| Behavioural adjustment | 0.08 | 0 | 0.16 | 0 | 1 | 26,555 |
| **Socio-economic: stance** | | | | | | |
| Cultural liberalism | 0.04 | 0 | 0.08 | −0.67 | 0.67 | 26,555 |
| Economic collectivism | 0.01 | 0 | 0.06 | −0.67 | 1 | 26,555 |
| **Share ever political** | | | | | | |
| On any topic | 0.75 | 1.00 | 0.43 | 0.00 | 1.00 | 26,555 |
| Climate action | 0.37 | 0.00 | 0.48 | 0.00 | 1.00 | 26,555 |
| Cultural liberalism | 0.56 | 1.00 | 0.50 | 0.00 | 1.00 | 26,555 |
| Economic collectivism | 0.28 | 0.00 | 0.45 | 0.00 | 1.00 | 26,555 |
| **Tone and style of expressions** | | | | | | |
| Egocentricism | 0.33 | 0.28 | 0.27 | 0 | 6.00 | 26,555 |
| Toxicity | 0.04 | 0.03 | 0.04 | 0.00 | 0.90 | 26,555 |
| Emotionality/reasoning | 0.75 | 0.71 | 0.29 | 0.09 | 9.59 | 26,555 |
| Gender: male | 0.60 | 1 | 0.49 | 0 | 1 | 99,274 |
| **Twitter metrics** | | | | | | |
| No. likes | 3,210.85 | 1,673 | 5,736.33 | 0 | 293,797 | 99,274 |
| No. retweets | 843.72 | 276 | 2,117.91 | 0 | 181,268 | 99,274 |
| No. posts | 1,575.24 | 1,032 | 2,052.13 | 4 | 117,088 | 99,274 |
| No. accounts followed | 749 | 529 | 1,116.55 | 0 | 191,923 | 99,274 |
| No. followers | 1,112.70 | 533 | 9,145.94 | 0 | 1,087,504 | 99,274 |
| **Field** | | | | | | |
| Humanities | 0.005 | 0 | 0.07 | 0 | 1 | 99,274 |
| STEM | 0.291 | 0 | 0.45 | 0 | 1 | 99,274 |
| Social sciences | 0.098 | 0 | 0.30 | 0 | 1 | 99,274 |
| **Publication metrics** | | | | | | |
| No. works | 57.58 | 24 | 114.38 | 1 | 7,711 | 99,274 |
| Impact factor (2 yr) | 16.93 | 9 | 38.56 | 0.02 | 4,205 | 99,274 |
| No. citations | 1,370.02 | 213 | 4,523.99 | 1 | 238,736 | 99,274 |

This table provides author-level summary statistics for 99,274 academics, analysing 138 million tweets from 2016 to 2022. It includes behavioural metrics (egocentrism, toxicity, emotionality/reasoning), stances on climate change (climate action, techno-optimism, behavioural adjustment) and socio-political issues (cultural liberalism, economic collectivism). Stance scores range from −1 (anti) to 1 (pro). Sampling focused on up to three English tweets per academic per topic monthly, narrowing to 26,555 politically active academics. During the sample period, 75% made at least one non-neutral tweet, with 37%, 56% and 28% engaging in climate, cultural and economic topics, respectively. Metrics also include Twitter activity, field and publication data for the full sample of academics.

narrative ($M = 0.0397$) relative to high-reach experts ($M = 0.0772$; $t(13,317) = −14.54$, $P < 0.001$, 95% CI [−0.0425, −0.0324], $d = −0.18$) and, albeit to a lesser extent, for behavioural adjustment ($M = 0.0726$ vs $M = 0.0886$; $t(14,355) = −5.72$, $P < 0.001$, 95% CI [−0.0215, −0.0105], $d = −0.06$). Furthermore, low-reach subject-matter experts demonstrate higher climate action support ($M = 0.1579$) than high-reach non-experts ($M = 0.0675$; $t(8,147) = −19.83$, $P < 0.001$, 95% CI [−0.0994, −0.0815], $d = −0.32$). In parallel, low-reach experts also show greater support of techno-optimism ($M = 0.0969$ vs $M = 0.0397$; $t(8,064) = −16.38$, $P < 0.001$, 95% CI [−0.0640, −0.0503], $d = −0.28$) and behavioural adjustment ($M = 0.1121$ vs $M = 0.0726$; $t(8,403) = −10.51$, $P < 0.001$, 95% CI [−0.0469, −0.0322], $d = −0.15$) relative to high-reach non-experts.

Academics from top-100 institutions show significantly lower support for climate action compared with those from rank 101–500 institutions (top 100: $M = 0.0808$, 95% CI [0.0774, 0.0841], $N = 30,134$; 101–500: $M = 0.0882$, 95% CI [0.0849, 0.0915], $N = 32,661$; Welch's $t(62,638.28) = −3.10$, $P = 0.002$, mean diff = −0.0074, $d = −0.025$). In contrast, academics from rank 501–1,500 institutions exhibit significantly higher support ($M = 0.0912$, 95% CI [0.0854, 0.0970], $N = 10,704$) than those from top 100 ($t(18,191.09) = 3.05$, $P = 0.0023$, mean diff = 0.0104, $d = 0.035$), while no significant difference is observed between top 100 and rank 1,501+ institutions (1,501+: $M = 0.0816$, 95% CI [0.0790, 0.0841], $N = 50,251$; Welch's $t(63,409.48) = −0.36$, $P = 0.7173$, mean diff = −0.0008, $d = −0.003$).

With respect to techno-optimism, no significant difference is observed between top 100 and rank 101–500 institutions (top 100: $M = 0.0502$, 95% CI [0.0478, 0.0527], $N = 30,134$; 101–500: $M = 0.0513$, 95% CI [0.0489, 0.0537], $N = 32,661$; Welch's $t(62,481.94) = −0.61$, $P = 0.541$, mean diff = −0.0011, $d = −0.005$). Likewise, the difference between top 100 and rank 501–1,500 institutions (501–1,500: $M = 0.0532$, 95% CI [0.0489, 0.0574], $N = 10,704$) is not significant ($t(18,384.57) = −1.16$, $P = 0.2449$, mean diff = −0.0029, $d = −0.013$). However, academics from rank 1,501+ institutions show significantly

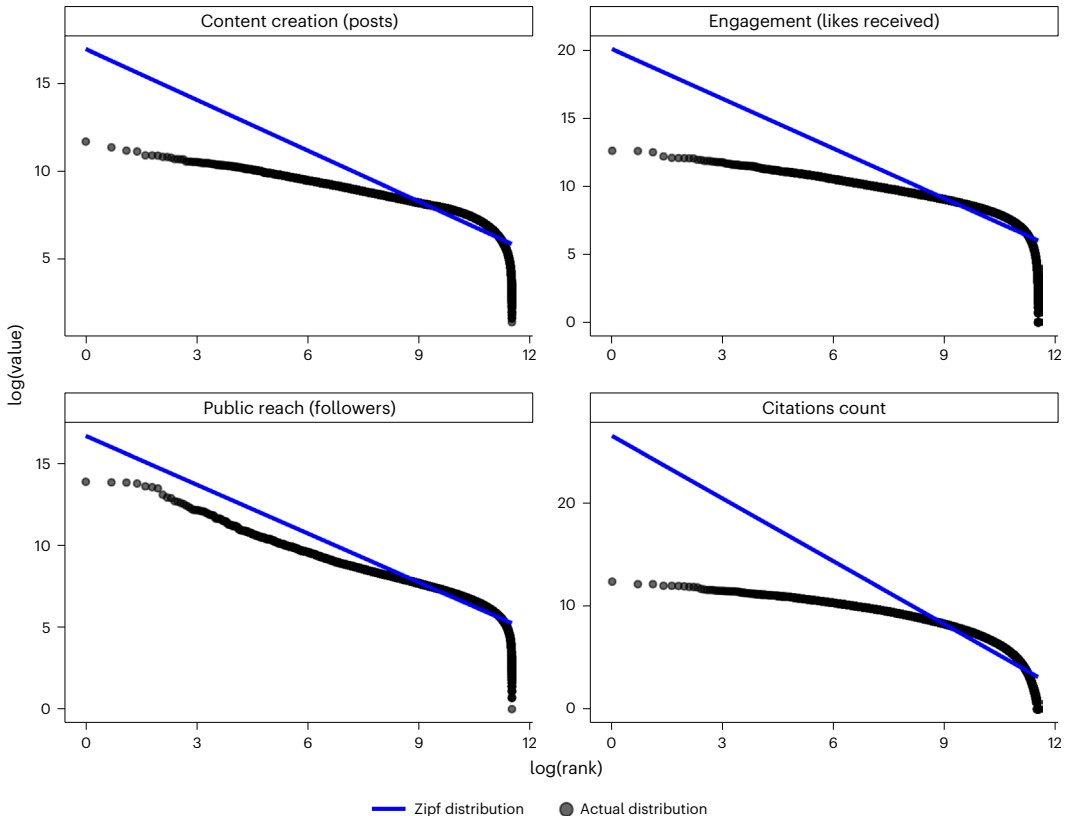

**Fig. 1 | Zipf's law for social media and publication metrics.** Data are drawn from a balanced sample of $n$ = 99,274 academics. Metrics (content creation, engagement, public reach, citations) are plotted by rank ($x$ axis) versus metric value ($y$ axis) on a log scale. Each point corresponds to one academic's metric (no error bars). Twitter data (2016–2022) and OpenAlex citations were analysed jointly.

lower techno-optimism ($M$ = 0.045, 95% CI [0.0432, 0.0468], $N$ = 50,251) relative to those from the top 100 ($t$(60,826.69) = 3.36, $P$ = 0.001, mean diff = 0.0052, $d$ = 0.025).

For behavioural adjustment, top-100 institutions display significantly lower support compared with rank 101–500 institutions (top 100: $M$ = 0.0764, 95% CI [0.0734, 0.0794], $N$ = 30,134; 101–500: $M$ = 0.087, 95% CI [0.0840, 0.0901], $N$ = 32,661; Welch's $t$(62,765.29) = −4.88, $P$ < 0.001, mean diff = −0.0107, $d$ = −0.039). Similarly, support in the top-100 institutions is significantly lower than in rank 501–1,500 institutions (501–1,500: $M$ = 0.0857, 95% CI [0.0804, 0.0910], $N$ = 10,704; $t$(17,990.12) = −3.00, $P$ = 0.003, mean diff = −0.0093, $d$ = −0.035). Finally, no significant difference is found between top 100 and rank 1,501+ institutions (1,501+: $M$ = 0.078, 95% CI [0.0756, 0.0803], $N$ = 50,251; Welch's $t$(63,909.47) = −0.83, $P$ = 0.407, mean diff = −0.0016, $d$ = −0.006).

Academics based in the United States ($M$ = 0.0711, s.d. = 0.2769) exhibit weaker climate action support than non-US academics ($M$ = 0.0895, s.d. = 0.307); Welch's $t$(73,530.02) = −10.92, $P$ < 0.001, mean diff = −0.0183, $d$ = −0.061. For techno-optimism, US-based scholars ($M$ = 0.0439, s.d. = 0.205) report significantly lower support than non-US scholars ($M$ = 0.0519, s.d. = 0.2217); Welch's $t$(71,850.73) = −6.42, $P$ < 0.001, mean diff = −0.0079, $d$ = −0.036. Similarly, US-based academics also show a less pronounced dip in behavioural adjustment (mean diff = −0.0206, $t$(75,116.07) = −13.64, $P$ < 0.001, $d$ = −0.076) than non-US academics.

Figure 3 presents results on stances revealed through academics' social media content covering broader socio-economic issues. We detect stances on topics such as racial relations, reproductive rights, immigration, welfare state, income redistribution and higher taxes for the wealthy. To reduce dimensionality, we create two indices: 'cultural liberalism' (average stance on racial equality, reproductive rights, immigration) and 'economic collectivism' (average stance on welfare state, redistribution, higher taxes), ranging from −1 (opposition) to 1 (support). The average net support for cultural liberalism is 0.043 (s.d. = 0.148, 95% CI [0.043, 0.044]), while for economic collectivism it is 0.018 (s.d. = 0.128, 95% CI [0.017, 0.019]). This suggests that the average academic expression leans leftward or 'progressive' on both dimensions.

Female academics express slightly more culturally liberal views than male academics: females ($M$ = 0.0497, s.d. = 0.1570, 95% CI [0.0485, 0.0509], $N$ = 66,675) vs males ($M$ = 0.0383, s.d. = 0.1411, 95% CI [0.0373, 0.0392], $N$ = 85,117); Welch's $t$(135,325.8) = 14.73, $P$ < 0.001, diff = 0.0114, $d$ = 0.081. They also slightly surpass males on view expressed on economic collectivism ($M$ = 0.0202 vs 0.0161); Welch's $t$(140,340.5) = 6.19, $P$ < 0.001, diff = 0.0041, $d$ = 0.033.

Social scientists favour economic collectivism more than others: social sciences ($M$ = 0.0207, s.d. = 0.1353, $N$ = 50,274) vs non-social sciences ($M$ = 0.0164, s.d. = 0.1236, $N$ = 101,518); Welch's $t$(92,608.9) = −6.00, $P$ < 0.001, diff = 0.0043, $d$ = −0.032. Their cultural liberalism ($M$ = 0.0424) does not differ from non-social scientists ($M$ = 0.0437); $t$(98,659.1) = −1.54, $P$ = 0.123, $d$ = −0.008. Humanities scholars ($M$ = 0.0511, s.d. = 0.1586) are more culturally liberal than non-humanities scholars ($M$ = 0.0432, s.d. = 0.1482); Welch's $t$(2,419.3) = −2.41, $P$ = 0.016, diff = 0.0079, $d$ = 0.05. However, on economic collectivism, humanities ($M$ = 0.0163) and non-humanities ($M$ = 0.0179) do not differ: $t$(2,425.3) = 0.57, $P$ = 0.572, $d$ = 0.012.

Academics with high Twitter reach but low subject-matter expertise exhibit lower cultural liberalism relative to high-reach experts. Specifically, high-reach experts demonstrate a mean cultural liberalism score of $M$ = 0.0495 (95% CI [0.0469, 0.0521], $N$ = 14,040) compared with $M$ = 0.0421 (95% CI [0.0410, 0.0431], $N$ = 79,915) for high-reach non-experts; Welch's $t$(18,499) = 5.15, $P$ < 0.001, mean diff = 0.00742,

## Table 2 | Summary of topics, stances and narratives

| Topics | No. tweets | % of own topic | % of total tweets |
|---|---|---|---|
| **Abortion rights** | 72,645 | | 0.052 |
| Abortion rights (sampled) | 9,873 | 100.00 | 0.007 |
| Pro | 3,545 | 35.9 | 0.0026 |
| Anti | 448 | 4.54 | 0.0003 |
| Neutral | 5,880 | 59.6 | 0.0043 |
| **Racial equality** | 608,039 | | 0.439 |
| Racial equality (sampled) | 35,162 | 100.00 | 0.025 |
| Pro | 25,352 | 72.1 | 0.018 |
| Anti | 5,109 | 14.5 | 0.0037 |
| Neutral | 4,701 | 13.4 | 0.0034 |
| **Immigration** | 734,692 | | 0.53 |
| Immigration (sampled) | 233,657 | 100.00 | 0.17 |
| Pro | 37,107 | 15.88 | 0.027 |
| Anti | 15,302 | 6.55 | 0.011 |
| Neutral | 181,248 | 77.57 | 0.131 |
| **Welfare state** | 679,206 | | 0.491 |
| Welfare state (sampled) | 40,007 | 100.00 | 0.029 |
| Pro | 6,207 | 15.11 | 0.0045 |
| Anti | 2,087 | 5.22 | 0.0015 |
| Neutral | 31,713 | 79.27 | 0.023 |
| **Higher tax for wealthy** | 311,888 | | 0.225 |
| Higher tax for wealthy (sampled) | 83,119 | 100.00 | 0.060 |
| Pro | 18,858 | 22.69 | 0.014 |
| Anti | 8,848 | 10.64 | 0.006 |
| Neutral | 56,911 | 68.47 | 0.041 |
| **Income redistribution** | 464,921 | | 0.336 |
| Income redistribution (sampled) | 36,269 | 100.00 | 0.026 |
| Pro | 7,183 | 19.8 | 0.005 |
| Anti | 1,588 | 4.38 | 0.001 |
| Neutral | 27,498 | 75.81 | 0.020 |
| **Climate action** | 2,057,187 | | 1.49 |
| Climate action (sampled) | 322,519 | 100.00 | 0.23 |
| Pro | 93,163 | 28.89 | 0.067 |
| Anti | 6,507 | 2.02 | 0.005 |
| Neutral | 222,849 | 69.10 | 0.161 |
| Techno-optimist (TO) | 47,430 | 14.05 | 0.034 |
| Behavioural adjustment (BA) | 89,619 | 26.55 | 0.065 |
| Both TO and BA | 43,431 | 12.87 | 0.031 |
| Neither TO nor BA | 157,063 | 46.53 | 0.114 |
| Topical tweets | 4,928,578 | – | 3.56 |
| All tweets | 138,372,165 | – | 100 |

This table summarizes the number and distribution of tweets across key socio-political topics from 138,372,165 tweets by academics (2016–2022). Topics include abortion rights, racial equality, immigration, welfare state, higher taxes, income redistribution and climate action. Stance detection (pro, anti, neutral) was performed using GPT-3.5 Turbo, validated with F-scores of 84–92. The sampling strategy involved extracting up to three English tweets per academic per topic monthly, ensuring consistency while managing computational costs. Within climate action, narratives were categorized into techno-optimism, behavioural adjustment, both, or neither, reflecting diverse academic perspectives on climate solutions. Political topics constitute 3.56% of all tweets.

$d = 0.05$. Furthermore, low-reach subject-matter experts exhibit higher cultural liberalism ($M = 0.0527$, 95% CI [0.0484, 0.0570], $N = 5,412$) than high-reach non-experts ($M = 0.0421$, 95% CI [0.0410, 0.0431], $N = 79,915$); Welch's $t(6,043) = 4.75$, $P < 0.001$, mean diff = 0.01065, $d = 0.066$.

A similar pattern is observed for economic collectivism. High-reach non-expert academics show lower economic collectivism relative to high-reach experts, with high-reach experts yielding a mean score of $M = 0.0245$ (95% CI [0.0219, 0.0270], $N = 12,052$) versus $M = 0.0168$ (95% CI [0.01595, 0.01762], $N = 81,903$) for high-reach non-experts; Welch's $t(14,743) = 5.59$, $P < 0.001$, mean diff = 0.00768, $d = 0.063$. Moreover, low-reach experts exhibit higher economic collectivism ($M = 0.0220$, 95% CI [0.01723, 0.02676], $N = 3,721$) compared with high-reach non-experts ($M = 0.0168$, 95% CI [0.01595, 0.01762], $N = 81,903$); Welch's $t(3,953) = 2.11$, $P = 0.0348$, mean diff = 0.00521, $d = 0.035$.

Top-100 universities' scholars ($M = 0.0497$, s.d. = 0.1565) are more culturally liberal than those from institutions ranked 101–500 ($M = 0.0407$, s.d. = 0.1451); Welch's $t(54,277) = 6.96$, $P < 0.001$, mean diff = 0.0089, $d = 0.062$. They do not differ significantly on economic collectivism ($t(55,245) = 1.64$, $P = 0.101$, mean diff = 0.0018, $d = 0.014$). Moreover, scholars from institutions ranked 1,500+ are significantly less culturally liberal than those at top 100 (mean diff = −0.0066, $t(53,775) = −5.58$, $P < 0.001$, $d = −0.045$), while no significant difference in economic collectivism is observed between these groups (mean diff = 0.0007, $t(55,468) = 0.68$, $P = 0.499$, $d = 0.005$).

US-based scholars ($M = 0.0547$, s.d. = 0.1663) lean more culturally liberal than non-U.S scholars. ($M = 0.0394$, s.d. = 0.1416); Welch's $t(58,960.6) = 16.23$, $P < 0.001$, diff = 0.0153, $d = 0.092$. No difference emerges on economic collectivism ($t(69,722.3) = −0.20$, $P = 0.845$, $d = −0.001$).

### Tone and style of expression online

Figure 4 shows mean levels of three behavioural features of 'how' academics communicate online: egocentrism, toxicity and emotionality. Overall averages serve as reference points to compare differences across individual-level characteristics.

Egocentrism, measured by the average number of self-referential terms ('I', 'me', 'my', 'myself') per tweet, averages 0.336 (s.d. = 0.363, 95% CI [0.334, 0.338]). Female academics ($M = 0.3482$, s.d. = 0.3661, $N = 66,675$, 95% CI [0.3454, 0.3510]) exhibit higher egocentrism than male academics ($M = 0.3267$, s.d. = 0.3603, $N = 85,117$, 95% CI [0.3243, 0.3291]); Welch's $t(142,130) = 11.44$, $P < 0.001$, 95% CI [0.0178, 0.0252], $d = 0.060$. Humanities scholars ($M = 0.3655$, s.d. = 0.3395, $N = 2,355$, 95% CI [0.3518, 0.3792]) exceed others ($M = 0.3357$, s.d. = 0.3633, $N = 149,437$, 95% CI [0.3338, 0.3375]); Welch's $t(2,439.7) = 4.23$, $P < 0.001$, 95% CI [0.0160, 0.0437], $d = 0.088$. Social sciences scholars ($M = 0.3293$, s.d. = 0.3376, $N = 50,274$, 95% CI [0.3263, 0.3322]) show slightly lower egocentrism than others ($M = 0.3395$, s.d. = 0.3749, $N = 101,518$, 95% CI [0.3372, 0.3418]); Welch's $t(110,091) = −5.38$, $P < 0.001$, 95% CI [−0.0065, −0.0140], $d = −0.030$. Academics with high reach but low academic credibility ($M = 0.3498$, s.d. = 0.3311, $N = 44,735$, 95% CI [0.3468, 0.3529]) exhibit significantly higher egocentrism than the contrasting profile, that is, those with low reach but high credibility ($M = 0.3167$, s.d. = 0.4095, $N = 24,463$, 95% CI [0.3116, 0.3219]); Welch's $t(42,132) = 10.85$, $P < 0.001$, 95% CI [0.0271, 0.0391], $d = 0.081$.

Egocentrism increases with university ranking: academics at top-100 institutions ($M = 0.3674$, s.d. = 0.3651, $N = 30,134$, 95% CI [0.3633, 0.3715]) exhibit higher egocentrism than those from institutions ranked 101–500 ($M = 0.3315$, s.d. = 0.3322, $N = 32,661$, 95% CI [0.3279, 0.3351]); Welch's $t(60,943) = 12.84$, $P < 0.001$, 95% CI [0.0304, 0.0413], $d = 0.108$. Similarly, top-100 institutions also show higher egocentrism than those ranked 501–1,500 ($M = 0.3286$, s.d. = 0.3345, $N = 10,704$, 95% CI [0.3223, 0.3350]); Welch's $t(20,383) = 10.04$, $P < 0.001$, 95% CI [0.0312, 0.0463], $d = 0.116$. US-based academics ($M = 0.393$, $N = 38,666$) show higher

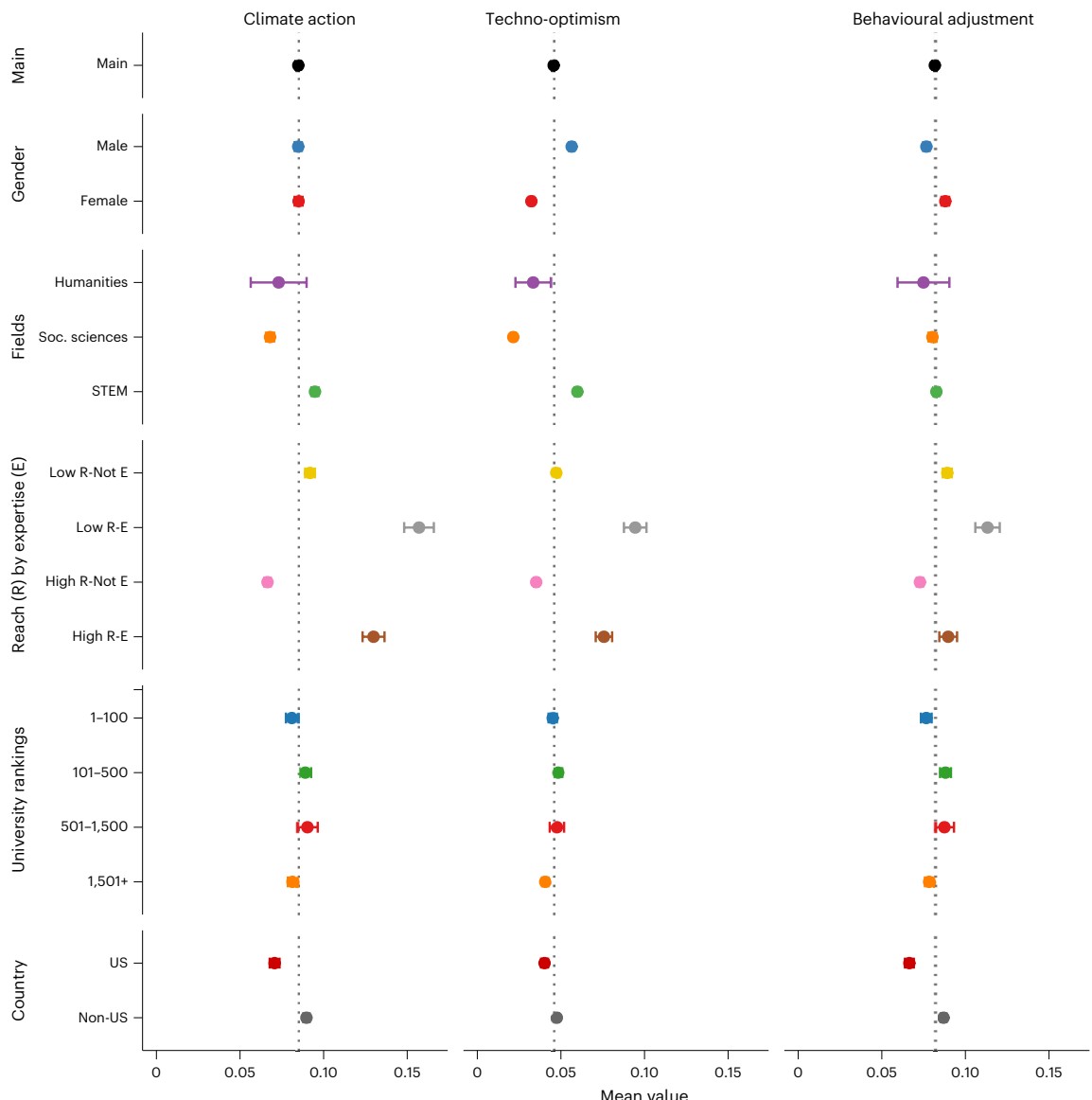

**Fig. 2 | How academics talk about climate action online, averaged by author characteristics.** Data come from a balanced sample of $n$ = 99,274 academics observed from 2016 to 2022. Stances (negative to positive) were classified by GPT-based methods and aggregated per academic from up to three tweets per month. Each panel plots the mean stance (−1 to +1) on 'climate action', 'techno-optimism', or 'behavioural adjustment' for different academic subgroups. Circles show group means; error bars are 95% confidence intervals computed via s.e.m. In each panel, the dotted vertical line represents the overall academic average (the 'Main' group mean) for that stance, serving as a baseline for subgroup comparisons. Methods detail sampling (ensuring each academic tweets at least once early and late in the period) and stance detection.

egocentrism than non-US academics ($M$ = 0.317, $N$ = 113,126); Welch's $t$(64,870) = 35.02, $P$ < 0.001, diff = 0.0768, $d$ = 0.204.

Toxicity, calculated using Google's Perspective Application Programming Interface (API) (probability a tweet is classified as toxic), averages 4.4% ($M$ = 0.044, s.d. = 0.069, 95% CI [0.0437, 0.0444]) among academics. Female academics ($M$ = 0.0439, s.d. = 0.0671, $N$ = 66,416, 95% CI [0.0434, 0.0444]) and male academics ($M$ = 0.0442, s.d. = 0.0702, $N$ = 84,733, 95% CI [0.0437, 0.0446]) do not differ significantly in toxicity; Welch's $t$(145,387) = −0.78, $P$ = 0.434, 95% CI [−0.0010, 0.0004], $d$ = −0.004.

Humanities scholars ($M$ = 0.0495, s.d. ≈ 0.0708, $N$ = 2,345, 95% CI [0.04668, 0.05241]) exhibit higher toxicity than non-humanities scholars ($M$ = 0.04395, s.d. ≈ 0.06887, $N$ = 148,804, 95% CI [0.04360, 0.04430]); Welch's $t$(2,414.5) = 3.80, $P$ < 0.001, 95% CI [0.00271, 0.00848], $d$ = 0.079. Similarly, social scientists ($M$ = 0.04569, s.d. ≈ 0.06966, $N$ = 50,050, 95% CI [0.04508, 0.04630]) show higher

toxicity than non-social scientists ($M$ = 0.04322, s.d. ≈ 0.06851, $N$ = 101,099, 95% CI [0.04280, 0.04365]); Welch's $t$(98,300) = 6.50, $P$ < 0.001, 95% CI [0.00172, 0.00320], $d$ = 0.035. In contrast, STEM scholars ($M$ = 0.04478, s.d. ≈ 0.06947, $N$ = 66,906, 95% CI [0.04425, 0.04530]) exhibit lower toxicity than non-STEM scholars ($M$ = 0.04345, s.d. ≈ 0.06844, $N$ = 84,243, 95% CI [0.04299, 0.04391]); Welch's $t$(142,533) = −3.71, $P$ = −0.00021, 95% CI [−0.00203, −0.00063], $d$ = −0.019. Academics with high reach but low academic credibility ($n$ = 39,315, $M$ = 0.04330, s.d. ≈ 0.06721, 95% CI [0.04263, 0.04396]) exhibit lower toxicity than those with the contrasting profile, that is, ones with low reach but high credibility ($n$ = 21,948, $M$ = 0.04728, s.d. ≈ 0.07327, 95% CI [0.04631, 0.04825]); Welch's $t$(42,210) = −6.64, $P$ < 0.001, 95% CI [−0.00516, −0.00281], $d$ = −0.054.

Academics at top-100 universities ($M$ = 0.0463, s.d. ≈ 0.07185, $N$ = 30,013, 95% CI [0.04550, 0.04713]) exhibit higher toxicity than those at institutions ranked 101–500 ($M$ = 0.04333, s.d. ≈ 0.06928, $N$ = 32,512,

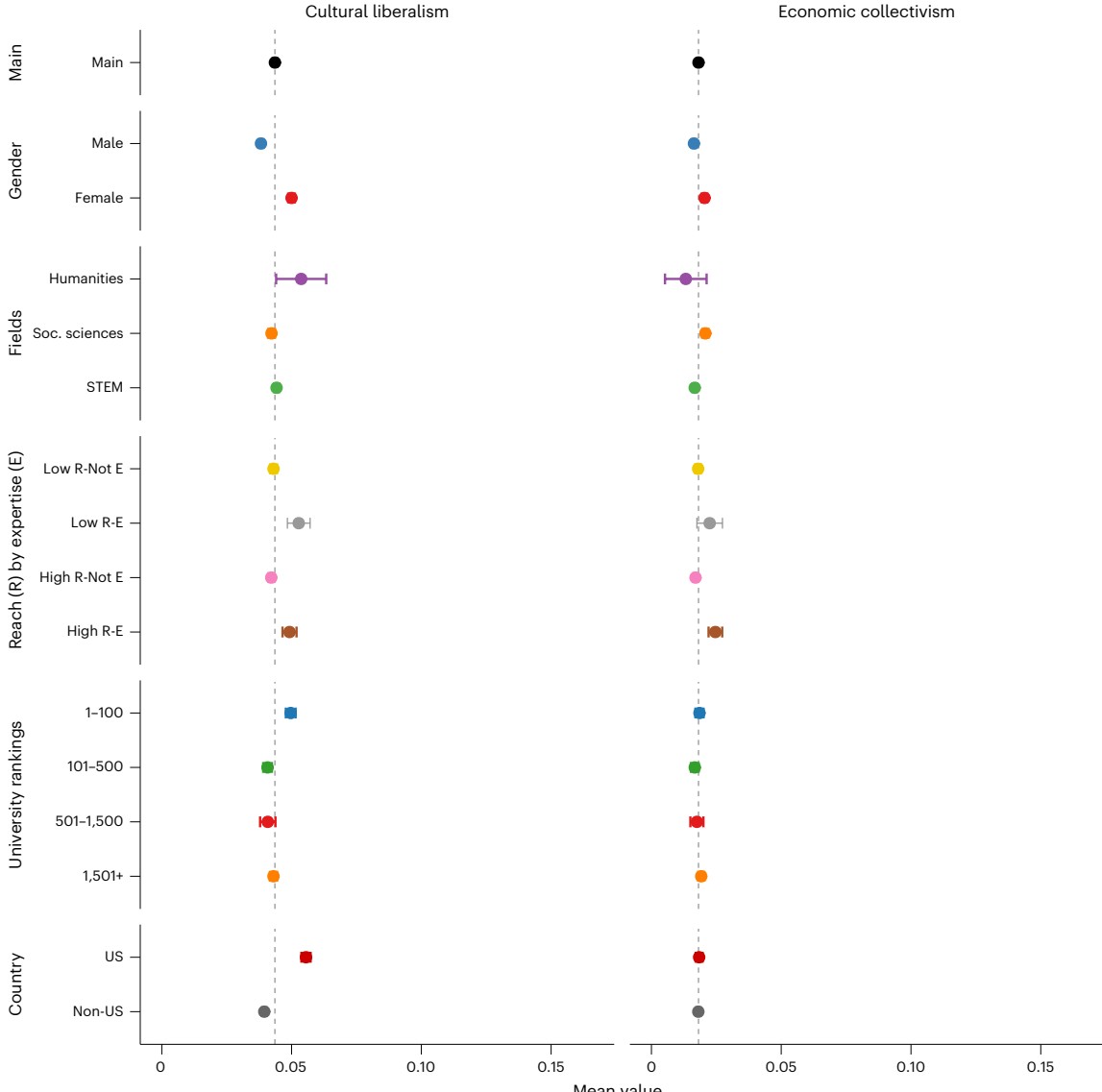

**Fig. 3 | How academics talk about socio-economic issues online, averaged by author characteristics.** Data come from a balanced sample of $n = 99,274$ academics observed from 2016 to 2022. Stances (negative to positive) were classified by GPT-based methods and aggregated per academic from up to three tweets per month. Each panel plots the mean stance (−1 to +1) on 'cultural liberalism' or 'economic collectivism' for different academic subgroups. Circles show group means; error bars are 95% confidence intervals computed via s.e.m. In each panel, the dashed vertical line represents the overall academic average (the 'Main' group mean) for that stance, serving as a baseline for subgroup comparisons. Methods detail sampling (ensuring each academic tweets at least once early and late in the period) and stance detection.

95% CI [0.04257, 0.04408]); Welch's $t(61,687) = 5.28$, $P < 0.001$, 95% CI [0.00188, 0.00409], $d = 0.043$. Similarly, top-100 universities also show higher toxicity than institutions ranked 501–1,500 ($M = 0.04318$, s.d. ≈ 0.06882, $N = 10,660$, 95% CI [0.04187, 0.04448]); Welch's $t(19,473) = 3.9951$, $P < 0.001$, 95% CI [0.00160, 0.00468], $d = 0.0456$. Moreover, US-based academics ($M = 0.04751$, s.d. ≈ 0.07245, $N = 38,506$, 95% CI [0.04679, 0.04823]) exhibit higher toxicity than non-US academics ($M = 0.04285$, s.d. ≈ 0.06761, $N = 112,643$, 95% CI [0.04246, 0.04325]); Welch's $t(62,934) = 11.07$, $P < 0.001$, 95% CI [0.00383, 0.00548], $d = 0.0643$.

Emotionality/reasoning, computed as the ratio of affective to cognitive words, averages 0.747 (s.d. = 0.491, 95% CI [0.7448, 0.7498]) among academics. Emotionality is significantly higher among female academics ($M = 0.7754$, s.d. = 0.5112, $N = 66,675$, 95% CI [0.7716, 0.7793]) than male academics ($M = 0.7253$, s.d. = 0.4734, $N = 85,117$, 95% CI [0.7221, 0.7285]); Welch's $t(137,670) = 19.60$, $P < 0.001$, 95% CI [0.0451, 0.0552], $d = 0.106$. STEM academics ($M = 0.7536$, $N = 84,594$,

95% CI [0.7504, 0.7569]) slightly exceed non-STEM academics ($M = 0.7393$, $N = 67,198$, 95% CI [0.7356, 0.7431]); $t(142,658) = 5.62$, $P < 0.001$, 95% CI [0.0093, 0.0193], $d = 0.029$. In terms of reach and credibility, high-reach/low-credibility scholars ($n = 44,735$, $M = 0.7632$, 95% CI [0.7586, 0.7678]) show significantly higher emotionality than low-reach/high-credibility scholars ($n = 24,463$, $M = 0.7308$, 95% CI [0.72498, 0.73666]); Welch's $t(53,039) = 8.53$, $P < 0.001$, 95% CI [0.0249, 0.0398], $d = 0.069$.

Academics at top-100 universities ($M = 0.7565$, s.d. =0.5291, $N = 30,134$, 95% CI [0.7505, 0.7625]) do not differ significantly in emotionality from those at institutions ranked 101–500 ($M = 0.7525$, s.d. = 0.4888, $N = 32,661$, 95% CI [0.7472, 0.7578]); Welch's $t(61,241) = 0.98$, $P = 0.329$, 95% CI [−0.0040, 0.0120], $d = 0.008$. Similarly, no significant difference is observed between top-100 universities and institutions ranked 501–1,500 ($M = 0.7562$, s.d. = 0.5028, $N = 10,704$, 95% CI [0.7467, 0.7657]); Welch's $t(19,695) = 0.05$, $P = 0.959$, 95% CI [−0.01095, 0.01154], $d = 0.001$. Finally, US-based academics ($M = 0.7675$,

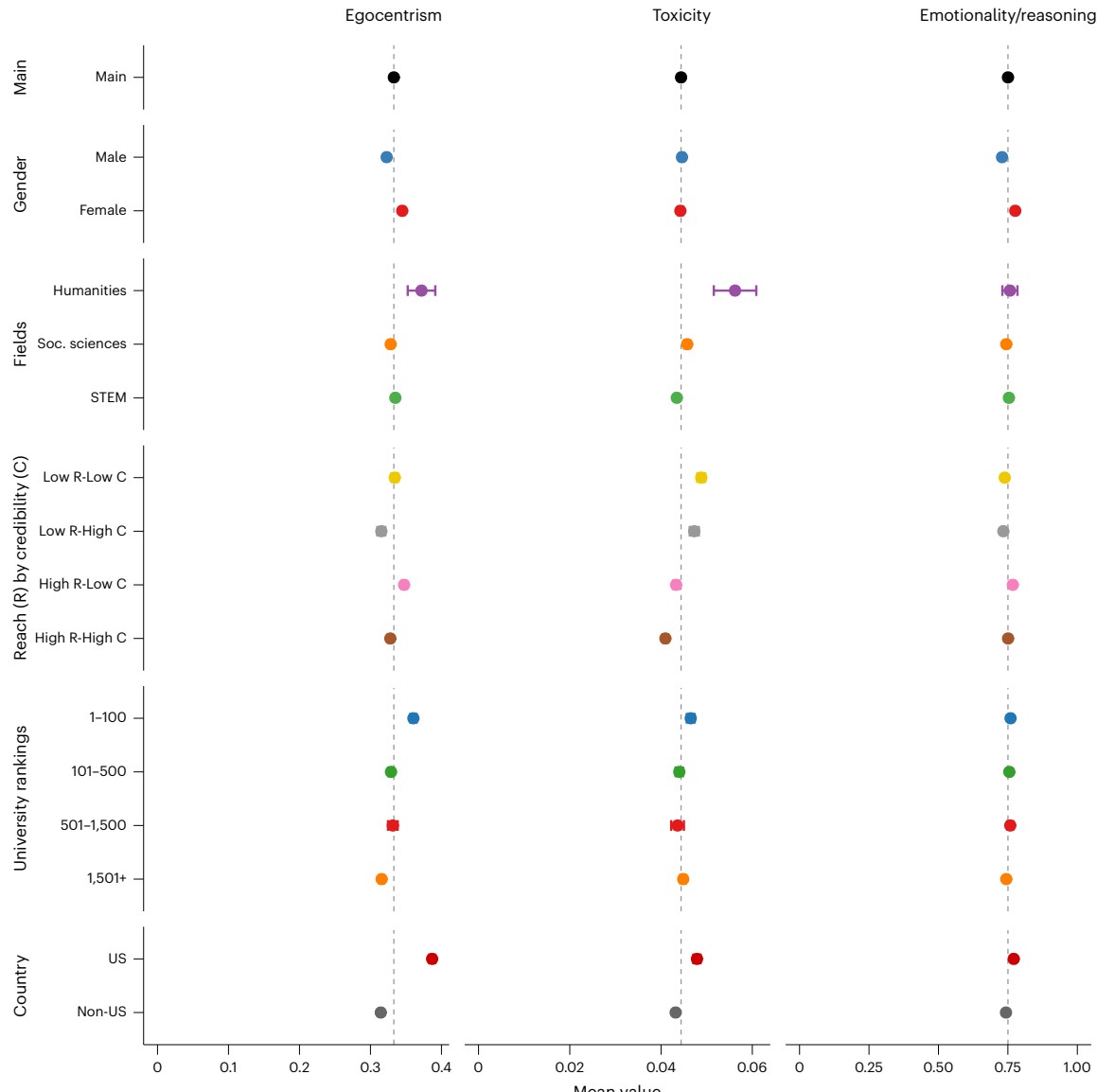

**Fig. 4 | How academics express themselves online, averaged by author characteristics.** Data reflect *n* = 99,274 academics (balanced panel) from 2016 to 2022. 'Egocentrism' and 'emotionality' (ratio of affective to cognitive words) were each measured from the full set of English tweets per user. 'Toxicity' (Google's Perspective API) was computed using up to 10 randomly sampled English tweets per user–year to manage API constraints; see Methods for variable measurement and sampling details. For each subgroup, circles indicate mean metric values;

error bars show ±95% confidence intervals (s.e.m.). In each panel, the dashed vertical line represents the overall academic average (the 'Main' group mean) for that measure, serving as a baseline for subgroup comparisons. Note that the three measures are on different scales: egocentrism can range from 0 upward, emotionality is typically close to 1, and toxicity is bounded between 0 and 1; they are plotted on their inherent scales, thereby preserving the natural variation in each metric.

*N* = 38,666, 95% CI [0.7623, 0.7728]) exhibit higher emotionality than non-US scholars (*M* = 0.7404, *N* = 113,126, 95% CI [0.7376, 0.7432]); $t(61,784) = 8.93$, $P < 0.001$, 95% CI [0.0211, 0.0330], $d = 0.051$.

**Stance differences robust to tone and style controls.** To rule out that subgroup differences in stance are confounded by variations in tone or style, we examined correlations between political stances and expression styles (emotionality, egocentrism, toxicity). As shown in Supplementary Table 7, these correlations are consistently low. For example, the correlation between cultural liberalism and toxicity, the highest among them, is modest ($r = 0.085$, $P < 0.0001$, 95% CI [0.0804, 0.0904]), suggesting only a weak relationship. While these results indicate a lack of strong evidence that tone or style substantially confounds political stances, they do not exclude the possibility of small influences. Regression analyses controlling for expression styles (Supplementary

Note 5.3, and Supplementary Figs. 9 and 10) show that subgroup differences remain qualitatively unchanged, suggesting that differences in stances reflect genuine differences in 'what' is expressed, not 'how.'

**Temporal dynamics in academic expression**
To examine temporal dynamics, we present monthly time trends from January 2016 to December 2022 using LOESS smoothing (Fig. 5). The figure displays trends for all key variables.

We estimate linear trends via ordinary least-squares regression on monthly aggregates. Regression estimates (*b*) quantify year–month changes: egocentrism rose ($b = 1.17 \times 10^{-4}$, 95% CI [$1.14 \times 10^{-4}$, $1.20 \times 10^{-4}$], $P < 0.001$), toxicity declined ($b = -5.35 \times 10^{-6}$, 95% CI [$-5.84 \times 10^{-6}$, $-4.85 \times 10^{-6}$], $P < 0.001$), and emotionality increased ($b = 3.30 \times 10^{-5}$, 95% CI [$2.95 \times 10^{-5}$, $3.66 \times 10^{-5}$], $P < 0.001$). Support for climate action grew ($b = 1.21 \times 10^{-5}$, 95% CI [$9.91 \times 10^{-6}$, $1.42 \times 10^{-5}$], $P < 0.001$) but slowed

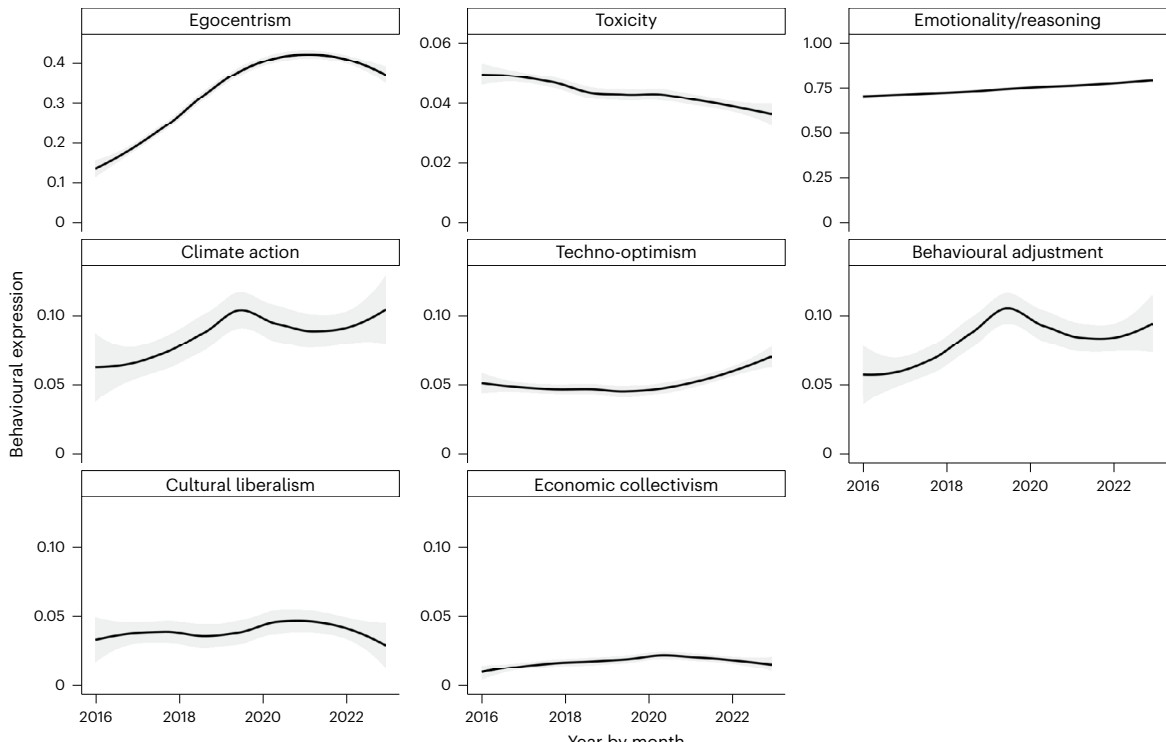

**Fig. 5 | Academic expression over time.** Temporal dynamics of academic expression from January 2016 to December 2022, using a balanced sample of $n$ = 99,274 academics who tweeted at least once in both the early (January–June 2016) and late (July–December 2022) periods of the dataset. Panels show distributions, smoothed trends and variability across key behavioural and political metrics. For the stance measures: climate action, techno-optimism, behavioural adjustment, cultural liberalism and economic collectivism, the natural scale ranges from −1 to +1. In contrast, the other measures: toxicity (bounded between 0 and 1), emotionality/reasoning (a ratio typically close to 1) and egocentrism (ranging from 0 upward), are plotted on their inherent scales. These scale choices preserve the natural variation in each metric. See Methods for details on stance detection, normalization and sampling. Each data point represents the monthly average for the sample, smoothed via LOESS with 95% confidence intervals around the trend line (shaded regions).

during the pandemic; techno-optimism increased ($b = 6.19 \times 10^{-6}$, 95% CI [$4.64 \times 10^{-6}$, $7.75 \times 10^{-6}$], $P < 0.001$) and behavioural-adjustment advocacy increased ($b = 1.16 \times 10^{-5}$, 95% CI [$9.62 \times 10^{-6}$, $1.35 \times 10^{-5}$], $P < 0.001$). Cultural and economic stances also trended upward ($b = 2.60 \times 0^{-6}$, 95% CI [$1.52 \times 10^{-6}$, $3.68 \times 10^{-6}$], $P < 0.001$ for cultural liberalism; $b = 2.63 \times 10^{-6}$, 95% CI [$1.72 \times 10^{-6}$, $3.54 \times 10^{-6}$], $P < 0.001$ for economic collectivism). Although residuals were somewhat skewed, the large time-series sample supports these robust trends.

Supplementary Fig. 17 indicates that these patterns remain largely consistent across subtopics. Supplementary Figs. 11–14 present the temporal evolution for subgroups with significant-level differences (gender, fields, country). Robustness checks controlling for individual author-level fixed effects (Supplementary Note 5.4 and Figs. 18–21) confirm that the observed trends represent genuine within-individual changes, not shifts in social media activity composition.

### US-based academics vs non-academics on US Twitter

To explore differences between academics and the general US Twitter population, we analysed a random sample of 61,259 US users (after filtering out bots) and compared them to 7,724 US-based academics, drawn from a politically active subset of 26,555. For reference, Extended Data Fig. 2 presents the average stance of US academics across eight measures, serving as a benchmark for our subsequent comparisons.

Figure 6 shows significant differences in political stances and expression style, described below. For the year 2022, egocentrism among US academics ($M = 0.467$, s.d. = 0.423, 95% CI [0.457, 0.477], $N = 6,853$) nearly converged with that of the general US Twitter population ($M = 0.477$, s.d. = 0.670, 95% CI [0.475, 0.479], $N = 362,033$); Welch's $t$-test ($t(7,517.3) = -2.01$, $P = 0.045$, mean diff = −0.010, $d = 0.025$) indicates a negligible effect. Note that this result is based solely on 2022 data, whereas the remaining analyses reflect averages over the full observation period.

US academics are notably less toxic ($M = 0.049$, s.d. = 0.075, 95% CI [0.048, 0.050], $N = 51,432$) than general US users ($M = 0.085$, s.d. = 0.132, 95% CI [0.085, 0.085], $N = 1,644,511$); Welch's $t$-test ($t(61,949) = -103.66$, $P < 0.001$, diff = −0.036, $d = -0.479$) shows a medium effect. US academics also display slightly lower emotionality ($M = 0.763$, s.d. = 0.518, 95% CI [0.759, 0.768], $N = 51,633$) relative to general US users ($M = 0.837$, s.d. = 0.386, 95% CI [0.837, 0.838], $N = 2,797,593$); Welch's $t$-test ($t(52,695) = -32.22$, $P < 0.001$, diff = −0.074, $d = -0.143$) suggests a small effect.

We observe large differences in political stance expression between US academics and the general US Twitter population. US academics ($M = 0.072$, s.d. = 0.280, 95% CI [0.070, 0.075], $N = 51,633$) show stronger climate action support than general US users ($M = 0.008$, s.d. = 0.097, 95% CI [0.008, 0.008], $N = 3,488,181$); Welch's $t$-test ($t(51,817) = 52.46$, $P < 0.001$, diff = 0.065, $d = -0.23$) indicates a small effect.

US academics are less techno-optimistic ($M = 0.0443$, s.d. = 0.2057, 95% CI [0.0425, 0.0461]) than general US users ($M = 0.1007$, s.d. = 0.2786, 95% CI [0.0987, 0.1027]); Welch's $t$-test ($t(126,751.9) = -41.567$, $P < 0.001$, mean diff = −0.0564, $d = -0.224$) indicates a small-to-medium effect. They also express lower support for behavioural adjustment ($M = 0.067$, s.d. = 0.2501, 95% CI [0.0649, 0.0692]) vs general US users ($M = 0.302$, s.d. = 0.4297, 95% CI [0.2989, 0.3050]); Welch's $t$-test ($t(124,950) = -123.124$, $P < 0.001$, mean diff = −0.2349, $d = -0.639$) shows a medium-to-large effect. (Individuals supporting both or neither narratives are excluded; among US academics [vs general US users], 26.55% [27.88%] favour behavioural adjustment, 14.05% [11.35%] favour techno-optimism, 12.87% [13.96%] favour both, 46.53% [46.82%] favour neither.)

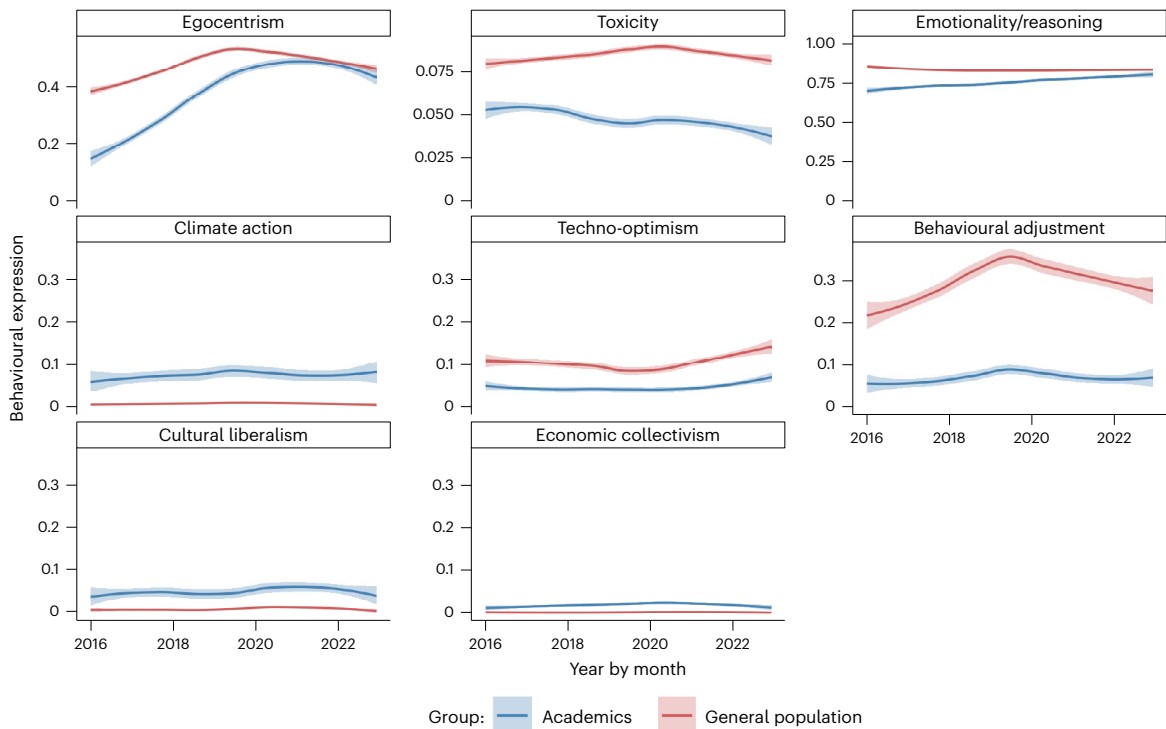

**Fig. 6 | Differences in expression between US-based academics and the general US Twitter population.** Comparison of the temporal dynamics of academic expression with the general US Twitter population from January 2016 to December 2022. Data comprise $n$ = 7,724 US academics and $n$ = 61,259 general US Twitter users. US-based academics were sampled from a larger, politically active balanced sample of 26,555 academics (from an overall balanced sample of 99,274) by ensuring that they tweeted in English and had at least one politically oriented tweet in both the early (January–June 2016) and late (July–December 2022) periods. Panels show distributions, smoothed trends and variability across key behavioural and political metrics. For the stance measure: climate action, techno-optimism, behavioural adjustment, cultural liberalism and economic collectivism, the natural scale ranges from −1 to +1. In contrast, the other measures: toxicity (bounded between 0 and 1), emotionality/reasoning (a ratio typically close to 1) and egocentrism (ranging from 0 upward), are plotted on their inherent scales. These scale choices preserve the natural variation in each metric. Each data point represents the monthly average for each group, smoothed via LOESS with 95% confidence intervals around the trend line (shaded regions).

US academics ($M$ = 0.055, s.d. = 0.1663, 95% CI [0.0536, 0.0564], $N$ = 51,633) hold more culturally liberal views than the general US population ($M$ = 0.0067, s.d. = 0.0608, 95% CI [0.0066, 0.0068], $N$ = 3,488,181); Welch's $t$-test ($t$(51,836.71) = 65.935, $P$ < 0.001, mean diff = 0.0483, $d$ = 0.29) indicates a small-to-moderate effect. Likewise, US academics ($M$ = 0.0184, s.d. = 0.1271, 95% CI [0.0173, 0.0195], $N$ = 51,633) express more support for economic collectivism than general US users ($M$ = 0.0007, s.d. = 0.0415, 95% CI [0.0007, 0.0008], $N$ = 3,488,181); Welch's $t$-test ($t$(51,795.5) = 31.584, $P$ < 0.001, mean diff = 0.0177, $d$ = 0.142) suggests a small effect.

Among US academics, climate action support declined when comparing the pre-pandemic period (2016–2019; $M$ = 0.077, s.d. = 0.292) with the post-pandemic period (2020–2022; $M$ = 0.067, s.d. = 0.264); Welch's $t$-test ($t$(51,344) = −3.94, $P$ < 0.001, diff = −0.010, $d$ = −0.03) shows a negligible effect. Economic collectivism rose modestly ($M$ = 0.017 to $M$ = 0.020; $t$(50,818) = −2.74, $P$ = 0.006, diff = −0.003, $d$ = −0.02, small effect). In the general US Twitter population, climate action support declined slightly ($t$(3,133,830) = −4.78, $P$ < 0.001, diff = −0.0005, $d$ = −0.01) while economic collectivism increased ($t$(2,586,863) = 15.47, $P$ < 0.001, diff = 0.0008, $d$ = 0.02), both negligible effects.

## Discussion

This paper introduces a dataset of academic expression online, documenting various descriptive patterns. Our findings show significant inequalities in content creation, engagement, followers and citations, reflecting the unequal nature of academic influence on Twitter. Notably, prominent academic influencers on social media are often not leading scholars, as demonstrated by the weak relationship between citation counts and Twitter metrics (that is, likes and followers), indicating a divergence between academic recognition and social media visibility that can potentially distort perceptions. Academics with high public reach on Twitter but limited subject-matter expertise show weaker support for climate action compared with high-reach experts, aligning more with average social media users. This suggests a division of labour in science communication but could also reflect individual incentives tied to popularity.

We take a descriptive approach, using AI tools to analyse expressed stances and their evolution over time. The narrative that academics online skew towards the political left is confirmed by their positive average net support for both cultural liberalism and economic collectivism. Such dynamics may be shaped by social media algorithms and could disproportionately influence public narratives and policy decisions, potentially creating a skewed representation of consensus within academia. While the consequences of this misalignment are unknown, it may contribute to perceptions of ideological distance between academic elites and ordinary citizens, which can be seized upon by populist politicians[30–32].

We observe a general decline in online toxicity. However, academics in the humanities display more toxic language than those in other fields, suggesting that toxicity is more prevalent in the humanities. Reference 33 has shown the alienating effect of toxicity on engagement, which is concerning for the dissemination of science. This concern is further compounded by our findings that academics at top-ranked institutions exhibit higher toxicity than those at lower-ranked ones, suggesting that even among highly credible scholars, elevated toxicity might contribute to audience alienation and potentially impede effective science communication.

Our findings echo concerns about ideological bias in social media, where progressive voices tend to dominate[34]. This aligns with the analysis of ref. 35 of political affiliations in academia, suggesting that Twitter may amplify these biases. Studies such as ref. 36 on digital moral outrage explain how online environments lower barriers for expressing outrage, increasing its frequency and intensity. This supports our findings that academic discourse on highly polarized topics, such as climate action, becomes more extreme and morally charged on social media. The incentives of platforms such as Twitter, which reward emotionally charged content with greater visibility, may further exacerbate this polarization. Similarly, ref. 14 find that a small, highly active subset of social and computer scientists disproportionately shapes conversations on Twitter, paralleling our findings. Reference 37 echoes concerns about algorithmic amplification, where non-expert academics with large followings may distort the perception of scientific consensus.

We find notable geographical differences among academics. US-based scholars exhibit higher levels of egocentrism and toxicity alongside comparatively weaker support for climate action, whereas non-US academics tend to express less culturally progressive views.

We also find notable differences between fields. For instance, STEM scientists exhibit higher climate-action support than scientists in social sciences and humanities. STEM is also more techno-optimistic than social sciences or humanities. Meanwhile, social sciences show slightly higher economic collectivism than other fields. This diversity may have implications for public perceptions of consensus, with risks of political bias in expert selection.

There are several limitations to consider. While our focus is on genuine expressions rather than engagement metrics such as retweets or likes, algorithmic amplification could still influence the visibility and prevalence of certain topics, potentially encouraging more political expression among academics and shaping broader discourse. In addition, our assumption of a one-to-one correspondence between Twitter users and academics may not always hold, as some manage multiple accounts or use pseudonyms. However, the large sample size and consistent trends across subgroups suggest that the key patterns remain reliable despite these potential mismatches. While Twitter is not representative of the general population or all academics, research shows it has a significant influence on public opinion, media coverage and policy debates. Tweets from influential figures can increase xenophobic sentiment and hate crimes[38], shape monetary policy and financial markets[39], even catalyse bank runs[40], and shape editorial decisions in mainstream news[1]. Despite its limitations, Twitter's demonstrated influence on public opinion and socio-economic outcomes show the value of analysing the behaviour of academics on the platform. While we recognize the potential influence of confounders such as algorithms or external events, our use of a balanced panel of academics and a sampling strategy that limits the influence of frequent posters ensure that our longitudinal comparisons remain robust and help mitigate the risk of skewed results caused by a small number of highly active users.

In addition, our topic detection relies on LLM-augmented dictionaries derived from iterative, prompt-based methods. Although this approach is cost efficient and adapts to the evolving language of Twitter, it may underclassify relevant tweets if differences in classification occur across groups (for example, users with high versus low follower counts). Alternative methods, such as keyword augmentation, could potentially mitigate this risk, as noted in ref. 41. While our analysis focuses on relative trends rather than precise quantitative levels, these methodological choices may influence the absolute measures of stance and topic prevalence.

Importantly, this study is descriptive, documenting trends without asserting causal relationships. We highlight differences in academic expression across subgroups but refrain from causal claims. Future research could investigate drivers behind these variations, such as institutional incentives or visibility, and explore how algorithms may amplify certain voices and shape public perception of science.

The over-representation of certain views—especially from high-reach, low-expertise academics, who express lower support for climate action and cultural liberalism—combined with under-representation of experts, could result in miscommunication of scientific consensus. This suggests that many 'public intellectuals' gain visibility more through public engagement than traditional academic influence, reinforcing that online prominence does not necessarily reflect expertise. Thus, those with the greatest public reach may not represent top scholars, potentially distorting public perceptions.

It is important to ensure that academics' engagement with social media enriches rather than distorts public discourse. Further research should quantify the impact of these ideological divisions on public trust and explore strategies to mitigate biases in academic communication on social media. Moreover, it is crucial to investigate 'why' academics express themselves politically, considering motivations such as name recognition, ideological drive, or the desire to share knowledge.

## Methods

### Ethics and inclusion statement

Our study was reviewed and approved under the Science Engineering Technology Research Ethics Committee (SETREC) process at Imperial College London (SETREC protocol number: 7296355). The protocol was approved on 2 July 2024 after review by the Head of Department and the Research Governance and Integrity Team. As no significant ethical issues were identified, the study did not require full committee review. It complies with the Data Protection Act 2018 and General Data Protection Regulation, and adheres to the ethical standards of social media research, utilizing only publicly accessible data from Twitter and OpenAlex. Data collection was performed using platform APIs in line with their terms of service. Access to the Twitter data was obtained through the Academic Research access provided by Twitter, which requires a formal application and approval process. OpenAlex data were accessed directly via its public API, which does not require additional permissions.

We acknowledge that the public nature of data on social media platforms such as Twitter does not negate the importance of safeguarding privacy. Although the content analysed was publicly available, the study employed anonymization protocols to prevent the identification of individuals. No personal identifiers, including names, usernames, or tweet IDs, are mentioned in the paper or in the publicly available dataset provided in the replication package. The dataset shared includes anonymized individual IDs. All data storage and processing were conducted using encrypted methods to protect the information's integrity.

### Twitter and OpenAlex data and measurement

We started with data on the social network of 300,000 academics on Twitter provided by ref. 42. This dataset includes a researcher's OpenAlex identifier matched to their Twitter/X user identifiers with high precision and moderate recall. OpenAlex, built upon the Microsoft Academic Graph, offers extensive data on researchers' publications, citations, affiliations, co-authors, fields and related academic metrics. OpenAlex is the largest open-access dataset on works, authors and metascience data. Compared with other bibliometric databases such as Scopus and Web of Science (WoS), OpenAlex typically outperforms in coverage, indexing a broader range of publications and journals, including those that are hard to find elsewhere[43]. Studies have shown that OpenAlex covers ~90% and 75% of all articles found in Scopus and WoS, respectively, and often includes significantly more works[44,45]. Although some limitations exist, such as occasional missing or incorrect data, OpenAlex remains a robust tool for bibliometric and metascience research, particularly for data from 2016–2022[46,47].

Using the matched Twitter user identifiers, we collected data on social media activity using Twitter's Academic API, which grants access to historical tweets for approved research purposes. The data were collected in January 2023 and include the full timelines of

academics, including (1) original posts (tweets), (2) shared posts of others (retweets), (3) quoted retweets and (4) comments (replies). In this paper, any mention of tweets refers to all four types of tweet mentioned above. OpenAlex data, being publicly available, were accessed via its API without additional permissions.

We defined two sets of users on the basis of their activity during the sample period. The balanced set includes academics who have tweeted at least once in the first 6 months of the sample period (January 2016 to June 2016) and at least once in the last 6 months of the sample period (July 2022 to December 2022). This ensures consistent presence across the entire period. The unbalanced set includes all academics who have tweeted at least once at any time between January 2016 and December 2022, without requiring activity in both the beginning and end of the sample period. In theory, one could extend the sample period back to 2007, but this would result in a very small balanced set of users. Our balanced sample includes 99,274 academics. Descriptive statistics at the author level are available in Table 1 and at the tweet level in Supplementary Table 8. These include basic Twitter and OpenAlex-based metrics as well as additional derived measures described in the rest of the section. The balanced and unbalanced samples have similar publication metrics, gender distribution and discipline distribution; however, the unbalanced sample is disproportionately noisier in Twitter engagement metrics, as detailed in Supplementary Table 9. Importantly, all analyses in this paper were conducted using the balanced sample only.

To measure the representation gap in the United States (in Fig. 6), we used a dataset of user IDs from a random sample of the US Twitter population, created by ref. 48. The user IDs were directly provided by one of the authors, Pablo Barberá, who granted permission for their use. Tweets and user metadata were subsequently collected using Twitter's Academic API. For this analysis, we restricted our academic sample to US-based academics, thereby ensuring a direct comparison with US users. To match the balanced nature of our US-based academics sample, we ensured users in this general US population sample have tweeted at least once in the beginning and end of the sample. The balanced sample of US users included 61,259 users. We employed the same topic and stance detection steps on their full set of posts during this period.

We also removed bots from the general US Twitter users sample, using a very conservative definition of a bot, that is, if the bot probability is >50%. Approximately 5% of users were considered bots if they have a bot probability >50%, 3% if probability is >80%, and 1.4% if probability is >0%. Bot detection was performed using the 'tweetbotornot2' R package, which provides an out-of-the-box classifier for detecting Twitter bots. This classifier uses various features from Twitter user profiles and activity patterns to estimate the probability of an account being a bot. The classifier is designed to be easy to use, interpretable, scalable and high performance. More information on tweetbotornot2 can be found at https://github.com/mkearney/tweetbotornot2.

For our temporal analyses, we defined the 'pre-pandemic period' as the years 2016–2019 (inclusive) and the 'post-pandemic period' as the years 2020–2022 (inclusive).

## Topic detection

Each tweet per user was classified into one of the four predefined topics. To cover international, salient and policy relevant discussions, we retrieved topics related to abortion rights, racial equality, climate action, immigration, welfare state, taxation policy and income redistribution. The full corpus of tweets contained ~296 million tweets from 1 January 2016 to 31 December 2022 (138 million tweets of which come from the balanced sample of academics), including users with and without location. First, a topic detection step reduced the sample to ~22 million tweets, which is 3.1% of the full sample. In the balanced sample of users, this was 4.9 million tweets and 3.56% of the balanced sample (Table 2).

A challenge inherent in keyword dictionary-based methods is their static nature. They tend to lack the dynamic adaptability required

to capture the ever-evolving lexicon of topical discourse, especially on platforms such as Twitter. For instance, a static dictionary might overlook terms such as 'Paris agreement' when curating a list for the topic of climate change, even though it is pivotal to the discourse. To counteract this limitation, we employed the capabilities of the GPT-4 model by OpenAI to generate keyword dictionaries that can incorporate the dynamic and context-specific nature of natural language. The prompt engineering follows best practices[49], which emphasizes the importance of clarity and specificity in tasks given to language models, especially when aiming for cost effectiveness and accuracy in large-scale projects. We engineered the following prompt:

> "Provide a list of *<ngrams>* related to the topic of *<topic>* in the year *<year>*. *<twitter fine tuning>*. Provide the *<ngrams>* as a comma-separated list."

To derive comprehensive and pertinent terms, the above prompt was applied in an iterative manner, considering various key parameters (in angle brackets '<input>') that we now define. The '<topic>' under scrutiny comprised a diverse spectrum of salient topics including 'climate change', 'climate change policy', 'immigration', 'immigration policy', 'abortion rights', 'racism or race relations', 'welfare state', 'redistribution of income' and 'tax policy'. Both 'climate change' and 'climate change policy' were separately prompted to GPT-4 to ensure breadth of terms extracted for dictionaries. However, in all subsequent analysis, they were unified as one topic to capture overarching sentiment on combating climate change. A parallel approach was employed for 'immigration' and 'immigration policy', distinguishing them initially but aggregating for stance detection. The temporal frame of reference '<year>' ranged from 2016 to 2022. In addition, the prompt allowed for a variety of '<ngrams>', ranging from unigrams to quadgrams, ensuring a comprehensive capture of the diverse lexical patterns that may be characteristic of a topic or Twitter-speak.

A specific point of interest in this method was the inclusion of a '<Twitter Fine Tuning>' directive in the prompt. This either explicitly instructed the model to 'Focus on language, phrases, or hashtags commonly used on Twitter' or was left empty. The reason behind this was to guarantee the inclusion of unique Twitter jargon and vernacular, ensuring that our dictionary remained contextually relevant to the platform's discourse. Building on the approach in ref. 50, which employs LLM-TAKE for context-aware keyword extraction in product categories, we adapted prompts for Twitter-specific terms to account for evolving vernacular and year-specific events. This method is inspired by previous studies[41], which highlight the limitations of static dictionaries in machine-learning models for capturing temporal shifts in language, emphasizing the potential of LLM-based techniques for real-time adaptability.

Multiplying the full set of prompt choices together gave ~432 unique combinations of prompts to ensure our dictionary was exhaustive. An illustrative example of the possible combinations is the prompt: "Provide a list of bigrams related to the topic of climate change in the year 2017. Focus on language, phrases, or hashtags commonly used on Twitter. Provide the bigrams as a comma-separated list". To process the combinations of responses, we took the union of the terms at the topic level, keeping only unique terms per topic. By doing this, each topic dictionary intrinsically incorporated terms from all the years, ngrams and vernacular types (Twitter specific or general). This approach ensured the inclusion of terms that, once introduced, persist in relevance over subsequent years. For instance, discussions using the term 'Paris Agreement' remain significant in 2022; hence they form an essential part of the overall climate change policy dictionary. Studies such as ref. 51 emphasize that contextual grounding in LLMs improves temporal and cultural relevance in keyword generation processes, especially in evolving domains such as Twitter. The same advantage was observed in our iterative, year-based prompt design, enabling effective tracking

of significant terms (for example, Paris Agreement) over time. Similar methodologies have shown improved accuracy when embedding context dynamically, proving beneficial in volatile discourse settings. A random sample of terms for several topics is provided in Supplementary Table 10, demonstrating key terms such as '#prolife' for abortion rights and '#netzero' for climate action. We used a temperature of 0.8 for this generative task. The relatively high temperature reflects the necessity to maximize recall even at the expense of precision.

In the next step of stance detection, we filtered out tweets classified as 'unrelated' to the target topic to reduce overclassification. While alternative methods such as keyword augmentation might further refine topic extraction, our analysis focused on relative trends rather than absolute measures, mitigating the impact of any residual misclassification. The use of LLM-augmented dictionaries, with a high temperature to capture Twitter's creative expression, is a cost-effective strategy for retrieving topic-relevant information from a large corpus.

Once applied to the full corpus, we reduced the size of the dataset to ~3.56% of full tweets. To contextualize academic discourse on Twitter, we identified tweets mentioning political figures such as 'Donald Trump' and 'Joe Biden', as well as scientific research papers. While our main focus is on policy-related topics, we found that 16.70% of academic tweets mentioned research, compared with 6.74% that mentioned a politician or candidate. Tweets about politicians constitute 40% of the volume relative to research mentions, highlighting the significant attention political figures receive among academics. These findings are intended solely to illustrate the broader context of academic engagement on Twitter.

### Stance detection

Having filtered the original corpus to a more focused subset of 4.9 million topical tweets using our dictionary-based approach, the next crucial step was to ascertain the stance each tweet holds towards its respective topic. We employed OpenAI's GPT-3.5 Turbo, known for its robustness and speed in zero-shot text classification tasks. While we used GPT-4 for generating topic dictionaries, we chose GPT-3.5 Turbo here for its optimal cost-accuracy trade-off. The GPT-4 task was more generative and only ~400 API calls. The stance detection step required one API call for each tweet; hence we chose GPT-3.5 Turbo since it is ~20 times cheaper than GPT-4 at time of classification. To guide the model efficiently, the following direct prompt was designed:

"Classify this tweet's stance towards <*topic*> as 'pro', 'anti', 'neutral', or 'unrelated'. Tweet: <*tweet*>."

Several aspects of this prompt deserve further explanation: (1) Choice of classification categories: the categories 'pro', 'anti' and 'neutral' are self-explanatory, capturing supportive, opposing and neutral stances, respectively. Classifying stances into these categories are common in both empirical[52] and recent theoretical work[53]. The 'unrelated' category serves as a corrective measure: despite the dictionary-based filtering, some tweets may still not be directly related to the topic in question. Identifying and excluding these ensures that the stance-detection results are both precise and relevant. (2) Efficiency considerations: given the massive number of tweets to be processed, the prompt was optimized for brevity without compromising clarity. This is not only cognizant of computational costs but also aligns with the model's strength in handling concise and direct tasks[49]. (3) Iterative refinement: preliminary tests were conducted on a smaller subset of tweets to refine the prompt for optimal results. Such an iterative approach is generally useful in ensuring that the chosen prompt structure yielded reliable stance categorizations.

The iterative refinement involved testing on a sample of 10 random prompts at a time to ensure consistency of output. The first prompt involved simply prompting "What is the stance of the tweet <*tweet*> towards the <*topic*>". Eventually we realized the following: (1) the importance of putting the entire instruction before entering tweet as a separate sentence. This ensures separation of tweet from instructions; (2) for purpose of pre-processing, it is useful to provide the output format by specifying the available choices explicitly; (3) inclusion of 'unrelated' helps limit the LLM response output to one word when the tweet was either completely unrelated or very hard to determine. This step also ensured that if the topic detection step over-retrieved tweets for any given topic, the stance detection step filtered out those tweets. A comparable error-filtering technique is seen in ref. 54, where context-aware LLM prompting dynamically adjusts to exclude false positives. By including 'unrelated' as a classification category, our method operates similarly to Grounding-Prompter, which ensures that domain-specific outputs remain accurate by refining prompts on the basis of evolving context. This approach, when combined with filtering, significantly enhances precision; (4) starting the prompt with 'Classify' ensured that the LLM gets clear and direct imperative instruction, making it well behaved and less random.

We used a temperature of 0.2, which is close to being a deterministic model. This is to minimize the measurement error, but allow a slight level of creativity in the model to interpret nuanced expressions. Given that the prompt includes the entire tweet, the model requires relatively less 'outside the box' thinking compared with the generative task of GPT-4.

We conducted a validation of the stance detection method using the publicly available SemEval-2016 Task 6 dataset provided by the Association of Computational Linguistics[52]. The dataset was accessed through the authors' official website (https://www.saifmohammad.com/WebPages/StanceDataset.htm) without requiring additional permissions. We present the results in Supplementary Note 1. Briefly, we found that our stance detection method using GPT-3.5 Turbo achieved very high F1 scores of ~84–92, depending on the topic (see Supplementary Table 11). In addition, a periodic manual validation was conducted on random samples to ascertain the accuracy of the stance classifications. Any systematic biases or errors identified during these checks would be vital feedback for potential refinements in subsequent iterations or studies.

Using our stance labels for each topical tweet, we can calculate a net stance metric at a suitable level of aggregation. Let $S_{um}$ represent the net stance towards climate action or immigration, aggregated at the user by year–month level. In the case of climate action, positive values mean that the user is on average pro-climate action (that is, supports the idea or policies of taking action to mitigate climate change or protect the environment):

$$S_{um} = \frac{\text{pro}_{um} - \text{anti}_{um}}{\text{pro}_{um} + \text{anti}_{um} + \text{neutral}_{um}} \quad (1)$$

Here, $S_{um}$ denotes the fraction of tweets about climate change or a climate change policy exhibiting a net pro stance by user $u$ in time-period $m$ relative to all climate-related tweets by the same user. The numerator quantifies the tweets that express a pro stance for the topic during the specific time frame. The denominator encompasses all tweets by the user about the topic, ensuring a metric that reflects subtle shifts in expression of opinion and allows comparisons across different timelines and users.

Since stance detection requires creating separate prompts for each tweet, we reduced total API costs by sampling the total set of topical tweets. We limited the sample to up to three tweets per academic per topic per year per month. This gave us enough tweets per period to classify the direction of political expression for each author. This sampling approach was intentional and served as a balancing criterion to create a balanced panel. The unit of analysis was academic–time, not the tweet, ensuring that our dataset was not distorted by the few extreme cases where some individuals might tweet excessively. Although we capped the sample at three tweets per author per topic per month for stance

detection, 99.38% of cases involved just one tweet, with only 0.62% involving two and 0.006% involving three. Thus, increasing the cap would affect less than 0.006% of the observations, making any impact on results negligible.

We also limited our analysis to only tweets labelled English by the Twitter API. This ensured that our measures did not suffer from any differences in measurement across languages. For analysis using the stance metrics, we continued to ensure that we have a balanced sample of political expression. This meant that we had to exclude academics that did not tweet politically about a topic in the first 6 months or last 6 months of the sample. This resulted in a total of 26,555 academics out of the overall balanced sample of 99,274. We show results from the topic and stance detection step in Table 2. For instance, there were 2,057,187 tweets about climate action and 734,692 tweets about immigration. The sampling procedure reduced them to 322,519 and 233,657, respectively. We also provide over-time plots of the average and total quantity of tweets with stance made in each topic in Supplementary Figs. 15 and 16.

Our topic and stance detection methods are flexible enough to allow researchers to focus on pre-specified topics. Certain topics are correlated, such as welfare and taxation, or immigration and racism. To reduce dimensionality, we created two 'normative' dimensions of political stances: cultural liberalism is the average of the net stances on racial equality, abortion rights and immigration; economic collectivism is the average of the net stance on welfare state, redistribution of income and wealth, and higher taxes for the wealthy. Both these measures vary at author × time level from −1 (always anti) to 1 (always pro). Table 1 provides author-level summary statistics on these stances and other author-level variables used in our paper.

### Narrative extraction

So far, we detected tweets belonging to the topic 'climate change' and 'climate change policy', and we detected the stance of the author on that topic in directional terms: pro, anti, or neutral towards 'climate action'. Within a supportive stance towards climate action, we identified two broad stances: techno-optimism and behavioural adjustment. These stances are our own classifications, developed to capture the narratives expressed in tweets.

An individual's expression might align with either or both stances. For example, someone might propose a behavioural solution that involves technological innovation, such as consumer behaviour related to electric vehicles or household solar photovoltaic (PV) installations. We were agnostic about the directional stance on climate action when classifying these narratives. Thus, we classified the narrative of all climate action tweets into one of four categories: (1) techno-optimism, (2) behavioural adjustment, (3) both, or (4) neither. We allowed people to hold both stances, which occurs, for instance, if one proposes a behavioural solution that involves technological innovation, such as consumer behaviour related to electric vehicles or household solar PV installations. We also allowed for narratives unrelated to the two stances. We employed a similar method as the stance detection step, using the model GPT-3.5 Turbo, a temperature of 0.2 and the following prompt:

> "Review the tweet provided and determine if it aligns more with the perspective of a Technology Optimist (who focuses on technological innovations and advancements as key solutions to climate change), a Behavioural Adjustment Advocate (who emphasizes the importance of individual and societal behavior changes for environmental sustainability), both, or neither. Use the following format for your response: 'Classification: [Technology Optimist/Behavioural Adjustment Advocate/Both/Neither]
>
> Tweet: '<*tweet*>'."

This prompt explicitly defined the two stances to minimize any ambiguities. We also show results from this narrative detection step in the bottom panel of Table 2. We found that 26.55% of climate action tweets were aligned with behavioural adjustment, 14.05% with techno-optimism, 12.87% with both of them, and 46.53% with neither.

Similar to our net stance metric $S_{um}$, we could create aggregate measures of climate narratives across academics. Let Behavioural$_{um}$, Techno$_{um}$, Both$_{um}$ and Neither$_{um}$ be counts of tweets by academic $u$ in a year–month $m$ about one of the four topics. Then $B_{um}$ is the proportion of all climate tweets made by an author in a given year–month that are advocating (exclusively) for behavioural change for environmental sustainability:

$$B_{um} = \frac{\text{Behavioural}_{um}}{\text{Behavioural}_{um} + \text{Techno}_{um} + \text{Both}_{um} + \text{Neither}_{um}} \quad (2)$$

Likewise, we could construct the proportion techno-optimistic for each user–time:

$$T_{um} = \frac{\text{Techno}_{um}}{\text{Behavioural}_{um} + \text{Techno}_{um} + \text{Both}_{um} + \text{Neither}_{um}} \quad (3)$$

We could then construct a relative measure called net techno-optimism (NT$_{um}$), which is the proportion 'techno-optimism minus the proportion behaviouralism':

$$\text{NT}_{um} = \frac{\text{Techno}_{um} - \text{Behavioural}_{um}}{\text{Behavioural}_{um} + \text{Techno}_{um} + \text{Both}_{um} + \text{Neither}_{um}} \quad (4)$$

Note that in the above three measures, we have ignored the intersection set Both$_{um}$ since our goal was to capture the relative differences across our units of analyses.

### Tone and style of expressions

We extracted measures of egocentrism, toxicity and emotionality/reasoning from an individual's tweets. These are relevant since political messages are relatively rare phenomena—only ~3.56% of all tweets in our academic sample belong to topics of immigration or climate action. It may be important to understand the general behavioural way an individual expresses themselves to add an additional layer of context to their political opinions.

We used Perspective API from Google to label a user's toxicity of conversation. While perspective API can handle multiple languages, it performs the best in English. We limited the sample to English tweets. Since the sample is still very large, we limited to up to 10 random tweets per year per user from 2016 to 2022. This ensured that there is a time-varying component incorporated for each user while keeping the sample small enough to match the API rate limits. For each tweet, we obtained the following attributes: toxicity (used in this paper), severe toxicity, insult, attack on identity, profanity and threat. For each attribute, we obtained a probability value between 0 and 1. Toxicity was defined as 'a rude, disrespectful, or unreasonable comment that is likely to make people leave a discussion'. There are many measures of toxicity. We used Perspective API (https://perspectiveapi.com/) as it is widely used in literature[55]. It is free to use by researchers and is used commercially for content moderation by a variety of publishers and platforms, including The New York Times. This measure is relevant because incivility or aggression in online discourse often arises when strongly held personal or political values are at stake, leading to heightened negative expressions[56–58]. For instance, group polarization can intensify antagonism toward outgroups, contributing to toxic rhetoric[58,59].

We measured egocentrism—a behavioural attribute that may be a relevant explanatory variable in explaining pro-social views. The hypothesis is that the more egocentric one's expression is, the less likely they are to prioritize the collective good. We measured this using another dictionary-based approach. We did not augment the dictionary

with LLM or used an expert dictionary since this is a relatively simple classification used in previous work in psychology and political science[22,23,60–63]. We counted the average number of self-focused words (I, me, my, myself, mine) per tweet. The measure is most helpful in relative terms. The higher this measure is relative to peers or another time period, the more egocentric an individual's language is, relatively. We used the full sample of English tweets for calculating egocentrism, since these are basic string detection tasks that are straightforward to implement on any SQL database. We emphasize that this measure is most conservatively measuring singular self-references and can also be interpreted as self-centredness. Egocentric expressions may not necessarily be negative. However, we show correlates of egocentrism in Supplementary Fig. 7 and found that toxicity is positively correlated with egocentrism.

The next measures of relevance are the spectrum of emotion to reason. Aristotle homed in on two features of persuasion: affect (pathos) and cognition (logos). With political debate featuring a mix of persuasive elements, the 'how' of what is being said might offer a signal for future opinions. Inspired by previous work[64], we measured emotionality of a piece of text on a scale from fully cognitive to fully affective. To measure cognition and affect, we used Linguistic Inquiry and Word Count (LIWC)[65]. LIWC contains a set of categorized dictionaries, from which we chose the 'cognitive processing' and 'affective processing' categories. The cognitive dictionary is designed to detect concepts of insight, causation, discrepancy, tentativeness, certainty and differentiation, while the affective dictionary is designed to pick up emotions, moods and other positive and negative sentiments. Equation (5) constructs a basic score of emotionality/reasoning, which can be constructed for each tweet or at a suitable level of aggregation.

$$\text{Emotionality} = \frac{\text{Affective Terms} + 1}{\text{Cognitive Terms} + 1} \qquad (5)$$

For instance, at an individual × time level, we computed the number of affect terms used by that individual in that time period plus one, divided by the number of cognitive terms made by that individual in that time period plus one.

### Gender

Gender is not directly available through OpenAlex or Twitter. To infer gender, we extracted the official names of academics as provided by OpenAlex. We passed the names individually through an LLM (OpenAI's GPT-3.5 Turbo model) in similar fashion as for tweets. We used the following prompt:

"Based on the given name, please indicate the most likely gender association. Name: '<*name*>'. Respond as 'Male', 'Female', or 'Unclear.'"

Once again, all the usual best practices of LLM prompting were applied. The three output categories were chosen to simplify the analysis. Any names that were not classified into 'Male' or 'Female' were classified as 'Unclear'. Since some names may be hard to classify, the 'Unclear' category can capture this. Applying this prompt to the full sample of academics, we obtained the following classification: 49% male, 49% female and 2% unclear. In the balanced sample of users, we found that 60% are male (see Table 1).

### Location, institutional affiliation and rankings

OpenAlex provides article-level institutional affiliation data, which we used to capture time-varying institutional affiliations for academics from 2016 to 2022. We accessed all publications by academics within this period through OpenAlex and extracted the institution name from the author metadata for each publication. By relying on an article-level affiliation data, we ensured that our analysis reflects changes in an author's affiliation over time, rather than a single, static affiliation. We used yearly aggregation based on the modal affiliation of each academic to maintain consistency.

We also ranked institutional affiliations that are purely higher education. OpenAlex provides the type of affiliation. In our sample, 71% of authors belong to higher education (for example, Harvard University), 9.1% in healthcare (for example, Brigham and Women's Hospital), 6.2% in facility (for example, The Francis Crick Institute), 4% non-profit (for example, Wellcome Trust), 2.6% government (for example, European Commission), 1.9% in companies (for example, Microsoft Research) and the rest in other categories. To assess the rankings of these institutional affiliations, we used the Times Higher Education World University Rankings from 2016 to 2022, accessed via Kaggle (https://www.kaggle.com/datasets/raymondtoo/the-world-university-rankings-2016-2024), where the data are publicly available and accessible. These yearly rankings allowed us to classify institutions into four categories: 1–100, 101–500, 501–1,500 and 1,501+ for each year within our study period. In cases where an academic did not publish in certain years, we generated a complete set of institution–year pairs, filling any gaps by carrying forward the nearest available ranking and backfilling as necessary to ensure continuity. This process was similarly applied to determine whether an academic was affiliated with a US institution in a given year.

Often, there are differences in how university names are mentioned, for example, University College London vs UCL, so we first employed an exact match, obtaining almost 50% matches. Then, we pre-processed the names and used cosine similarity for fuzzy matching using a strict distance threshold of 0.05. After applying fuzzy matching, we were able to obtain a ranking for 84.4% of high-education-affiliated authors in our sample. In terms of ranking group distribution, 24.4% of academic–year entries were from universities ranked 1–100, 26.4% were from universities ranked 101–500, 8.9% were from universities ranked 501–1,500, and 40.6% were from universities ranked 1,501+.

### Concepts

We determined the field of study the author works in on the basis of a machine-learning classification of their respective works. OpenAlex describes 'concepts' as abstract ideas that a work is about: 'Concepts are hierarchical, such as a tree. There are 19 root-level concepts and six layers of descendants branching out from them, containing about 65 thousand concepts all told'. OpenAlex classifies each work with a high level of accuracy. More details can be found at https://docs.openalex.org/api-entities/concepts. We combined the 19 root-level concepts into 3 broad categories for ease of comparison: Humanities, Social sciences and STEM. Humanities include Art, History, Philosophy, Literature, Religion, Music, Theater, Dance and Film. STEM includes Biology, Chemistry, Engineering, Environmental science, Geography, Geology, Materials science, Mathematics, Medicine and Physics. Social sciences include Business, Economics, History, Political science, Psychology and Sociology.

Each work by an author could belong to multiple concepts and a score from 0 to 1 was given to each concept, where values closer to 1 reflects the likelihood that the work belongs to that concept. For each academic, we took the average score for all the root-level concepts across all their works from 2016–2022. We then picked the primary concept of that academic as the root-level concept with the highest average score.

Each work may be assigned to multiple concepts, with a score between 0 and 1 indicating the likelihood that the work belongs to that concept. For each academic, we computed the average score for each of the 19 concepts across all their works from 2016–2022, and then selected the primary concept as the one with the highest average score. Because the concept assigned to any given work can vary over time, some academics exhibited substantial scores in more than one field. In our data, 134,729 academics were assigned exclusively to one category, while 1,247 academics (~0.9% of those with any field assignment) had

overlapping field memberships. For our cross-sectional analyses, we excluded the overlapping cases to ensure mutually exclusive group comparisons.

### Citations and impact factor

We measured academic impact using two author-level metrics from OpenAlex: citations and impact factor (2-year mean citedness). Both measures were based on data as of January 2023, the end of our sample period.

Citations reflect the total number of works that cite any work authored by an individual, representing their cumulative citation stock. This measure captures lifetime scholarly influence, independent of career length or discipline.

The impact factor measures the mean number of citations received for works published 2 years before the current year. It is calculated as the number of citations received in the previous year (numerator) divided by the number of citation-receiving publications from the two preceding years (denominator).

### Credibility, expertise and Twitter reach

We used a researcher's publication impact as a proxy for their academic credibility. Specifically, we classified an academic's credibility as 'high' or 'low' on the basis of whether their impact factor is above or below the median impact factor for their respective field. This field-specific comparison accounted for discipline-specific citation norms, ensuring that researchers were evaluated relative to their peers within the same academic discipline (for example, Biology vs Political science).

In addition to overall credibility, we distinguished subject-matter (or domain) expertise. An academic was considered a subject-matter expert if their recent publications (2016–2022) aligned significantly with the relevant topic under study. For example, when analysing stances on climate action, an academic was classified as a climate expert if their primary concept (that is, the root-level concept with the highest average score across their works) was one that corresponds to environmental or climate science. Researchers working in unrelated fields but discussing climate issues were coded as non-experts for these analyses.

We measured an academic's social media influence using their total number of Twitter followers. Academics with follower counts above the median in our sample were classified as 'high reach', while those below the median were considered 'low reach'. This measure captured the potential audience size for an academic's tweets, although it does not necessarily reflect engagement metrics such as retweets or likes.

In our subgroup analyses, we used terms such as 'high-reach, low-credibility academics' or 'low-reach, high-credibility experts' to combine these definitions. 'High-reach' and 'low-reach' referred to whether an academic's Twitter follower count is above or below the median in our sample. 'High-credibility' and 'low-credibility' correspond to whether an academic's impact factor is above or below the median in their field. Finally, 'expert' and 'non-expert' distinguished whether an academic's publications aligned substantially with the relevant topic (for example, climate action). By distinguishing credibility (impact factor), domain expertise (field-specific knowledge) and social media reach (follower count), we avoided conflating an academic's general scholarly profile with their topic-specific specialization or their online popularity.

It is important to note that we used the term 'credibility' when discussing tone and style (the 'how') measures, such as egocentrism, toxicity and emotionality/reasoning, since general academic recognition may influence communication style. Conversely, we used 'expertise' when analysing stances on political topics (the 'what'), as domain expertise is often more relevant for understanding topic-specific expressions and positions. This distinction ensured clarity and precision in interpreting the roles of credibility and expertise across different dimensions of our analysis.

### Framework for statistical testing

All statistical tests in this study were two-tailed, as our aim was to remain conservative when assessing possible effects in either direction. For group mean comparisons, we used Welch's $t$-test, which does not assume equal variances across groups. Pearson's correlation coefficient ($r$) was used for association analyses. For linear regressions, including time-series models, significance levels ($P$) are reported for two-tailed tests, with 95% confidence intervals (CIs) to indicate the plausible range of effect sizes.

We systematically checked normality and variance assumptions for each comparison. Normality assessments involved evaluating skewness and kurtosis of each group's distribution, while Levene's test was used to assess homogeneity of variances. If Levene's test yielded a significant result ($P < 0.05$), we assumed unequal variances and applied Welch's $t$-test (for means) or robust correlation/regression methods as appropriate. Full results of these checks, including skewness, kurtosis, Levene's $F$-statistic and $P$ values, are documented in Supplementary Tables 2–5.

Because large sample sizes can make even mild distributional deviations appear statistically significant, Welch's $t$-test was adopted as the default for mean comparisons. When distributions were heavily skewed or kurtotic, the same Welch correction was used to ensure validity. Exact $P$ values are reported unless below 0.001, in which case we write $P < 0.001$.

### Reporting summary

Further information on research design is available in the Nature Portfolio Reporting Summary linked to this article.

## Data availability

The datasets generated and/or analysed during the current study are available on Zenodo at https://doi.org/10.5281/zenodo.15115397 (ref. 66). Due to Twitter's data-sharing policies, raw Twitter data cannot be publicly shared. However, aggregate measures derived from the raw data are included. The raw tweets are securely stored on Google's BigQuery SQL database and can be accessed upon request to the authors, subject to Twitter's policies. Data about academics from OpenAlex is publicly available and can be accessed via the OpenAlex API (see https://docs.openalex.org/ for documentation). Additional datasets used in this study include the following: SemEval-2016 Stance Dataset available at https://www.saifmohammad.com/WebPages/StanceDataset.htm; Gender Classification Dataset: combined from US Social Security Card Applications (1880–2019), UK Baby Names (2011–2018), British Columbia Baby Names (1918–2018), and Australian Baby Names (1944–2019), available at the UC Irvine Machine Learning Repository (https://archive.ics.uci.edu/dataset/591/gender+by+name); General Social Survey (GSS): data available at https://gss.norc.org/us/en/gss/get-the-data.html; random sample of US Twitter Users: User IDs provided by Pablo Barberá upon request, with tweets and metadata hydrated via the Twitter API; Times Higher Education World University Rankings (2016–2024): available on Kaggle (https://www.kaggle.com/datasets/raymondtoo/the-world-university-rankings-2016-2024). We commit to transparency by making a replication package, including anonymized data and analysis code, available to the academic community.

## Code availability

The code used to analyse the data and replicate the main results of the paper is available on Zenodo at https://doi.org/10.5281/zenodo.15115397 (ref. 66). This repository includes all scripts and documentation necessary to reproduce the analyses presented in this study. The repository also provides guidance on using R (v.4.3.0), and relevant packages used in the analysis.

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

## Acknowledgements
P.G. acknowledges funding from Imperial College London. T.F. acknowledges funding from the Leverhulme Prize in Economics, a European Research Council Starting Grant (ERC, MEGEO, 101042703), and the Deutsche Forschungsgemeinschaft (DFG, EXC 2126/1 – 390838866). The funders had no role in study design, data collection and analysis, decision to publish or preparation of the manuscript.

## Author contributions
P.G. and T.F. jointly conceptualized the study, contributed to data analysis, and produced the figures. P.G. collected the data. Both authors collaborated on drafting the paper, refining it, and providing final revisions.

## Competing interests
The authors declare no competing interests.

## Additional information
**Extended data** is available for this paper at https://doi.org/10.1038/s41562-025-02199-1.

**Correspondence and requests for materials** should be addressed to Prashant Garg.

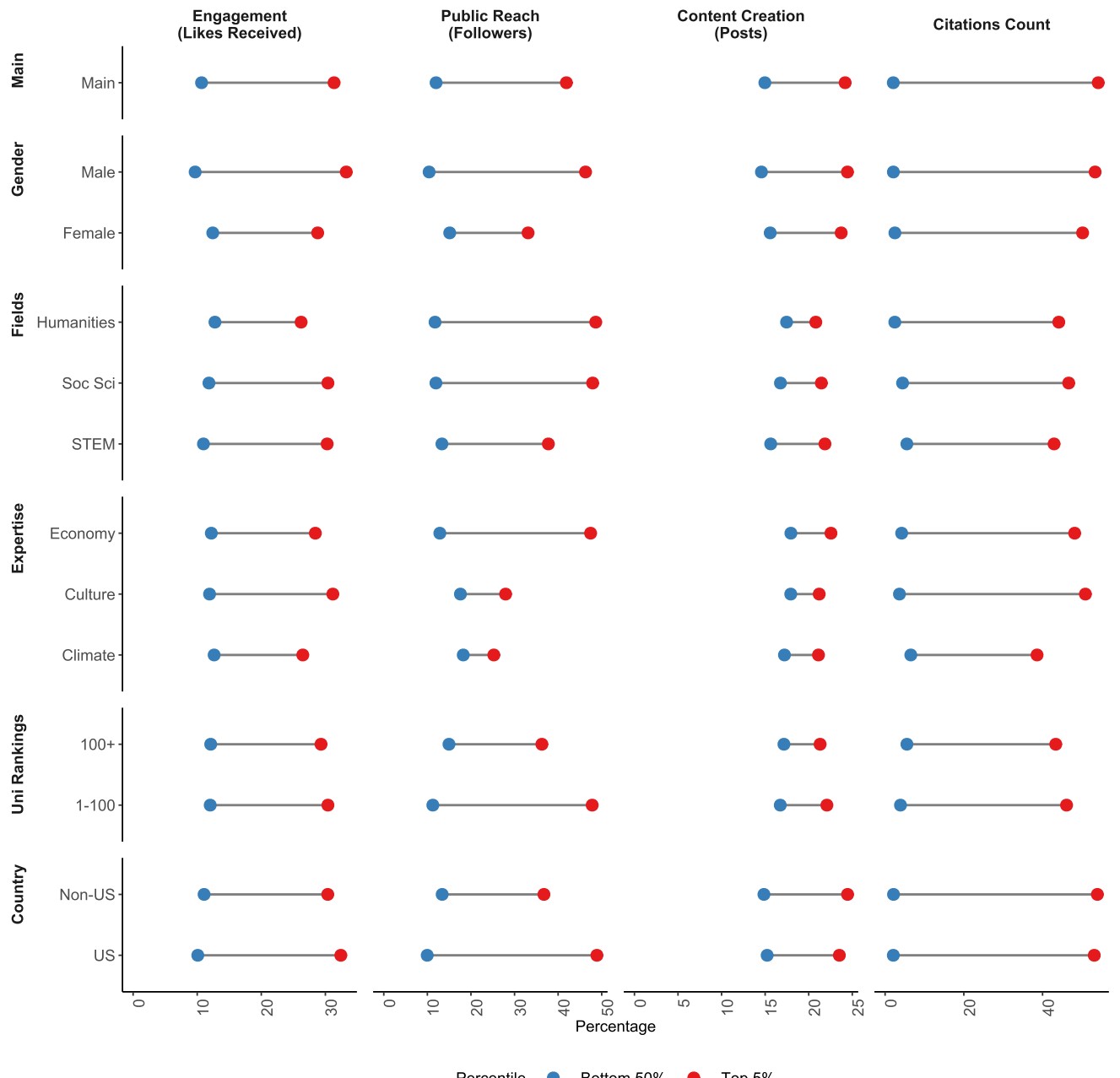

**Extended Data Fig. 1 | Inequality in Twitter Engagement by Subgroups.**
This figure shows subgroup differences in inequality across four key metrics: Engagement (Likes Received), Public Reach (Followers), Content Creation (Posts), and Citations Count. Each row represents a subgroup (Gender, Field, Expertise, University Rankings, Country), and each column corresponds to one of the metrics. Inequality is measured by the proportion of total activity attributed to the Top 5% (red points) and Bottom 50% (blue points) of users within each subgroup. Lines connect subgroups across metrics to highlight patterns. The x-axis shows the percentage of total activity (0–100%). Data were derived from 99,274 academics active on Twitter (2016–2022), capturing all tweet types (originals, replies, retweets, quoted tweets).

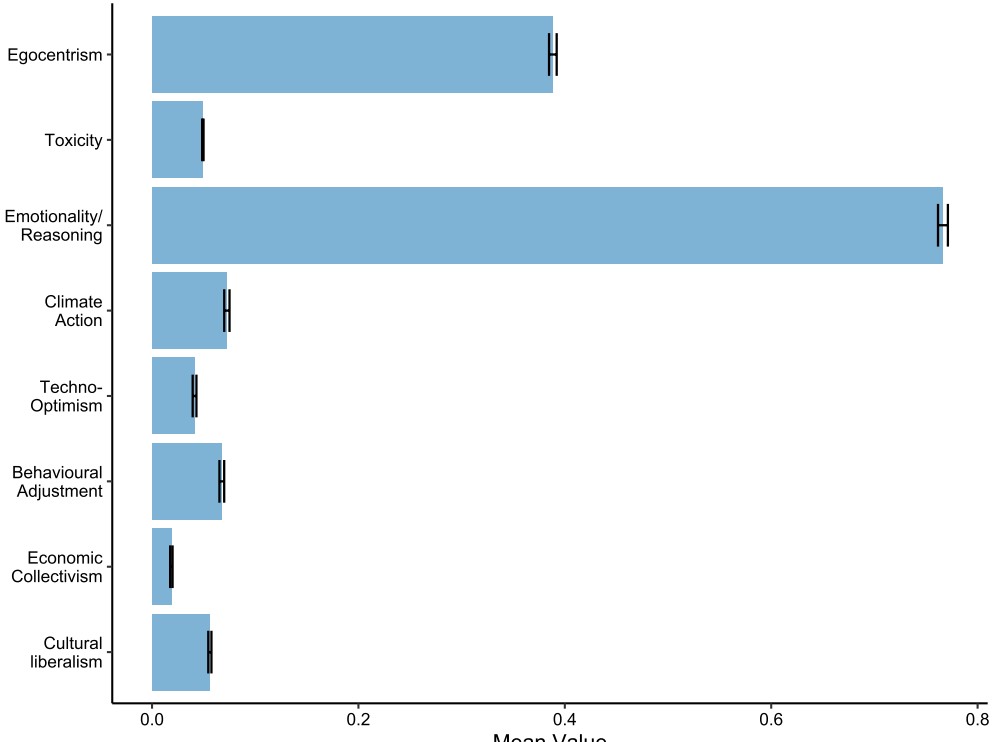

**Extended Data Fig. 2 | How U.S. Academics Express Themselves Online: Average Measures of Political Expression, Tone, and Style.** Data include 7,724 U.S.-based academics from a balanced panel (each academic tweeted both early and late during 2016–2022). Stances were classified via up to three tweets per user-month for measures such as Climate Action, Techno-Optimism, Behavioral Adjustment, Cultural Liberalism, and Economic Collectivism, while Egocentrism, Toxicity, and Emotionality/Reasoning were computed from all available English tweets (with Toxicity based on up to 10 tweets per user-year). See Methods for variable definitions and sampling details. Bars represent the mean stance on each measure, and error bars indicate ± 95% confidence intervals derived from standard errors.

# Reporting Summary

## Statistics

For all statistical analyses, confirm that the following items are present in the figure legend, table legend, main text, or Methods section.

| n/a | Confirmed | |
|---|---|---|
| ☐ | ☒ | The exact sample size (*n*) for each experimental group/condition, given as a discrete number and unit of measurement |
| ☒ | ☐ | A statement on whether measurements were taken from distinct samples or whether the same sample was measured repeatedly |
| ☐ | ☒ | The statistical test(s) used AND whether they are one- or two-sided<br>*Only common tests should be described solely by name; describe more complex techniques in the Methods section.* |
| ☐ | ☒ | A description of all covariates tested |
| ☐ | ☒ | A description of any assumptions or corrections, such as tests of normality and adjustment for multiple comparisons |
| ☐ | ☒ | A full description of the statistical parameters including central tendency (e.g. means) or other basic estimates (e.g. regression coefficient) AND variation (e.g. standard deviation) or associated estimates of uncertainty (e.g. confidence intervals) |
| ☐ | ☒ | For null hypothesis testing, the test statistic (e.g. *F*, *t*, *r*) with confidence intervals, effect sizes, degrees of freedom and *P* value noted<br>*Give P values as exact values whenever suitable.* |
| ☒ | ☐ | For Bayesian analysis, information on the choice of priors and Markov chain Monte Carlo settings |
| ☒ | ☐ | For hierarchical and complex designs, identification of the appropriate level for tests and full reporting of outcomes |
| ☐ | ☒ | Estimates of effect sizes (e.g. Cohen's *d*, Pearson's *r*), indicating how they were calculated |

*Our web collection on statistics for biologists contains articles on many of the points above.*

## Software and code

Policy information about availability of computer code

| | |
|---|---|
| Data collection | The data collection process for this study utilized several tools and software. We accessed Twitter's academic API using the R packages academictwitteR (version 0.2.3) and rtweet (version 0.7.0). The raw tweets were collected in January 2023 and included tweets from users who consented to the public availability of their data. To ensure compliance with Twitter's policies, only aggregate measures at the individual-by-time level are provided. The raw tweets are securely stored on Google's BigQuery SQL database and are accessible upon request, subject to Twitter's data-sharing policies. Additionally, data about academics were accessed through OpenAlex using the R package openalexR (version 0.1.1). The collection processes were conducted using R (version 4.3.0). |
| Data analysis | Data analysis was conducted using R (version 4.3.0) and Python (version 3.11.4). In R, the following packages were utilized: tidyverse (version 2.0.0) for data manipulation and visualization, magrittr (version 2.0.3) for piping operations, data.table (version 1.14.8) for high-performance data manipulation, purrr (version 1.0.1) for functional programming, scales (version 1.2.1) for formatting visualizations, psych (version 2.4.3) for summary statistics, broom (version 1.0.4) for regression analysis, car (version 3.1-3) for diagnostic testing, moments (version 0.14.1) for skewness and kurtosis metrics, and effsize (version 0.8.1) for effect size calculations. For text analysis, commercial machine learning models from OpenAI, such as GPT-4-turbo and GPT-3.5 Turbo, were utilized via the Python package openai (version 1.54.4) and the R package rgpt3 (version 0.4). Additionally, R package tweetbotornot2 (version 0.0.1) was used for bot detection on Twitter accounts. |

For manuscripts utilizing custom algorithms or software that are central to the research but not yet described in published literature, software must be made available to editors and reviewers. We strongly encourage code deposition in a community repository (e.g. GitHub). See the Nature Portfolio guidelines for submitting code & software for further information.

# Data

Policy information about availability of data

All manuscripts must include a data availability statement. This statement should provide the following information, where applicable:
- Accession codes, unique identifiers, or web links for publicly available datasets
- A description of any restrictions on data availability
- For clinical datasets or third party data, please ensure that the statement adheres to our policy

The datasets generated and/or analyzed during the current study are available at the following Zenodo repository (DOI 10.5281/zenodo.15115397): https://zenodo.org/records/15115397.

Due to Twitter's data-sharing policies, raw Twitter data cannot be publicly shared. However, aggregate measures derived from the data are provided. Raw tweets are securely stored on Google's BigQuery SQL database and can be accessed upon reasonable request to the authors, subject to Twitter's policies. Data about academics from OpenAlex is publicly available and can be accessed using the OpenAlex API (https://docs.openalex.org/) or the R package openalexR (version 0.1.1).

We also utilized several publicly available datasets:
- SemEval-2016 Stance Dataset: Available at https://www.saifmohammad.com/WebPages/StanceDataset.htm.
- Gender Classification Dataset: Compiled from the US Social Security Card Applications (1880–2019), UK Baby Names (2011–2018), British Columbia Baby Names (1918–2018), and Australian Baby Names (1944–2019), available from the UC Irvine Machine Learning Repository (https://archive.ics.uci.edu/dataset/591/gender+by+name).
- General Social Survey (GSS): Data available at https://gss.norc.org/us/en/gss/get-the-data.html.
- Random Sample of U.S. Twitter Users: Created by Siegel et al. (2021), with user IDs kindly provided by Pablo Barberá upon request. The tweets and metadata were hydrated using the Twitter API.
- Times Higher Education World University Rankings (2016–2024): Available on Kaggle at https://www.kaggle.com/datasets/raymondtoo/the-world-university-rankings-2016-2024.

This study provides a replication package, including anonymized data and analysis code, ensuring transparency and reproducibility. See Zenodo repository for details.

# Research involving human participants, their data, or biological material

Policy information about studies with human participants or human data. See also policy information about sex, gender (identity/presentation), and sexual orientation and race, ethnicity and racism.

| | |
|---|---|
| Reporting on sex and gender | In this study, the terms sex (biological attribute) and gender (shaped by social and cultural circumstances) were used carefully to avoid confusion. Gender was determined based on self-reporting via OpenAlex profiles and through the use of a large language model (OpenAI's GPT-3.5 Turbo) for name-based classification. The method is described clearly in "Methods" section in subsection 4.6, titled "Gender" and validated in the Appendix. The findings apply to both male and female genders, with 60% of the balanced sample identified as male and 40% as female. No data was collected for non-binary or other gender identities, and consent for sharing individual-level data was not applicable as we used publicly available information. The results show notable differences in political expression across genders, which are detailed in the paper. |
| Reporting on race, ethnicity, or other socially relevant groupings | Race, ethnicity, and other socially relevant categorization variables were not explicitly collected or analyzed in this study. The focus was on publicly available data regarding academic affiliations, social media activity, and publication records. The lack of race and ethnicity data was due to the constraints of the datasets used (Twitter and OpenAlex), which do not provide this information. The study did, however, examine variations in political expression across different countries, fields of study, and institutional affiliations. |
| Population characteristics | The study analyzes a sample of 99,274 academics from 12,675 institutions across 174 countries, covering diverse fields of study, including STEM (29.1%), Social Sciences (9.8%), and Humanities (0.5%). The sample includes academics active on Twitter from January 2016 to December 2022, ensuring a balanced representation with activity recorded in both early (January–June 2016) and late (July–December 2022) periods. |

While age data was not available, academic seniority was inferred from publication metrics, such as the number of citations (mean = 1,370.02, SD = 4,523.99), the 2-year impact factor (mean = 16.93, SD = 38.56), and the number of published works (mean = 57.58, SD = 114.38). These proxies provide indirect insights into the career stages of academics in the sample.

Social media activity levels varied widely: academics averaged 1,575 posts (SD = 2,052.13), 3,211 likes (SD = 5,736.33), 843 retweets (SD = 2,117.91), and 1,113 followers (SD = 9,145.94). Behavioral expression metrics include Egocentrism (mean = 0.33 words per tweet, SD = 0.27), Toxicity (mean = 0.04, SD = 0.04), and Emotionality/Reasoning (mean = 0.75, SD = 0.29).

The sample exhibited gender diversity, with 60% male and 40% female representation. However, data on non-binary or other gender identities was unavailable due to dataset constraints. Geographically, the dataset includes academics from six continents, with North America and Europe more prominently represented. Comparisons with the general U.S. Twitter population (61,259 users) were conducted to contextualize the findings.

Key political and behavioral metrics include stances on Climate Action (mean = 0.09, SD = 0.18), Cultural Liberalism (mean = 0.04, SD = 0.08), and Economic Collectivism (mean = 0.01, SD = 0.06). Among 26,555 politically active academics, 75% made at least one non-neutral tweet, with 37%, 56%, and 28% engaging on climate, cultural, and economic topics, respectively.

| Recruitment | Participants were included based on their presence in the OpenAlex database and their public Twitter activity. The sample was selected using the high-precision, moderate-recall matching of OpenAlex identifiers to Twitter user IDs (originallly done by Mongeon et al. [2023]), and their data was collected through Twitter's academic API in January 2023. This method ensured a comprehensive and representative sample of academics who engage in social media discourse. Potential biases include self-selection bias, as only academics who actively use Twitter are included, and the use of English-only tweets for certain analyses. |
|---|---|
| Ethics oversight | Under the Science Engineering Technology Research Ethics Committee (SETREC) process of Imperial College London, the study that has been reviewed by the Research Governance and Integrity Team and Head of Division/Department, where no significant ethical issues have been identified in the protocol or ethics application, can be given RGIT approval without requiring it to go to full committee. SETREC number: 7296355. |

Note that full information on the approval of the study protocol must also be provided in the manuscript.

# Field-specific reporting

Please select the one below that is the best fit for your research. If you are not sure, read the appropriate sections before making your selection.

☐ Life sciences  ☒ Behavioural & social sciences  ☐ Ecological, evolutionary & environmental sciences

For a reference copy of the document with all sections, see nature.com/documents/nr-reporting-summary-flat.pdf

# Behavioural & social sciences study design

All studies must disclose on these points even when the disclosure is negative.

| Study description | This study is a quantitative analysis of social media behavior, focusing on the political and behavioral expression of academics on Twitter. The study uses a novel dataset linking the social media content of around 100,000 academics to their academic records. The data includes various forms of tweets (original posts, retweets, quoted retweets, and replies) and covers a range of politically salient topics such as climate action, cultural, and economic concepts. The analysis employs large-language model-based classification techniques to categorize and label the content, assessing both the substantive content and the tone of the expression. |
|---|---|
| Research sample | The research sample consists of 99,274 academics from 12,675 institutions across 174 countries and 19 disciplines, as well as a random sample of 61,259 users representative of US Twitter population for comparison. The sample of academics was constructed using a dataset provided by Mongeon et al. (2023), which matched researchers' OpenAlex identifiers with their Twitter/X user identifiers. This comprehensive dataset captures the full Twitter activity of the selected academics from 2016 to 2022, ensuring a balanced sample. |
| Sampling strategy | The sampling procedure involved selecting academics who had tweeted at least once in the first six months of the sample period (January to June 2016) and the last six months (July to December 2022). This balanced sample includes those actively engaging on Twitter throughout the period. The statistical methods employed included scalable large-language model-based classification techniques to categorize and label the socially shared content. For the general US Twitter population, users were similarly included if they had tweeted at least once in the beginning and end of the sample period, ensuring a balanced comparison. These general users were randomly selected from US Twitter, and originally compiled by Siegel et al. (2021). |
| Data collection | Data was collected using the Twitter API (academic access) in January 2023. The data includes the full timelines of academics, encompassing original posts, retweets, quoted retweets, and replies. The Twitter data was linked to academic records from OpenAlex, providing comprehensive metadata about the academics' publications, affiliations, and research outputs. Additional data about general US Twitter users was also collected in January 2023 to measure the representation gap in political expression. The raw tweets are securely stored on Google's BigQuery SQL database. Since this study does not involve a randomized controlled trial, the researchers were not blinded to the study hypothesis or conditions |
| Timing | The data was accessed and collected during the third week of January 2023. The collection lasted around one week due to rate limits in Twitter API. |
| Data exclusions | Data exclusions involved removing bots from the general US Twitter user sample using the tweetbotornot2 R package. Bots were identified based on a probability threshold, with approximately 5% of users excluded if their bot probability was greater than 50%. Additionally, to ensure a balanced comparison, only users who had tweeted at least once in both the first six months (January to June 2016) and the last six months (July to December 2022) of the sample period were included in the analysis. This balancing criterion was applied to both the academic and general US Twitter user samples. |
| Non-participation | Since the data was collected from publicly available Twitter accounts and academic records, there were no instances of non-participation or dropouts in the traditional sense. All included users had publicly available data and were active on Twitter during the specified period. |
| Randomization | Participants were not allocated into experimental groups. The study is observational, focusing on the naturally occurring behavior of academics and general Twitter users. |

# Reporting for specific materials, systems and methods

We require information from authors about some types of materials, experimental systems and methods used in many studies. Here, indicate whether each material, system or method listed is relevant to your study. If you are not sure if a list item applies to your research, read the appropriate section before selecting a response.

## Materials & experimental systems

| n/a | Involved in the study |
|-----|----------------------|
| ☒ ☐ | Antibodies |
| ☒ ☐ | Eukaryotic cell lines |
| ☒ ☐ | Palaeontology and archaeology |
| ☒ ☐ | Animals and other organisms |
| ☒ ☐ | Clinical data |
| ☒ ☐ | Dual use research of concern |
| ☒ ☐ | Plants |

## Methods

| n/a | Involved in the study |
|-----|----------------------|
| ☒ ☐ | ChIP-seq |
| ☒ ☐ | Flow cytometry |
| ☒ ☐ | MRI-based neuroimaging |

## Plants

| | |
|---|---|
| Seed stocks | *Report on the source of all seed stocks or other plant material used. If applicable, state the seed stock centre and catalogue number. If plant specimens were collected from the field, describe the collection location, date and sampling procedures.* |
| Novel plant genotypes | *Describe the methods by which all novel plant genotypes were produced. This includes those generated by transgenic approaches, gene editing, chemical/radiation-based mutagenesis and hybridization. For transgenic lines, describe the transformation method, the number of independent lines analyzed and the generation upon which experiments were performed. For gene-edited lines, describe the editor used, the endogenous sequence targeted for editing, the targeting guide RNA sequence (if applicable) and how the editor was applied.* |
| Authentication | *Describe any authentication procedures for each seed stock used or novel genotype generated. Describe any experiments used to assess the effect of a mutation and, where applicable, how potential secondary effects (e.g. second site T-DNA insertions, mosiacism, off-target gene editing) were examined.* |

