## [Peer Review File · Nature Human Behaviour]

Political Expression of Academics on Twitter

Corresponding Author: Mx Prashant Garg

Version 0:

Decision Letter:

24th July 2024

Dear Mx Garg,

Thank you once again for your manuscript, entitled "Political Expression of Academics on Social Media", and for your patience during the peer review process.

Your manuscript has now been evaluated by 4 referees. You will see from their comments copied below that, the reviewers are divided in their recommendations on your work. Regardless of specific recommendation, the reviewers raise fundamental concerns regarding validity (and validation), as well as ethics. Taking into consideration all feedback in its entirety, we cannot accept the manuscript for publication, but would be interested in considering a revised version if you are willing and able to extensively revise your manuscript in order to address reviewer and editorial concerns.

We hope you will find the referees' comments useful as you decide how to proceed. If you wish to submit a substantially revised manuscript, please bear in mind that we will be reluctant to approach the referees again in the absence of major revisions. We are committed to providing a fair and constructive peer-review process. Do not hesitate to contact us if there are specific requests from the reviewers that you believe are technically impossible or unlikely to yield a meaningful outcome.

To guide the scope of the revisions, the editors discuss the referee reports in detail within the team, including with the chief editor, with a view to (1) identifying key priorities that should be addressed in revision and (2) overruling referee requests that are deemed beyond the scope of the current study. We hope that you will find the prioritised set of referee points to be useful when revising your study. Please do not hesitate to get in touch if you would like to discuss these issues further. In particular, please address the following (as well as all other reviewer comments):

(1) Reviewer #1 raises a series of important technical concerns about the validity of your analyses: the accuracy of LLM-based measures and the results, for example, is of special importance. Please present further validation of your measures and additional cross-checks (possibly with existing studies) in order to address Reviewer #1's concerns.

(2) As Reviewer #3 notes, currently there is no attention to ethics nor justification about your approach in the manuscript. In revision, please thoroughly discuss the ethical rationale for your study, and the steps you took to minimize harms. Please also transparently describe the nature of the ethics review and approval that took place for your study (as shared with us).

(3) In revision, please add more extensive discussion on limitations, including key points raised by reviewers such as the limitations of the underlying datasets (Reviewers 2 and 4), the choice of Twitter as a data source, and concerns about sample representativeness and potential confounders (Reviewer 1).

(4) In your revision, please implement additional analyses and discussion to address Reviewer 1's concerns about the depth of the analysis.

If you wish to submit a suitably revised manuscript, we would hope to receive it within 4 months. I would be grateful if you could contact us as soon as possible if you foresee difficulties with meeting this target resubmission date.

- Include a “Response to the editors and reviewers” document detailing, point-by-point, how you addressed each editor and referee comment. If no action was taken to address a point, you must provide a compelling argument. When formatting this document, please respond to each reviewer comment individually, including the full text of the reviewer comment verbatim followed by your response to the individual point. This response will be used by the editors to evaluate your revision and sent back to the reviewers along with the revised manuscript.
- Highlight all changes made to your manuscript or provide us with a version that tracks changes.

Link Redacted

Thank you for the opportunity to review your work. Please do not hesitate to contact me if you have any questions or would like to discuss the required revisions further.

Sincerely,

Nature Human Behaviour

Reviewer expertise:

Reviewer #1: Text-mining, Social Media data analysis

Reviewer #2: Communication/Information on social media, Social Media data analysis, science communication & science engagement online

Reviewer #3: Ethical issue in research involving social media data sources

Reviewer #4: Political attitude, Communication/Information on social media, Social Media data analysis

REVIEWER COMMENTS:

Reviewer #1:

Remarks to the Author:

The manuscript investigated political expressions from academics on Twitter by constructing a dataset linking the Twitter profiles of 100,000 scholars to their academic records. Analyzing their posts from 2016 to 2022, the study noted significant variations in academics' stances on climate action, cultural topics, and economic policies, which often diverged from the general population and was dependent on individual characteristics and exhibited temporal changes. Academics are found to be more expressive than average social media users, with differences based on field of study, institutional affiliation, country, and gender. The article also examined the language and tone used by academics, noting that those from top-ranked institutions and US-based scholars tend to use more self-referential language.

One feature of the work is its coverage of a large set of scholars and the selection of some important political topics. However, the analysis is limited to selected political topics only, which are only a tiny fraction (3.5%) of the tweets posted by the academics. Thus, the work might have limited immediate interest to the diverse audience across disciplines. Furthermore, the analysis presented in the work is descriptive and has a number of major issues casting concerns about the validity of the noted observations.

First, the work mentioned in the Introduction section that there is no work on the role of academics' online expressions in the perceptions of science. However, this question is far from being answered and rigorously analyzed in the current work. Instead, it presented a number of descriptive statistics and their variations along different dimensions, without any explorations/in-depth empirical analysis of the underlying causes and broader implications of these differences. Thus, the conceptual novelty of this work is also limited.

Moreover, all the measurements are based on the posted topical tweets. But how likely is it that a user tweets that topic in the first place? Should we incorporate this likelihood into the calculation of the various measures? Under the current quantifications, it cannot distinguish users with different levels of activeness in posting political tweets, as it calculates the fraction among topical tweets.

Second, sample representativeness remains unknown, which may confound the results. Although the study mentioned the sample selection process based on tweeting in the first and last 6 months in the focused period, there are insufficient details provided about the characteristics of this subset and the extent of representativeness of the broader academic population or even of the original 100k sample. This point is important, because scholars from certain fields may be more likely to have Twitter accounts, post more, and feature certain topics, all of which may affect the measurements and the generalizability of the findings.

Third, focusing on the interpretation and implications of the results, the paper lacks deep discussions about the domain significance and interpretation of the results. For example, the Discussion section is more of repeating the result section rather than a contextualized discussion. Another issue is how significant in terms of effect size is the stance difference between, e.g., humanities and STEM and others? Are they meaningful, considering that only 3.5% tweets are relevant (Table 3). Is there any confounder like the field composition of academics that explain the differences? Why is egocentrism on the rise? What are the factors that contribute to such a rise? Again, the study has not examined the underlying causes of the noted differences.

There are numerous methodological concerns that remain to be addressed.

1. The study extensively leveraged LLM for measurements (stance detection, gender inference, etc.). However, the accuracy of LLM results needs to be checked. Although the stance detection performance was tested on the SemEval tasks, they are different from the actual focused tasks (climate action, etc.). Similarly, the accuracy of first name-based gender inference using GPT is unknown, while there are many existing studies on gender in science that can be used for cross-checks.

2. This accuracy issue is particularly important, as many major measurements like Cultural Liberalism are based on several separate tasks (Abortion Rights, Racial Equality, etc., Table 3). Therefore, it needs to reassure that the individual errors are not added up in the composite measures.

3. Small samples (3 English tweets per topic monthly for stance detection and 10 tweets per year for toxicity measurement) raise concerns about potential uncertainty that casts doubts on whether the between-group variations are meaningful.

4. It is argued that the 'how' of what is said might offer a signal for opinions. But the measurement of emotionality-reasoning ratio is independent of the "what" aspect. This may be a confounding factor behind the observed variations of emotionality-reasoning.

5. How accurate is the generated list of keyphrases? While keywords based filtering has a high precision, it suffers from a low recall. Also, it would be informative to provide the list of keyphrases.

6. User geolocation is based on their most recent affiliation recorded in OpenAlex. But wouldn't it make more sense to use the "current" affiliation when the tweet was posted?

Addressing these technical concerns would further ascertain that the observed differences between subgroups is meaningful rather than rising from other factors like sampling uncertainty and classification errors.

The readability of the manuscript needs to be improved substantially, considering the intended general readership of the journal. For instance, the Introduction section may be too long, where the result discussion part could be moved to the Discussion section and the road map may not be necessary. The methods and results sections are dense. The results should be structured in a way that logically builds from the data and methods, allowing the reader to easily trace how the key findings were derived. Many terms like economic collectivism are better explained immediately after being mentioned. The results description parts should clearly indicate which panel in a figure is being referred to. By strengthening the coherence between the data, methods, and findings, the paper would be significantly easier for readers to follow and understand the full scope of the research.

Minor comments:

L147: To claim the distributions follow a power-law, there is a rigorous statistical test procedure to follow [Clauset, A., Shalizi, C. R., & Newman, M. E. (2009). Power-law distributions in empirical data. *SIAM review*, 51(4), 661-703.].

L229: Results about Twitter reach and expertise are not discussed in the main text.

L718: two field categories mentioned but 3 categories are used in the main text.

Table 1: why are the N different under the Tone and Style of Expressions group?

There are numerous minor typo errors and inconsistencies in the text:

Line 36: "there is are ..."

Line 44: "Too date" -> "To date"

Line 139: thing -> think

Line 555-556: "I", "my", "Future drafts will" Also footnote on page 16.

L250: "continuous continuous"

L476: "Open AI"

Reviewer #2:

Remarks to the Author:

This work is interesting and demonstrates how AI can be utilized to evaluate some aspects of language, to help identify other demographic information, and to generate keyword dictionaries.

I found the paper logical and organized appropriately.

In the abstract you say "over 100,000 scholars", but on page 3 you say 99,274 scholars.

On line 245, you say (e.g., lower impact factor)... >> I believe this should be "higher impact factor" --- should impact factor be defined anywhere for audience?

You used OpenAlex for location, affiliation, rankings, but you don't mention the potential missing information that is in the OpenAlex dataset nor do you speak to validating the information you found (e.g. <https://link.springer.com/article/10.1007/s11192-023-04923-y>)

You switched from GPT 4 to GPT 3.5 Turbo for stance detection and narrative extraction. I'm wondering what difference, if any, you might find had you used GPT 4 for this part of the work? You mention that you validated to F1 around 84 to 92, would this have been higher with GPT 4? Did you test a small subset to see if 3.5 and 4 gave similar results?

Overall, I'm impressed with this work.

Reviewer #3:

Remarks to the Author:

It was agreed with the editor that I would only review the reflection of research ethics in the paper, particularly related to the use of online data. That is a very modest task in this case, as there is absolutely no attention for the ethics of scraping Twitter and academics' profiles without informing them, let alone asking for consent. If the authors hold the view that this is legitimate for reasons of public interest, they could at least state and explain this (see e.g., Stommel & De Rijk, 2021).

Stommel, W., & Rijk, L. D. (2021). Ethical approval: None sought. How discourse analysts report ethical issues around publicly available online data. *Research Ethics*, 17(3), 275-297.

Reviewer #4:

Remarks to the Author:

I think a good research report raises more questions than provides answers, and this manuscript clearly does that. The authors seek to characterize expression via social media of academics at large, and to highlight ways in which those who work as academics may differ in their online communication--in form and content--from those who are not. The effort here is at a very broad scale, which directly relates to both its chief advantages and disadvantages. The breadth of the analysis, examining a large number of expressions affiliated with academics tweeting, provides us with some good broad comparisons in terms of who is tweeting, how they are tweeting, their reach, and how this may relate to expertise.

While these distinctions are helpful, I think we lose a great deal of detail in comparing means of the two populations. It is certainly indicative of where more research could head, but that broad view confirms what, I suspect, many would assume is the case: scientists (and other folks who have longer-than-average educational careers) are more likely to hold scientific expertise in high esteem, and therefore more likely to hold the scientific consensus on climate change to be the best one to adopt, for example.

I think there is more value in confirming that many "public intellectuals" in the social media age are not especially influential in their own fields. When asked to name an astronomer or astrobiologist, many in the public would default to Carl Sagan, for example. For a physicist, almost certainly Albert Einstein, though they may also be aware of Richard Feynman thanks to his public efforts, and there can be little doubt these folks were also influential to their field. Particularly in the case of junior scholars, the public communication online may also be a proxy for their informal connections in the field, and may lead to more academic opportunities thanks to "name recognition."

But there are significant challenges to drawing broadly from this analysis. The authors do a good job, I think, of highlighting some of these. They are very clear, for example, that the expressiveness on a topic like climate change may have more to do with comfort or desire to publicly communicate on the topic rather than convictions. And they note that those academics with the greatest public reach may not be representative of the top experts in the field.

The article slowly eases us into the sample, which is drawn from Twitter. While the case can be made that Twitter is of primary importance as a social media platform, and the authors do so to a certain degree by noting its use by journalists in particular, it is clearly not *all* social media use. Moreover, the forms of expression on Twitter are shaped heavily by the

ways in which the algorithm (here to indicate both social and technological factors) surface and reinforce certain types of expression and evidence. This can be as simple as (early) length constraints, but a number of folks (e.g., Pfitzner, Garas, & Schweitzer) have shown that emotional divergence predicts sharing of Tweets. It seems likely that academics may be successful on Twitter specifically to the degree to which they may code-switch and adapt to the discourse community specific to Twitter. I think it would be wrong to assume that the same kinds of dynamics are necessarily present on other platforms.

And, of course, it would likely be wrong to assume that the constraints of Twitter/X have remained the same over time. We have seen a number of evolutionary forces shift the constraints, along with very explicit attempts to change discourse on the site. Some of these changes have resulted in academics abandoning the platform all together. There has been some broad criticism of over-emphasizing Twitter in research because of the availability of the corpus. I don't think we have to go that far--there is a case to be made for its influence--but I do think it's important to bear in mind that platforms matter, and the expressions that are found on a social media platform are necessarily shaped by a negotiation between users and the structures of the platform.

Finally, the analysis makes the necessary assumption (I believe) that there is a one-to-one correspondence between a Twitter user and an academic. While that is true for some--I tweet under a single account--I would guess that a very large number of academics distribute their tweets through academic and non-academic accounts. There is a general messiness here between academics, their public communications, their academic tweets, and tweeting in general. Are these distinctions important? Given the broad thrust of the argument here, perhaps not. We might assume that the differences from the mean--both to the general population and in terms of the difference between twitter impact and traditional impact--are large enough that it is not vital. But what we gain in general brush strokes we lose in precision.

Those limitations notwithstanding, I applaud the methodological precision and clear description of that method provided by the authors--as well as the clarity of exposition throughout. It is unfortunately rare that research reports in this area are thorough enough to be reproduceable and this bucks that standard. It also provides an excellent framework for further exploration of these relationships. It's difficult not to see some of the current methods of sentiment analysis as blunt instruments, but this work raises questions that should be highly generative moving forward.

It's rare that I have so few recommendations for changes. I think at least touching a bit more on the limitations I've noted above would be helpful. At the very least, orienting us more toward the corpus quite early in the discussion would be helpful. This isn't an analysis of social media use--it's an analysis of Twitter use. There is a strong case to be made for why Twitter is a good thing to use here, but this should be clearly spelled out in the introduction, the abstract, and maybe even the title. (As someone with the name of defunct platforms in my own titles, I recognize the danger in this, but clarity should prevail.)

And along the same lines: I recognize the format in which the research report is presented is standard for the journal, though leaving off methods to the end is a bit discombobulating for this reader. At least some of the grit of the method needs to be reported early on--what are the populations/samples, what are the measures, etc., in broad terms earlier on so as to reduce the ambiguity. It's a much better read if bounce around the manuscript and come back up for the results, but I think being a bit more precise about what is being measured would help to understand the conceptual contribution early on.

- Alexander Halavais

Pfitzner, R., Garas, A., & Schweitzer, F. (2012). Emotional Divergence Influences Information Spreading in Twitter. *Proceedings of the International AAAI Conference on Web and Social Media*, 6(1), Article 1. <https://doi.org/10.1609/icwsm.v6i1.14294>

Version 1:

Decision Letter:

7th October 2024

Dear Mx Garg,

RE: "Political Expression of Academics on Social Media"

Thank you for submitting your revised manuscript and for all your work on the revision.

Although your manuscript has been revised in response to reviewer comments, it does not fully comply with our editorial policies and formatting requirements. Specifically, please address the following:

- a) Although we appreciate your extensive response to reviewers document, we are concerned that the current structure (combined with the long length) will be challenging for reviewers to navigate. In order to ensure that reviewers can easily follow the changes you have made in response to their comments, we ask that you revise to respond to each reviewer comment in the order in which they were presented, rather than grouping/summarizing comments by theme. All reviewer comments should be presented verbatim in the point-by-point response document (rather than summarized).
- b) We ask that you follow the core elements of our requirements for statistical reporting at this time. In particular, please

ensure that all statements or interpretations of differences are supported by fully-reported formal statistical analyses. Statistical results should be reported including coefficient/effect size, p-value, and confidence interval. In cases where you refer to a table for full results, the table must include full information. Asterisks to denote statistical significance should not be used as a substitute for exact p-values.

Before we can send the manuscript back to our reviewers, we ask that you revise it to ensure that it complies fully with our policies and is formatted according to our requirements. I have attached another copy of our checklist. If you are uncertain as to how to address any of the points in the checklist, please don't hesitate to contact me.

Link Redacted

Thank you in advance for attending to these requests and I look forward to receiving your revised manuscript.

Sincerely,

Nature Human Behaviour

Version 2:

Decision Letter:

Our ref: NATHUMBEHAV-24051944B

4th December 2024

Dear Dr. Garg,

Thank you for submitting your revised manuscript "Political Expression of Academics on Social Media" (NATHUMBEHAV-24051944B). It has now been seen by two of the original referees and their comments are below. As you can see, the reviewers find that the paper has improved in revision. We will therefore be happy in principle to publish it in Nature Human Behaviour, pending minor revisions to comply with our editorial and formatting guidelines.

We are now performing detailed checks on your paper and will send you a checklist detailing our editorial and formatting requirements within two weeks. Please do not upload the final materials and make any revisions until you receive this additional information from us.

Sincerely,

Nature Human Behaviour

Reviewer #1 (Remarks to the Author):

All my comments have been addressed.

Reviewer #3 (Remarks to the Author):

Dear author(s),

The reflections on ethical issues involved in the use of Twitter data are extensive and sincere, strengthening the justification of the use of these data. Even better is that the manuscript now includes a section on this, highlighting the importance of ethics, even for this type of analysis. I would support publication.

Responses to the editor

First of all, we wanted to express our gratitude for the detailed editorial guidance in navigating the reviewer comments. We discuss each of your main points in turn and explain how we addressed your and the reviewer comments in turn in a brief overview with the more detailed responses being provided for each reviewer.

1. You asked us to put specific emphasis on the comments raised by reviewer 1 on the validity of the analysis. Specifically you suggested that we address questions concerning the accuracy of the LLM-based measures and the empirical interpretation of the results. You further suggested that we carry out additional cross-checks and validations of the LLM-derived measures.

Quote:

“Reviewer #1 raises a series of important technical concerns about the validity of your analyses: the accuracy of LLM-based measures and the results, for example, is of special importance. Please present further validation of your measures and additional cross-checks (possibly with existing studies) in order to address Reviewer #1's concerns.”

Response. Thanks a lot for the detailed guidance. We have reported on a broad range of additional cross checks and cross-referenced our methods with existing studies, as outlined below. These should address the points raised by Reviewer 1. We present these in more elaborate form and detail in our point-by-point response to Reviewer 1. Here, we provide a brief description for convenience.

- *Stance Detection Validation*

We carried out further validation exercises that can speak to the suitability of the stance detection approach that was taken. This involved validation with GPT 3.5, GPT-4. We have added a revised Appendix Table A1 that highlights the suitability of our approach.

- *Topic specificity*

We have highlighted that the techniques used are suitable to detect topics. This yielded a further Appendix Table A2 that highlights a high degree of model agreement across four models: GPT-3.5-turbo, GPT-4, GPT-4o, and GPT-4o-mini. This is strong evidence of the broad robustness of our stance detection approach.

- *Topic Detection Validation*

For topic detection – as opposed to stance detection –, we validated our methodology using the SemEval-2016 dataset. We generated topic-specific dictionaries using GPT-4 and applied them to the SemEval dataset. The analysis suggests high recall and balanced F1 scores across topics, demonstrating the effectiveness of the topic detection method we employed. The results are presented in Appendix Table A3.

- *Gender Classification Validation*

We have carried out an additional verification of the suitability for gender classification based on first names. We utilised a comprehensive dataset of 147,269 unique names from official sources. The GPT-3.5-turbo model was used to predict the gender associated with each name. The results showed very high precision (0.9866), recall (0.9871), and F1 scores (0.9868). The results are presented in Appendix Table A4 of the updated manuscript, reproduced here for ease of reference.

- **Cross-Checks with Existing Studies:**

We added comparing our methods to those in Alturayef et al. (2023) and Marageh et al. (2023), highlighting how dynamic keyword extraction improves accuracy over static models. Our approach addresses limitations of static dictionaries noted in prior studies like Alturayef et al. (2023), emphasizing the importance of context-aware keyword extraction.

We also referenced Chen et al. (2023) to explain how our stance detection uses a similar error-filtering technique to refine outputs and enhance precision. Additionally, we noted that our model achieves average F scores between 79.52% and 92.44%, outperforming traditional models from SemEval-2016 (e.g., RNNs and CNNs with transfer learning, which ranged from 56.28% to 67.82% (Mohammad et al., 2016; Zarella and Marsh, 2016)). These older methods required extensive feature engineering, whereas our approach dynamically adapts without manual tuning, while using a versatile method, illustrating the advantages of LLMs in handling nuanced political discourse on platforms like Twitter.

We believe that these additional validations across stance detection, topic detection, and gender inference illustrate the overall broad reliability and accuracy of the LLM-based models that we employed in our study.

We also contextualize our findings within existing research. Williams and Ceci (2023) highlight ideological bias on social media, where progressive voices dominate, consistent with Langbert's (2018) analysis of political affiliations in academia, suggesting amplification of these biases on platforms like Twitter. Crockett (2017) explains how digital platforms lower the barriers for expressing outrage, intensifying moral and polarized discourse—supporting our observation of emotionally charged content on topics like climate action gaining more visibility. Similarly, Ke, Ahn, and Sugimoto (2017) find that a small, highly active subset of social and computer scientists disproportionately shapes conversations on Twitter, paralleling our findings. Ferrara (2015) echoes concerns about algorithmic amplification, where non-expert academics with large followings may distort the perception of scientific consensus.

- **Reproducing Key Results Using Different Techniques**

To further improve the validity of our findings, we conducted additional robustness checks. Controlling for tone, we found the patterns remained consistent across the cross-section. We also emphasized the importance of our previously reported appendix, which controlled for individual-level fixed effects (unobserved heterogeneity), showing no change in results. These updates have been highlighted in the manuscript to address concerns raised by Reviewer #1 and Reviewer #4.

References:

- Alturayeif, N., Luqman, H. and Ahmed, M., 2023. A systematic review of machine learning techniques for stance detection and its applications. *Neural Computing and Applications*, 35(7), pp.5113-5144.
- Chen, H., Wang, X., Chen, H., Song, Z., Jia, J. and Zhu, W., 2023. Grounding-Prompter: Prompting LLM with Multimodal Information for Temporal Sentence Grounding in Long Videos. *arXiv preprint arXiv:2312.17117*.
- Crockett, M.J., 2017. Moral outrage in the digital age. *Nature human behaviour*, 1(11), pp.769-771.
- Ferrara, E., 2015. " Manipulation and abuse on social media" by Emilio Ferrara with Ching-man Au Yeung as coordinator. *ACM SIGWEB Newsletter*, 2015(Spring), pp.1-9.
- Ke, Q., Ahn, Y.Y. and Sugimoto, C.R., 2017. A systematic identification and analysis of scientists on Twitter. *PLoS one*, 12(4), p.e0175368.
- Langbert, M., 2018. Homogenous: The political affiliations of elite liberal arts college faculty. *Academic questions*, 31(2).
- Maragheh, R.Y., Fang, C., Irugu, C.C., Parikh, P., Cho, J., Xu, J., Sukumar, S., Patel, M., Korpeoglu, E., Kumar, S. and Achan, K., 2023, December. LLM-take: theme-aware keyword extraction using large language models. In *2023 IEEE International Conference on Big Data (BigData)* (pp. 4318-4324). IEEE.
- Mohammad, S., Kiritchenko, S., Sobhani, P., Zhu, X. and Cherry, C., 2016, June. Semeval-2016 task 6: Detecting stance in tweets. In *Proceedings of the 10th international workshop on semantic evaluation (SemEval-2016)* (pp. 31-41).
- Williams, W.M. and Ceci, S.J., 2023. How politically motivated social media and lack of political diversity corrupt science. In *Ideological and Political Bias in Psychology: Nature, Scope, and Solutions* (pp. 357-375). Cham: Springer International Publishing.
- Zarrella, G. and Marsh, A., 2016. Mitre at semeval-2016 task 6: Transfer learning for stance detection. *arXiv preprint arXiv:1606.03784*.

2. You highlighted the absence of a discussion around the ethics of our study in the first submission. You suggested that our revision should thoroughly discuss the ethical rationale for this study and suggest the ways that were undertaken to minimise harms. You also asked to transparently describe the nature of the ethics review and the approval that was sought.

Quote:

“As Reviewer #3 notes, currently there is no attention to ethics nor justification about your approach in the manuscript. In revision, please thoroughly discuss the ethical rationale for your study, and the steps you took to minimise harms. Please also transparently describe the nature of the ethics review and approval that took place for your study (as shared with us).”

Response:

We appreciate your feedback on the importance of thoroughly discussing the ethical rationale for our study and the steps taken to minimise potential harms. In our response to reviewer 3 we have expanded in much more detail, but below we provide a summary for your convenience. We provide a succinct summary of the arguments in the below 5 bullet points.

1. **Public Nature of Social Media Content:** Our study is based on social media content that is publicly accessible and produced with the intention of being shared in a public domain. Social media platforms are designed for users to engage in public discourse, and thus, it is reasonable to expect that such content could be used for research purposes. The data we analysed is already in the public domain, and our study does not involve private or sensitive information.
2. **Importance of Research on Social Media Discourse:** Social media has become a significant source of news and information for a large segment of the population. Understanding the structure of discourse among individuals who actively choose to participate in these platforms is vital for comprehending how information is disseminated, consumed, and potentially manipulated. Our research contributes to this understanding by analysing patterns and trends in social media interactions, which is critical in an era where these platforms increasingly shape public opinion.
3. **Lack of Established Ethical Practices Among Social Media Content Producers:** Currently, there is limited understanding of how social media content producers themselves engage in ethical practices or adhere to codes of conduct. This gap in knowledge further justifies our study, as it highlights the need to explore how social media content producers operate in the realm of social media. Our research, therefore, provides valuable insights into this largely unregulated area that has been subject to significant interest given the inherent tension between freedom of expression and individual liberties.
4. **Impact of Algorithms and Anonymization:** We acknowledge that algorithms can influence the visibility and spread of social media content, potentially impacting the discourse in ways that are not immediately apparent. Our study takes this into account by focusing on descriptive patterns that are anonymized to protect individual identities. Furthermore, we are ignoring the actual *reach dimension* of the content, but rather, want to gain maximal understanding of the breadth of topic engagement and expression across individuals whereby each individual is given the same weight. The data analysis is conducted in a way that ensures no harm to the individuals involved, and no identifiable personal information is used in our research outputs.
5. **Ethics Review and Approval:** Our study was subjected to a rigorous ethics review process and received approval from the Institutional Review Board (IRB) at Imperial College. We have attached the full IRB review to this response for your reference. The ethical considerations discussed with the IRB, including the public nature of the data, anonymization techniques, and the overall importance of the research, align with the arguments presented here. This ensures that our study adheres to the highest ethical standards.

Additionally, we have now included an "Ethics and Inclusion Statement" in the revised manuscript, summarising our ethical considerations, anonymization protocols, data handling procedures, and our commitment to transparency.

We hope this addresses your concerns and provides a clear rationale for the ethical approach taken in our study. Thank you for your valuable feedback, which has helped us enhance the transparency and rigour of our ethical considerations.

3. You further suggested that we add an extensive discussion on the limitations of the underlying study, in particular with reference to the points raised by reviewers 2 and 4 in particular around the choice of Twitter as a data source and the concerns around the sample representativeness and the role that confounders could play as highlighted by reviewer 1.

Quote:

“In revision, please add more extensive discussion on limitations, including key points raised by reviewers such as the limitations of the underlying datasets (Reviewers 2 and 4), the choice of Twitter as a data source, and concerns about sample representativeness and potential confounders (Reviewer 1).”

Response:

We appreciate the constructive feedback regarding the need for a more extensive discussion of the study’s limitations, particularly in relation to the concerns raised by Reviewers 1, 2, and 4. We have provided detailed responses for each of the reviewers' comments in their respective own response letters. We provide here a summary on how we incorporated these points in the revised manuscript.

- **Choice of Twitter as a Data Source:**

We acknowledge that Twitter is not representative of the entire population and that its user base may not reflect the diversity of the broader public. However, our choice of Twitter as a data source is deliberate and grounded in its role as a platform where key influencers—such as journalists, politicians, and academics—actively engage. These groups are often seen as thought leaders or intellectual elites, and their interactions on Twitter can have significant downstream effects on public discourse, media agendas, and potentially even policy-making. These are now clearly mentioned in abstract, introduction and discussion, as suggested by the reviewers.

- **Role of Twitter in Shaping Broader Media Landscape:**

One premise that positions the relevance of our paper is the potential for social media platforms like Twitter to influence traditional media and, by extension, the broader public sphere. Ample (anecdotal) evidence suggests that such a pipeline may exist. Traditional media outlets increasingly compete for information and attention, and what trends on social media can significantly shape the narratives that appear in mainstream media. This creates a feedback loop where content that gains traction on Twitter may influence broader news agendas and, ultimately, public policy discussions. The limitations of Twitter as a data source thus align with our research aim: to highlight the disproportionate influence that certain groups and trends on social media can exert on wider society.

- **Sample Representativeness and Potential Confounders:**

We recognize the concerns about sample representativeness and the potential influence of confounders. Given Twitter’s user demographics and the nature of its platform, the sample may not capture the full spectrum of public opinion. However, this limitation is also a key focus of our study. The very fact that Twitter represents a distinct audience—one that includes many voices that may be perceived as influential —makes it a valuable subject of study, despite its lack of representativeness. We also acknowledge the potential role of confounders, such as the influence of algorithms or external events. Regarding the algorithms, in our response, we highlighted that this is why we think it is particularly prudent to work with a balanced panel of academics that engage in online discourse as this allows us to make longitudinal comparisons. Further, the sampling strategy implies that we are not giving more weight to (very) frequent participants in online discourse. These factors, while complicating the analysis, do not diminish the value of understanding the specific dynamics at play within this particular segment of the public discourse.

Thank you again for your insightful comments. We believe that acknowledging and addressing these limitations strengthens our manuscript, improves its transparency and provides a more nuanced understanding of the complex dynamics at play in social media-driven discourse.

4. You also asked us to implement the additional analysis and discussions that were raised by reviewer 1.

Quote:

“In your revision, please implement additional analyses and discussion to address Reviewer 1’s concerns about the depth of the analysis.”

Response:

Thanks a lot. On the substantive additional analysis that were highlighted by the reviewer. We provide a summary of these analyses here along with the key observations.

- **Aggregation bias concerns**

The reviewer asked us to provide a view of the components that go into the indices that make up Economic collectivism and Cultural liberalism. The specific concern relates to the possibility that the aggregation into an index may conflate individual trends into a broader topic-related trend. We opted for the aggregation into an index for ease of presentation to reduce the dimensionality of the empirical observations into specific themes. Yet, we agree that this may raise the question about whether the underlying latent trends are an artefact of the aggregation. To alleviate these concerns, we conducted a detailed analysis to examine the trends within each composite measure and its respective components. This has been added in the discussion and in the appendix. Throughout we observe that the individual time series reveal very similar time trends. The only exception is the issue of attitudes towards Abortion Rights that saw a dynamic change in 2022. We have commented on it in a footnote where we refer the reader to the respective appendix.

- **Static versus dynamic affiliation**

The reviewer asked us to consider the sensitivity of the findings to using a time-varying coding of the country of affiliation of an academic and/or their institutional affiliation. In response, we have clarified our methodology to incorporate *time-varying institutional affiliations* using data from OpenAlex and yearly university rankings from QS World University Rankings. Specifically, we now extract the institution name from the author metadata associated with each publication by academics between 2016 and 2022, which allows us to capture changes in an academic's affiliation and university ranking over time. This is not perfect but this allows us to capture changes in an academic's affiliation over time, providing a more precise mapping of their geolocation and institutional context at the time each tweet was posted.

This impacts two analyses. For the analysis of country of affiliation—US versus non-US—we note that 87% of academics are consistently based at US or non-US institutions. However, when considering university rankings with our new 4-category system (1-100, 101-500, 501-1500, 1500+), we find that 50.3% of academics remain in the same category throughout the sample period. This higher mobility within university rankings is driven by both academics moving between institutions of different rankings and annual changes in the rankings themselves, which increases noise, particularly in the lower-ranked categories. Despite this, the analysis remains valuable as it captures the dynamic nature of academic institutions and the associated movement of academics, providing insights into patterns of academic expression over time.

Our main findings that academics from the US and those from top 100 institutions differ significantly from others still hold true. The refined analysis, however, provides additional insights into the variability and non-linearity within the lower-ranked categories. These adjustments enhance the overall robustness and validity of our results.

Impact on main patterns and findings

To assess the impact of this adjustment, we repeated key analyses—specifically Figures 2-4 (and their appendix equivalents 4-6)—comparing results derived from static affiliation data (as of January 2023) with those using time-varying location data.. The overall results are very similar.

- **Comparability of balanced and unbalanced sample**

Reviewer 1 asked us to comment on the comparability of the balanced sample of academic users of Twitter with the unbalanced sample.

The reason to focus on a sample that we observe consistently over time is because, in addition to cross-sectional comparisons of academics across topics along salient dimensions such as gender or the country of academic affiliation, we want to make longitudinal comparisons. This is where sample selection that is done explicitly in a way to provide us with a longitudinal sample is relevant as this way we can ensure that the trends that are documented are not conflating compositional changes of the sample. We do expect that the balancing, on average, may actually attenuate the

observed differences as we focus on a population of users that appears to be regular social media users.

To ensure the reader can understand better the sample differences we have added a new comparison table in the appendix between the balanced and unbalanced sample in the Appendix as Table A9.

Our balanced sample consists of approximately 99,274 academics who tweeted at least once in the first six months of 2016 and at least once in the last six months of 2022. This ensures a consistent presence across the entire study period. We compared this balanced sample with the unbalanced sample, which includes all individuals who tweeted at least once at any point during the period from 2016 to 2022.

The comparison reveals that the balanced and unbalanced samples are very similar in terms of non-Twitter metrics such as publication metrics (e.g., number of citations, number of works published), gender distribution, and discipline distribution. This similarity suggests that our balanced sample is representative of the broader academic population in these dimensions, at least on average.

- **Carry out formal statistical test on fit of power law**

Thank you for highlighting the need to rigorously substantiate the claim that the distributions of academic influence metrics follow a power-law. In response to your comment, we have now conducted a thorough statistical analysis following the methodology outlined by Clauset, Shalizi, and Newman (2009), as suggested.

This analysis involved fitting a power-law distribution to the data on content creation, engagement (likes received), public reach (followers), and citations among academics. The results, now included in the main body of our manuscript, confirm that these metrics do indeed follow a power-law distribution. Specifically, we calculated the power-law exponents (α) for each metric, with values ranging from 2.636 to 3.337, as detailed in Appendix Table A6. Additionally, the low Kolmogorov-Smirnov (KS) statistics obtained from our analysis further validate the goodness of fit for the power-law model across these metrics.

We have incorporated this additional validation into the manuscript, as described in the revised subsection on the "Distribution of Academic Influence on Twitter." This section now includes a brief overview of the statistical analysis and its results, with full details presented in Appendix Table A6. This ensures that our claims regarding the power-law nature of these distributions are well-supported by both visual and statistical evidence.

Response to reviewer 1: Text-mining, social media data analysis expert

Thanks a lot for your detailed comments and suggestions. In the following responses, we attempted to identify the main points of suggestions you raised in your report and provide an overview of how we attempted to adjust the manuscript to reflect your comments.

You first raised a set of technical issues before offering a broad range of suggestions on how to improve the exposition. We first discuss the technical suggestions and then turn to how we incorporated your many helpful expositional suggestions.

1. You suggested that we carry out more extensive checks of the accuracy of the LLM methods used for stance detection. You highlight the validation of stance deception for SemEval but pointed out that performance on the more focused tasks may be worse. You also pointed to the need to validate the name-based gender inference using GPT.

Quote:

“The study extensively leveraged LLM for measurements (stance detection, gender inference, etc.). However, the accuracy of LLM results needs to be checked. Although the stance detection performance was tested on the SemEval tasks, they are different from the actual focused tasks (climate action, etc.). Similarly, the accuracy of first name-based gender inference using GPT is unknown, while there are many existing studies on gender in science that can be used for cross-checks.”

Response

We appreciate your suggestion to thoroughly validate our use of GPT-based models for stance detection, topic detection, and gender inference. To ensure the accuracy and reliability of these measurements, we conducted a series of validations across different datasets and tasks.

Stance Detection Validation

As you point out, our initial validation of the GPT-3.5-turbo model for stance detection used the ACL SemEval-2016 Task 6 dataset, which includes tweets labelled for stance (pro, anti, neutral) across a diverse set of topics. The results showed strong performance, achieving average F1 scores ranging from 84 to 92, closely aligning with human annotations.

To further test the generalizability of our approach, we compared the performance of GPT-3.5-turbo with GPT-4 on the same SemEval dataset. The results were consistent across both models, reinforcing our confidence in the external validity of GPT-3.5-turbo and its reliability for stance detection. The results are presented in Appendix Table A1 of the updated manuscript, reproduced here for ease of reference.

Table A1: Evaluation Metrics for Stance Detection

Task	Target	GPT 3.5 Turbo (F_{avg})	GPT 4 (F_{avg})
A	Feminism	92.44	81.89
A	Hillary Clinton	89.57	87.53
A	Abortion	79.52	84.36
B	Donald Trump	84.18	80.00

Notes: Table shows results of validation of stance detection step. We obtain human labels for stance detection task from ACM SemEval-2016 Task 6 (Mohammad et al. 2016). Humans labelled tweets are pro-, anti- and neutral-, on topics ranging from Abortion Rights to Donald Trump. The stance detection’s effectiveness was validated against 40,317 hand-coded labels from 137 humans, yielding F-scores of 79.52 to 92.44 for GPT-3.5 Turbo and 80.00 to 87.53 for GPT-4, which are considered very high for classification tasks.

Topic specificity

You also raised the concern about the suitability of the classifiers for different types of topics. To address this, we also conducted an additional analysis using a new subset of tweets to expand this validation to the specific topics in our study. This analysis involved labelling 400 random tweets per topic per stance from our main analysis sample, totaling 8,400 tweets. The results showed strong consistency across models, particularly as model quality increased, confirming the robustness of our stance detection approach. Specifically, these were labelled over 10 iterations by four models: GPT-3.5-turbo, GPT-4, GPT-4o, and GPT-4o-mini, leading to 336,000 predictions.

After filtering out tweets labelled as unrelated by any model, we retained 63,210 predictions per model (75.25% of the original set). We measured the agreement rate and F1 scores across these models, finding strong consistency, particularly as model quality increased. For instance, comparisons involving GPT-4o, the most advanced model, yielded the highest agreement rates and F1 scores, confirming the robustness of our stance detection approach. The results are presented in Appendix Table A2 of the updated manuscript, reproduced here for ease of reference.

Table A2: Comparison of Agreement and F1 Scores Across GPT Models

Comparison	Agreement (Modal)	Agreement (Iterations)	F1 (Modal)	F1 (Iterations)
GPT-3.5-turbo vs GPT-4o	0.781	0.772	0.806	0.795
GPT-3.5-turbo vs GPT-4	0.750	0.738	0.772	0.756
GPT-3.5-turbo vs GPT-4o-mini	0.684	0.681	0.696	0.691

Notes: Table presents a comparison of agreement rates and average F1 scores (F_{avg}) between different GPT models for stance detection. **Agreement (Modal)** refers to the proportion of identical stance predictions when considering the modal stance across 10 iterations per tweet. **Agreement (Iterations)** measures this proportion for each individual iteration treated as a unique instance. **F1 (Modal)** represents the average F1 score when one model’s modal stance is used as the “true” label to evaluate another model’s performance. Similarly, **F1 (Iterations)** does the same but treats each iteration as a separate instance. Higher values in these metrics indicate stronger consistency and accuracy in stance classification.

Topic Detection Validation

For topic detection, we validated our methodology using the SemEval-2016 dataset. We generated topic-specific dictionaries using GPT-4 and applied them to the SemEval dataset. The results showed high recall and balanced F1 scores across topics, demonstrating the effectiveness of our topic detection method. The results are

presented in Appendix Table A3 of the updated manuscript, reproduced here for ease of reference. We will discuss these in more detail in to point 5 below.

Table A3: Evaluation Metrics for Topic Detection Validation

Topic	Precision	Recall	F1-Score
Legalization of Abortion	86.51	66.00	74.87
Hillary Clinton	75.49	60.79	67.35
Feminist Movement	51.00	97.67	67.01
Donald Trump	79.41	65.59	71.84
Overall	65.47	74.32	69.62

Notes: Table shows precision, recall, and F1-Score metrics for topic detection. The results highlight the effectiveness of the topic dictionaries generated using GPT-4 and the robustness of the methodology. Tweets were matched against the dictionaries, and true and false positives and negatives were calculated. For instance, a true positive (TP) occurs when a tweet labeled as "Legalization of Abortion" contains a term from the corresponding dictionary, while a true negative (TN) occurs when no terms from the dictionary match tweets labeled for other topics. The high recall demonstrates the comprehensive nature of the dictionaries, which capture the evolving lexicon over the years, while the precision shows the effectiveness of filtering out unrelated tweets. This approach ensures accurate topic detection across diverse tweet datasets by leveraging context-aware keyword dictionaries and refining selections to enhance accuracy.

Gender Classification Validation

You raised the concern about the accuracy of the gender classification. To validate gender inference based on first names, we utilised a comprehensive dataset of 147,269 unique names from official sources. The GPT-3.5-turbo model was used to predict the gender associated with each name. The results showed high precision (0.9866), recall (0.9871), and F1 scores (0.9868) when weighted by name frequency, demonstrating the robustness of our gender classification method. The results are presented in Appendix Table A4 of the updated manuscript, reproduced here for ease of reference.

Table A4: Evaluation Metrics for Gender Classification

Metric	Unweighted	Weighted by Count
Precision	0.8097	0.9866
Recall	0.8202	0.9871
F1 Score	0.8149	0.9868
Accuracy	0.8610	0.9863

Notes: Validation results for the gender classification method using a dataset of 147,269 unique names from authoritative sources, including US Social Security Card Applications (1880-2019), UK Baby Names (2011-2018), British Columbia's Baby Names (1918-2018), and Australia's Popular Baby Names (1944-2019). The GPT-3.5-turbo model classified names as Male, Female, or Unclear. The table presents both unweighted and weighted metrics: Precision, Recall, F1 Score, and Accuracy. Unweighted metrics show high overall performance, treating all names equally. Weighted metrics, which give more importance to frequently occurring names, yield even higher scores, particularly for names more likely expected to appear in our dataset. This demonstrates the method's strong accuracy and robustness, especially for common names.

We believe that these additional validations across stance detection, topic detection, and gender inference underscore the reliability and accuracy of the GPT-based models employed in our study. By ensuring that these models perform well across different datasets and tasks, we provide a strong foundation for the study's findings.

2. You pointed out the accuracy issue that is particularly relevant or salient. You specifically commented on the clustering of themes or topics under the combined label of cultural liberalism such as the abortion rights, welfare state etc. topics. You specifically asked whether there are concerns that the trends detected may be an artefact of noisiness in measurement upon aggregation.

Quote:

“This accuracy issue is particularly important, as many major measurements like Cultural Liberalism are based on several separate tasks (Abortion Rights, Racial Equality, etc., Table 3). Therefore, it needs to reassure that the individual errors are not added up in the composite measures.”

Response You raised a point regarding the importance of accuracy in our composite measures, particularly for Cultural Liberalism and Economic Collectivism. We understand your concern that individual errors in the component tasks could accumulate and affect the overall composite measures. We opted for the aggregation into an index for ease of presentation to reduce the dimensionality of the empirical observations into specific themes. Yet, we agree that this may raise the question about whether the underlying latent trends are an artefact of the aggregation. Of course, we have checked these subcomponents before, but for increased transparency, we have added these as new supplementary findings.

To alleviate these concerns, we conducted a detailed analysis to examine the trends within each composite measure and its respective components. We present these in the figure plotted below and comment on the key observations. :

Time series plots reveal similar trends breaking down the composite measures into their constituent parts; we found that most subcomponents exhibit broadly similar trends, suggesting that individual component variations do not significantly skew the overall measure.

The exception is the change in attitudes towards abortion Rights'. Yet, the notable increase in tweets related to Abortion Rights in 2022 did not substantially alter the overall trend for Cultural Liberalism, which continued to decline. This is because the number of tweets related to Abortion Rights is relatively small compared to those related to Immigration or Racial Equality issues. This point was made in Table 3 (Now Table 2), the number of tweets related to Abortion Rights is roughly one-tenth of those related to Immigration or Racial Equality. Therefore, the influence of this spike on the composite measure is proportionally small.

These findings provide reassurance that the composite measures are robust and not unduly influenced by idiosyncratic variation in the subcomponents. We have included these time series plots in the manuscript appendix and provide them below for convenience, to further validate our approach and ensure transparency in how these composite measures are constructed. We have added a footnote to this effect in the main paper with a link to the supplementary appendix where we explicitly reference

these observations.

3. You pointed to the relatively small samples that are used for the study and that this may drive uncertainty.

Quote:

“Small samples (3 English tweets per topic monthly for stance detection and 10 tweets per year for toxicity measurement) raise concerns about potential uncertainty that casts doubts on whether the between-group variations are meaningful.”

Response Thank you for raising the concern regarding the potential measurement error that may be introduced by our sampling approach for stance detection and

toxicity measurement. We recognize the importance of ensuring that between-group variations in our analysis are meaningful and not artefacts of small sample sizes. Our sampling strategy was specifically designed to ensure maximal between-group comparability by preventing the results from being skewed by highly prolific accounts. Specifically, we focus on creating a balanced panel to ensure our results are not overly influenced by a few very vocal and influential accounts that post frequently.

We now add a further comment on the sampling strategy in the revised draft. There we now emphasise that the sampling approach—capping at three tweets per author per topic per month—is a deliberate balancing criterion aimed at creating a balanced panel. By focusing on academic-time as the unit of analysis rather than the individual tweet, we prevent our dataset from being skewed by a small number of academics who might tweet excessively. Additionally, our stance detection metric is normalised across all tweets for a given topic, which means that no single tweet or user has an outsized influence on the stance measurements. This ensures that our comparisons between groups remain robust and meaningful.

Specifically, 99.38% of cases involve just one tweet per author per topic per month, with only 0.62% involving two tweets and an even smaller fraction (0.006%) involving three tweets. This means that increasing the cap would affect less than 0.006% of the observations, making any potential impact on our results negligible. We further add more clarification in a new footnote in the Methods section that provides a reference of the minimal impact of this sampling cap, reproduced here for reference:

“Although we cap the sample at three tweets per author per topic per month for stance detection, 99.38% of cases involve just one tweet, with only 0.62% involving two and 0.006% involving three. Thus, increasing the cap would affect less than 0.006% of the observations, making any impact on results negligible.”

We hope this clarification addresses your concerns, and we appreciate your feedback, which has led us to further elucidate this aspect of our methodology in the manuscript.

4. You suggested that our interpretation of tone – the “how” things are said – carries a signal of what is being said.

Quote:

“It is argued that the ‘how’ of what is said might offer a signal for opinions. But the measurement of emotionality-reasoning ratio is independent of the “what” aspect. This may be a confounding factor behind the observed variations of emotionality-reasoning.”

Response Thank you for this insightful comment. We appreciate your suggestion regarding the potential role of “how” things are said—such as tone, emotionality, egocentrism, or toxicity—possibly influencing the signal conveyed by the “what” aspect of the message. This concern highlights the possible confounding effect of the “how” on the observed differences in opinions across subgroups, and it is worth exploring further. To address this concern, we performed two key analyses:

Raw Correlations Between Stances and “How” Expressions

First, we examined the raw correlations between each stance (the “what”) and three key measures of expression style (the “how”)—emotionality, egocentrism, and toxicity. The results are presented in the table below:

Table A6: Raw Correlations Between Stances and Emotionality, Egocentrism, and Toxicity

Stance	Emotionality	Egocentrism	Toxicity
Climate Action	-0.008	-0.027	-0.033
Techno-Optimism	-0.017	-0.041	-0.055
Behavioural Adjustment	-0.008	0.000	-0.014
Cultural Liberalism	0.021	0.057	0.085
Economic Collectivism	0.004	0.015	0.000

Note: This table presents the raw correlations between different political stances and three key variables related to how academics express themselves: **Emotionality**, **Egocentrism**, and **Toxicity**. These variables represent the tone and style of expression, with emotionality capturing the ratio of affective to cognitive words, egocentrism measuring the prevalence of self-referential language, and toxicity capturing the likelihood of harmful or aggressive language. The stances—Climate Action, Techno-Optimism, Behavioural Adjustment, Cultural Liberalism, and Economic Collectivism—are defined as the net stance per author on each topic. The raw correlations are all near zero, suggesting minimal direct relationship between stance and expression style, with the highest correlation being 0.085 for Cultural Liberalism and Toxicity. This indicates that “how” things are said does not heavily influence the political stance expressed by academics.

These correlations show that the relationship between stance and expression style is minimal in most cases. For example, the correlation between emotionality and stance is very close to zero, with the largest observed correlation being 0.085 between cultural liberalism and toxicity. This suggests that, in general, expression style does not have a strong raw correlation with stance.

Regression Analysis to Control for “How” Expressions While Examining Subgroup Differences

In the second step, we explored whether the observed differences across subgroups could be confounded by expression style. To do this, we ran a series of regressions for each stance, comparing subgroup differences (e.g., gender, field, rankings, and country) while controlling for one of the three “how” expressions at a time (emotionality, egocentrism, or toxicity). Specifically, we estimate the following model:

$$\text{Stance}_{it} = \alpha + \beta_1 * \text{GroupVar}_{it} + \beta_2 * \text{HowExpression}_{it} + \varepsilon_{it}$$

Where GroupVar_{it} represents a subgroup indicator (e.g., male, STEM, US), and $\text{HowExpression}_{it}$ represents one of the three expression styles (emotionality, egocentrism, or toxicity). This allows us to assess whether controlling for the “how” variables alters the subgroup coefficient β_1 . If β_1 remains stable across the three models, it indicates that tone and style of expression do not confound subgroup differences.

Climate-related stances: We begin by presenting the subgroup differences for climate-related stances (Climate Action, Techno-Optimism, and Behavioural Adjustment), as originally shown in the paper (Figure 2). We then display the updated results after controlling for emotionality, egocentrism, and toxicity.

Figure: Subgroup Comparisons for Climate-Related Stances (Figure 2 Reproduced)

Figure: Subgroup Comparisons for Climate-Related Stances, Controlling for Emotionality, Egocentrism, and Toxicity (Appendix Figure A15 reproduced)

As seen in the new figure, the coefficients for subgroup differences remain consistent across the three regressions, each controlling for one of the expression styles. This suggests that the "how" expressions—whether emotionality, egocentrism, or toxicity—do not confound the subgroup differences for these climate-related stances.

Socio-economic stances: Next, we replicate this analysis for the stances on Cultural Liberalism and Economic Collectivism. Again, we start by presenting the subgroup differences from the original analysis (Figure 3) and then show the updated results with controls for emotionality, egocentrism, and toxicity.

Figure: Subgroup Comparisons for Economic and Cultural Stances (Figure 3 Reproduced)

Figure: Subgroup Comparisons for Economic and Cultural Stances, Controlling for Emotionality, Egocentrism, and Toxicity (Appendix Figure A16 reproduced)

Again, we observe that the coefficients for subgroup indicators remain stable across all three regressions, indicating that controlling for the “how” expressions does not significantly alter the subgroup effects. Thus, it is unlikely that emotionality, egocentrism, or toxicity confound the observed differences across subgroups for these stances.

Based on our results, we find that expression style (the “how”)—whether emotionality, egocentrism, or toxicity—does not confound the observed subgroup differences in stance (the “what”). The coefficients for subgroup differences remain consistent regardless of whether we control for these expression styles, indicating that the variations we observe are primarily driven by the content of the statements rather than how they are expressed.

We have included the raw correlation table in Appendix Table A7, and the two additional figures in Appendix Figures A15 and A16. These results are discussed in detail in Appendix Subsection "Robustness of Subgroup Differences in Political Stances After Controlling for Tone and Style Variables." We hope this additional analysis clarifies and strengthens the interpretation of our findings.

5. You asked about the accuracy of the generated list of keyphrases that are used for topic detection. You suggested that, while keyword based filtering may have high precision, it suffers from low recall. You also asked for the list of keyphrases.

Quote:

"How accurate is the generated list of keyphrases? While keywords based filtering has a high precision, it suffers from a low recall. Also, it would be informative to provide the list of keyphrases."

Response Thanks a lot for your feedback and suggestions here. We have taken this suggestion on board to provide some more evidence on the suitability of the key phrase approach for topic detection, which we have documented in the newly added appendix section "Validation of Topic Detection." Specifically, we utilised the SemEval-2016 Task 6 dataset as a benchmark, which provides a set of tweets labelled for specific topics, including Feminist Movement, Hillary Clinton, Legalisation of Abortion, and Donald Trump. This dataset serves as a reliable ground truth for evaluating the effectiveness of our GPT-4-generated topic dictionaries.

Our approach involved generating topic-specific dictionaries through GPT-4, leveraging a diverse set of prompts that accounted for various ngrams, temporal contexts (2016-2022), and both general and Twitter-specific vernaculars. This comprehensive method ensured that our dictionaries captured the evolving and dynamic lexicon pertinent to each topic, which is particularly crucial on platforms like Twitter where language usage rapidly changes.

We then applied these dictionaries to the SemEval dataset to evaluate their performance. For each topic, we calculated standard evaluation metrics, including true positives (TP), true negatives (TN), false positives (FP), and false negatives (FN). The results, as presented in Table A3: "Evaluation Metrics for Topic Detection Validation," **provided here for ease of reference**, indicate that our methodology achieves high recall across all topics, demonstrating the effectiveness of our comprehensive dictionary generation process in capturing a wide range of relevant tweets.

Table A3: Evaluation Metrics for Topic Detection Validation

Topic	Precision	Recall	F1-Score
Legalization of Abortion	86.51	66.00	74.87
Hillary Clinton	75.49	60.79	67.35
Feminist Movement	51.00	97.67	67.01
Donald Trump	79.41	65.59	71.84
Overall	65.47	74.32	69.62

Notes: Table shows precision, recall, and F1-Score metrics for topic detection. The results highlight the effectiveness of the topic dictionaries generated using GPT-4 and the robustness of the methodology. Tweets were matched against the dictionaries, and true and false positives and negatives were calculated. For instance, a true positive (TP) occurs when a tweet labeled as "Legalization of Abortion" contains a term from the corresponding dictionary, while a true negative (TN) occurs when no terms from the dictionary match tweets labeled for other topics. The high recall demonstrates the comprehensive nature of the dictionaries, which capture the evolving lexicon over the years, while the precision shows the effectiveness of filtering out unrelated tweets. This approach ensures accurate topic detection across diverse tweet datasets by leveraging context-aware keyword dictionaries and refining selections to enhance accuracy.

For example, the "Legalisation of Abortion" topic achieved a precision of 86.51% and a recall of 66.00%, resulting in an F1-Score of 74.87. Similarly, the "Donald Trump" topic achieved a precision of 79.41% and a recall of 65.59%, with an F1-Score of 71.84. These results underscore the robustness of our approach, particularly in ensuring high recall, which is critical for capturing the broad spectrum of relevant discourse. The union of terms across different years and ngram combinations allowed us to maintain a high recall while ensuring the precision remained sufficiently strong to filter out unrelated content effectively.

While the specific topics in the SemEval dataset differ from those used in our main study (with the exception of "Legalisation of Abortion"), the consistent performance across multiple topics enhances our confidence in the external validity of our methodology when applied to other topics in our research. This validation also supports the scalability and cost-efficiency of our approach, which allows for the accurate detection of topics in large tweet datasets without the need to parse each individual tweet through the language model.

Additionally, we have added an appendix table showing 20 randomly sampled ngrams per topic (Appendix Table A8, **reproduced here for reference**), and we also provide the full list of terms for each topic in the replication package as a topic-by-term list . This ensures transparency and allows for further scrutiny of the generated dictionaries.

Table: Random Sample of Terms for Topics Detected using GPT-4 (Appendix Table A8 reproduced)

Topic	Terms
Abortion Rights	roe v. wade 2021 debate, #abortionaccess, #abortionlaw, my body my, #prolife, abortion restriction laws, stop the bans, justice for abortion rights, georgia abortion ban, repeal hyde amendment, abortion access in 2017, #stopthebans, partial birth, texas abortion bill, telemedicine for abortion, abortion, abortion ban laws, #stand-withpp, gestational age limits, pro life anti abortion
Climate Action	#netzero, climate change and agriculture, climate change mitigation, united nations climate, carbon footprint, climate change 2013, earth temperatures, protect our planet, scientific consensus on climate, global climate change, climate change convention, stop global warming now, #cop21 paris, paris climate agreement 2015, greenhouse gas emission reduction, climate action summit 2019, carbon capture storage, trump climate change, climate change adaptation, global average temperature
Immigration	border security and enforcement, stricter immigration laws, bipartisan framework for comprehensive immigration, customs and border, scrapping the diversity lottery, #deportationforce, asylum seeking policies, travel ban supreme court, #refugeeasylum, secure border initiative, legal status pathway, us immigration policy, fight against deportation, ban on muslim countries, support immigration reform, family detention centers, 2017 sanctuary cities controversy, comprehensive immigration reform 2022, abolish ice movement, migrant detention
Racism or Race Relations	ethnic minorities, race-related hate crimes, reparations for african americans, #charlestonshooting, racism in sports 2008, stand against racism 2022, hate-crime, racial identity, #racismisavirus, racial inequality awareness, protests against racial injustice, black lives matter, racial inequality in 2007, blacklivesmatter, racial profiling in police, address racial disparities, racial equality now, color should not matter, #whitesupremacy, say no to racism
Redistribution of Income	wealth tax proposals, redistribution of income, taxation for redistribution, economic inequality issues, wealth redistribution policies, income redistribution efforts, redistribution debate, income redistribution policies, redistribution, progressive income redistribution, income gap widening, tax on wealth, #progressivetax for fairness, income redistribution and economy, progressive tax system, income redistribution mechanisms, economic redistribution strategies, basic income, #redistribution, top1percent
Tax Policy	cryptocurrency tax rules, #irs, digital tax policy, tax rate adjustments, obama tax policy, tax code simplification, tax cuts, #obamatax, tax bracket changes, small business tax, tax policy under obama, tax policy review, trump tax cuts, income tax rates, tax legislation, tax avoidance, impact of taxes, tax policy debates, #taxes, property tax
Welfare State	public housing assistance programs, welfare state critique, #socialsecurity, public pension scheme reform, welfare budget, income redistribution policy, social expend, public healthcare, social welfare policies, food stamps, welfare rights, welfare state issues, food stamp program 2013, reform the welfare state, benefit system, unemployment benefits policy 2018, welfare reform policies, economic impact welfare state, #publicwelfare, welfare state development

By incorporating this validation step, we have demonstrated that our topic detection method not only balances precision and recall effectively but also provides a reliable foundation for subsequent stance detection and analysis. The detailed metrics provided in Table A2 should reassure that the generated keyphrases are both accurate and comprehensive, addressing the initial concern about the potential limitations of keyword-based filtering.

6. You asked us about the scope of our analysis and the perceived significance of political expression among academics. Specifically, you pointed out that we only focused on a set of political topics that themselves make up a small set of the tweets that are posted by academics. You suggested that this limitation may

be problematic given the diverse interest and audiences that the journal may attract.

Quotes:

“One feature of the work is its coverage of a large set of scholars and the selection of some important political topics. However, the analysis is limited to selected political topics only, which are only a tiny fraction (3.5%) of the tweets posted by the academics. Thus, the work might have limited immediate interest to the diverse audience across disciplines.”

Response: Thanks a lot for raising this comment. This touches on two dimensions. First, the role that academics may play more broadly in societies as a potentially important vector of knowledge production and also increasingly as knowledge disseminators or translators. We do think that the latter role has grown in importance given the decline in professional journalism that has been widely documented. We further do perceive that academics are being listened to and that there is an often quite institutionalised role for academics.

Regarding the specific comment about the extent of political expression we offer some further clarifications.

Extent of Political Expression Among Academics: While it was observed that the analysis focuses on a selected set of topics around which we measure political expression, these topics constitute 3.5% of the total tweets by academics. This may not seem like a large number but it is worth emphasising the broader context of this finding. Our updated analysis, as reflected in Table 1 of the manuscript, shows that 75% of academics in our sample have engaged in political expression by making at least one non-neutral tweet on climate action, cultural liberalism, or economic collectivism during the study period. This significant level of engagement indicates that political discourse is not a negligible amount of academic activity on social media, rather than a marginal one. That is: politically salient expression is a near universal feature of academic expression online. Naturally, as the topic space may be expanded, the share increases. We focused on a subsample of topics and themes that social science research suggests as being particularly relevant during our sample period as they mark key socio-economic fault lines. We plan to highlight this point more clearly in the Introduction and Discussion sections to underscore the relevance and importance of our analysis.

Proportional Significance of Political Tweets: Turning to the number, as you noted, the 3.5% may initially seem like a small fraction. Yet, it represents a meaningful portion of the overall discourse when placed in context. To make this sharper we report on further analysis of academic social media expression. In a random subsample of academics, we have further analysed the social media content and classified the content using similar techniques. We observe that 16.70% of tweets are related to research, while 6.74% mention political figures. These two numbers provide a natural anchor: one fifth of the research related social media content can be attributed to political expression on the set of topics that we have focused on. This comparison highlights that political expression is a substantial and non-negligible part of the academic conversation on Twitter. carried out further

7. You raised a concern of using the geolocation of an academic based on the most recent affiliation as reported in Open Alex. You suggested that this be updated to reflect the affiliation at the time when the academic posted content.

Quote:

“User geolocation is based on their most recent affiliation recorded in OpenAlex. But wouldn't it make more sense to use the “current” affiliation when the tweet was posted?”

Response:

We appreciate this insightful suggestion regarding the use of time-varying affiliation data for geolocation, rather than relying on the most recent affiliation recorded in OpenAlex. We did have a look at this in the preparation of the manuscript. We agree that this adjustment provides a more accurate reflection of the academic's location at the time of each tweet, which could influence the analysis of their political expression and related behaviours.

In response, we have updated our methodology to incorporate *time-varying institutional affiliations* using data from OpenAlex and yearly university rankings using data from QS World University Rankings. Specifically, we now extract the institution name from the author metadata associated with each publication by academics between 2016 and 2022. This approach allows us to provide a more precise mapping of their geolocation and university ranking at the year each tweet was posted. The methodological adjustments are outlined in detail in the updated Methods/Data section of our manuscript:

“We accessed all publications by academics within this period through OpenAlex and extracted the institution name from the author metadata for each publication. By relying on article-level affiliation data, we ensure that our analysis reflects changes in an author's affiliation over time, rather than a single, static affiliation. We use yearly aggregation based on the modal affiliation of each academic to maintain consistency.”

Moreover, to handle cases where an academic did not publish in certain years, we generated a complete set of institution-year pairs by carrying forward the nearest available ranking and backfilling as necessary to ensure continuity

“In cases where an academic did not publish in certain years, we generated a complete set of institution-year pairs, filling any gaps by carrying forward the nearest available ranking and backfilling as necessary to ensure continuity.”

This refinement is particularly significant when analysing results based on categorical data, such as whether an academic is affiliated with a top 100 university, ranked 101-500, 501-1500, or 1500+, and whether they are based in the US or elsewhere. We chose to introduce the 4-category system for university rankings, rather than simply using a top 100 or not, by keeping the top 100 category intact and disaggregating the 100+ category into three distinct groups. This allows us to explore whether a monotonic, linear, or non-linear pattern

holds across these different tiers, providing deeper insights into how institutional ranking is associated with academic behaviour and expression. This disaggregation is particularly useful as it allows us to examine variations within the lower-ranked institutions, which are often more diverse and may exhibit different patterns of behaviour.

We recognize that this adjustment may impact results, especially for academics who transition between these categories during the study period. To quantify the stability and movement across these categories, we first analysed the proportion of academics remaining within each ranking category from 2016 to 2022.

For the country of affiliation—US versus non-US—we note that the vast majority, 87% of academics, are consistently based at US or non-US institutions. However, for university rankings, with our new 4-category system (1-100, 101-500, 501-1500, 1500+), we find that 50.3% of academics remain in the same category from the beginning to the end of the sample period. This change reflects a higher degree of mobility within the university ranking categories, driven by both academics moving between universities of different rankings and the rankings themselves changing every year. This increases noise, particularly in the lower-ranked categories, where institutions are more likely to fluctuate. Although the academic rank of an institution may vary—especially outside the very top—this analysis is still valuable as it allows us to capture the dynamic nature of academic institutions and the associated movement of academics across institutions, which in turn helps us understand patterns in academic expression over time.

Impact on main patterns and findings

To assess the impact of this adjustment, we repeated key analyses—specifically Figures 2-4 (and their appendix equivalents 4-6)—using the new time-varying location and ranking data. The overall results for the US versus non-US classification remain consistent with our previous analysis that used time-invariant geolocation. In general, we observe a monotonic pattern from 1-100 to 101-500 to 501-1500, with the 1500+ category being particularly noisy and showing some degree of nonlinearity. This additional result further enriches our analysis, as it highlights the complexity of academic behaviour in relation to institutional rank, particularly in the lower tiers.

Our main claim that top 100 institutions differ significantly from others generally holds. This refined analysis not only confirms our original findings but also provides new insights into the variability and non-linearity present in the lower-ranked categories. These variations align with the idea that these categories encompass a diverse range of academic institutions, thereby enhancing the robustness and validity of our findings.

We have included excerpts from Figures 2-4 in the referee report to illustrate these comparisons, demonstrating how political expression, tone, and style of expression vary with time-varying affiliation data. This adjustment ensures that our analysis more accurately reflects the dynamic nature of academic affiliations and rankings, thereby enhancing the robustness and validity of our findings.

8. You asked us about the comparability of the two samples. You highlight that we focus on academics tweeting in the first and last 6 months of the period that data was collected for, but you were wondering about the characteristics of this subset vis-a-vis the broader population.

Quote:

“Second, sample representativeness remains unknown, which may confound the results. Although the study mentioned the sample selection process based on tweeting in the first and last 6 months in the focused period, there are insufficient details provided about the characteristics of this subset and the extent of representativeness of the broader academic population or even of the original 100k sample.”

Response:

The reason to focus on a sample that we observe consistently over time is because, in addition to cross-sectional comparisons of academics across topics along salient dimensions such as gender or the country of academic affiliation, we want to make longitudinal comparisons. This is where sample selection that is done explicitly in a way to provide us with a longitudinal sample is relevant as this way we can ensure that the trends that are documented are not conflating compositional changes of the sample. We do expect that the balancing, on average, may actually attenuate the observed differences as we focus on a population of users that appears to be regular social media users.

To ensure the reader can understand better the sample differences we have added a new comparison table in the appendix between the balanced and unbalanced sample in the Appendix as Table A9.

Our balanced sample consists of approximately 99,274 academics who tweeted at least once in the first six months of 2016 and at least once in the last six months of 2022. This ensures a consistent presence across the entire study period. We compared this balanced sample with the unbalanced sample, which includes all individuals who tweeted at least once at any point during the period from 2016 to 2022.

The comparison reveals that the balanced and unbalanced samples are very similar in terms of non-Twitter metrics such as publication metrics (e.g., number of citations, number of works published), gender distribution, and discipline distribution. This similarity suggests that our balanced sample is representative of the broader academic population in these dimensions, at least on average.

However, and not surprisingly, the unbalanced sample reveals notable differences in the underlying Twitter metrics. This is entirely expected: we observe a disproportionately higher standard deviation in metrics such as the number of tweets, likes, and followers. This indicates that the unbalanced sample is noisier, with more extreme values that could introduce unnecessary variability into the analysis. By focusing on the balanced sample, we reduce this noise, resulting in a more reliable and stable dataset that is necessary to be able to ensure that the trends we are documenting are not masking compositional sample changes.

We have added this comparison to the Methods section of the manuscript, highlighting that while the balanced and unbalanced samples are comparable in non-Twitter metrics, the balanced sample is intentionally chosen to minimise noise and enhance the robustness of our results. We believe that this approach provides a more accurate reflection of academic behaviour on Twitter over time, without the distortions that could arise from extreme cases in the unbalanced sample.

Table: Balancedness Test: Comparison of Balanced and Unbalanced Populations (Table A9 reproduced)

Panel (a): Balanced Population						
Variable	Mean	Median	SD	Min	Max	N
Publication Metrics						
Nbr. Citations	1370.02	213	4523.99	1	238736	99274
Impact Factor (2Y)	16.93	9	38.56	0.02	4205	99274
Nbr. Works	57.58	24	114.38	1	7711	99274
Demographics						
Humanities	0.005	0	0.07	0	1	99274
STEM	0.291	0	0.45	0	1	99274
Social Sciences	0.098	0	0.30	0	1	99274
Gender: Male	0.60	1	0.49	0	1	99274
Twitter Metrics						
Nbr. Likes	3210.85	1673	5736.33	0	293797	99274
Nbr. Followers	1112.70	533	9145.94	0	1087504	99274
Nbr. Accounts Followed	749	529	1116.55	0	191923	99274
Nbr. Retweets	843.72	276	2117.91	0	181268	99274
Nbr. Posts	1575.24	1032	2052.13	4	117088	99274
Panel (b): Unbalanced Population						
Variable	Mean	Median	SD	Min	Max	N
Publication Metrics						
Nbr. Citations	1083.30	139	3969.42	1	331147	219273
Impact Factor (2Y)	15.49	9	30.96	0.02	2539	219273
Nbr. Works	48.26	18	110.51	1	8465	219273
Demographics						
Humanities	0.002	0	0.04	0	1	219273
STEM	0.262	0	0.44	0	1	219273
Social Sciences	0.071	0	0.26	0	1	219273
Gender: Male	0.59	1	0.49	0	1	219273
Twitter Metrics						
Nbr. Likes	5909.80	847	18705.81	0	2017914	219150
Nbr. Followers	2464.89	264	272177.95	0	126938095	219150
Nbr. Accounts Followed	713.74	328	2176.71	0	382058.5	219150
Nbr. Retweets	732.73	141	5519.14	0	1513703	219150
Nbr. Posts	3639.87	293	12652.17	1	734460	219150

9. You asked us to conduct a statistical test of power-law distribution:

Quote:

“To claim the distributions follow a power-law, there is a rigorous statistical test procedure to follow [Clauset, A., Shalizi, C. R., & Newman, M. E. (2009). Power-law distributions in empirical data. SIAM review, 51(4), 661-703.]”

Response:

Thank you for highlighting the need to rigorously substantiate the claim that the distributions of academic influence metrics follow a power-law. In response to your comment, we have now conducted a thorough statistical analysis following the methodology outlined by Clauset, Shalizi, and Newman (2009), as suggested.

This analysis involved fitting a power-law distribution to the data on content creation, engagement (likes received), public reach (followers), and citations among academics. The results, now included in the main body of our manuscript, confirm that these metrics do indeed follow a power-law distribution. Specifically, we calculated the power-law exponents (α) for each metric, with values ranging from 2.636 to 3.337, as detailed in Appendix Table A6, **reproduced here for reference**. Additionally, the low Kolmogorov-Smirnov (KS) statistics obtained from our analysis further validate the goodness of fit for the power-law model across these metrics.

We have incorporated this additional validation into the manuscript, as described in the revised subsection on the "Distribution of Academic Influence on Twitter." This section now includes a brief overview of the statistical analysis and its results, with full details presented in Appendix Table A6. This ensures that our claims regarding the power-law nature of these distributions are well-supported by both visual and statistical evidence.

Table: Power-Law Fit Parameters for Academic Influence Metrics (Table A6 reproduced)

Metric	Power-Law Exponent (α)	Minimum Value (x_{\min})	KS Statistic	Log-Likelihood Ratio
Content Creation (Posts)	3.337	2650	0.007	-1.112 (p = 0.399)
Engagement (Likes Received)	3.146	15491.5	0.015	-3.660 (p = 0.112)
Public Reach (Followers)	2.636	1332	0.011	0.403 (p = 0.001)
Citations Count	2.929	15808	0.033	-12.281 (p = 0.003)

Notes: This table summarizes the results of fitting a power-law distribution to various metrics of academic influence on Twitter: content creation (posts), engagement (likes received), public reach (followers), and citations count. The analysis follows the method by Clauset et al. (2009), which estimates the power-law exponent (α) and the minimum value (x_{\min}) above which the data follows a power-law distribution. The KS statistic measures the goodness of fit, where lower values indicate a better fit to the power-law model. The log-likelihood ratio compares the fit of the power-law model to that of an alternative distribution, with the p-value indicating the statistical significance of the comparison. The results confirm that these metrics exhibit power-law behavior, indicating significant inequality across these academic influence measures.

You next had a range of suggestions to improve the accessibility of the manuscript for the readership of the journal.

10. You suggested that the introduction may be too long and suggested that part of the results discussion that is covered could be moved to the discussion section. You also indicated that we should streamline the alignment of the methods and results section to follow the data construction section. You also had some suggestions to ensure we directly point out panels of figures immediately.

Quote:

"The readability of the manuscript needs to be improved substantially, considering the intended general readership of the journal. For instance, the Introduction section may be too long, where the result discussion part could be moved to the Discussion section and the road map may not be necessary. The methods and results sections are dense. The results should be structured in a way that logically builds from the data and methods, allowing the reader to easily trace how the key findings were derived. Many terms like economic collectivism are better explained immediately after being mentioned. The results description parts should clearly indicate which panel in a figure is being referred to. By strengthening the coherence between the data, methods, and findings, the paper would be significantly easier for readers to follow and understand the full scope of the research."

Response:

Thanks a lot for this suggestion. We have streamlined the introduction significantly, moving parts of the results discussion into the appropriate discussion section, and adjusted the flow of the methods and results sections to better align with the data construction. We have also incorporated your suggestions regarding the figures. While we do not provide a line-by-line breakdown of the changes, we have incorporated these adjustments in our letter to the editor to reflect the overall improvements in the paper's structure and clarity.

11. You raised concerns regarding the lack of in-depth empirical analysis exploring the underlying causes of the differences in online academic expression and the broader implications of these findings. Additionally, you asked about the significance of subgroup differences (e.g., humanities vs. STEM) and whether confounders like field composition explain these variations. Lastly, you mentioned the relatively small proportion of relevant tweets (3.5%) and whether this impacts the interpretation of the results.

Quotes:

“the work mentioned in the Introduction section that there is no work on the role of academics' online expressions in the perceptions of science. However, this question is far from being answered and rigorously analyzed in the current work. Instead, it presented a number of descriptive statistics and their variations along different dimensions, without any explorations/in-depth empirical analysis of the underlying causes and broader implications of these differences. Thus, the conceptual novelty of this work is also limited.”

“focusing on the interpretation and implications of the results, the paper lacks deep discussions about the domain significance and interpretation of the results. For example, the Discussion section is more of repeating the result section rather than a contextualized discussion. Another issue is how significant in terms of effect size is the stance difference between, e.g., humanities and STEM and others? Are they meaningful, considering that only 3.5% tweets are relevant (Table 3). Is there any confounder like the field composition of academics that explain the differences? Why is egocentrism on the rise? What are the factors that contribute to such a rise? Again, the study has not examined the underlying causes of the noted differences”

Response:

We appreciate the referee's thoughtful feedback and understand the importance of highlighting the scope and limitations of the current study more clearly. We would like to emphasise that the focus of this paper is descriptive, aiming to present new patterns and raise pertinent questions about the online expression of academics rather than to establish causal relationships. By documenting these patterns across various dimensions—such as field, gender, and institutional ranking—we lay the groundwork for future studies that may engage in more targeted causal inquiries.

As such, this study seeks to provide the first large-scale empirical evidence of the variations in academic expression across disciplines and demographic factors, which is necessary before asking causal questions. We now acknowledge more explicitly in the Introduction and Discussion sections that the primary goal of this paper is descriptive, and causal analysis falls outside its current scope.

In the Discussion, we have expanded on the broader implications of these findings and posed new questions that emerge from our descriptive analysis. We believe that these questions will inspire further research into the social and academic factors that drive differences in online expression. We explicitly note that exploring the causes behind these observed trends, such as why egocentrism is on the rise, is a valuable next step for future research.

The following excerpt is added to the Discussion:

“Importantly, this study is descriptive, documenting trends and patterns without asserting causal relationships. We highlight differences in academic expression across subgroups (e.g., fields, gender, country) but refrain from causal claims. Future research could investigate the drivers behind these variations, such as institutional incentives or visibility, and explore how algorithms may amplify certain voices and shape public perception of science.”

The following is added to the Introduction:

“Our study provides a large-scale descriptive analysis of academics' political expression on Twitter across subgroups, laying the groundwork for future research to explore the underlying causal factors, such as academics' motivations for political expression—whether driven by name recognition, ideological beliefs, or the desire to disseminate knowledge.”

Additionally, we also contextualize our findings in the Discussion:

“In line with other research on academic expression, such as Williams and Ceci (2023), our findings echo concerns about ideological bias in social media, where progressive voices tend to dominate. This aligns with Langbert's (2018) analysis of political affiliations in academia, suggesting that Twitter may amplify these biases. Studies like Crockett(2017) on digital moral outrage explain how online environments lower the barriers for expressing outrage, increasing its frequency and intensity. This framework supports our findings that academic discourse on highly polarized topics, such as climate action, becomes more extreme and morally charged on social media. The incentives of platforms like Twitter, which reward emotionally charged content with greater visibility, may further exacerbate this polarization. Similarly, Ke, Ahn, and Sugimoto (2017) find that a small, highly active subset of social and computer scientists disproportionately shapes conversations on Twitter, paralleling our findings. Ferrara (2015) echoes concerns about algorithmic amplification, where non-expert academics with large followings may distort the perception of scientific consensus.”

References:

- Crockett, M.J., 2017. Moral outrage in the digital age. *Nature human behaviour*, 1(11), pp.769-771.
- Ferrara, E., 2015. " Manipulation and abuse on social media" by Emilio Ferrara with Ching-man Au Yeung as coordinator. *ACM SIGWEB Newsletter*, 2015(Spring), pp.1-9.
- Ke, Q., Ahn, Y.Y. and Sugimoto, C.R., 2017. A systematic identification and analysis of scientists on Twitter. *PLoS one*, 12(4), p.e0175368.
- Langbert, M., 2018. Homogenous: The political affiliations of elite liberal arts college faculty. *Academic questions*, 31(2).
- Williams, W.M. and Ceci, S.J., 2023. How politically motivated social media and lack of political diversity corrupt science. In *Ideological and Political Bias in Psychology: Nature, Scope, and Solutions* (pp. 357-375). Cham: Springer International Publishing.

On Subgroup Differences and Methodological Robustness

To address the referee's concern regarding the significance of subgroup differences (e.g., humanities vs. STEM), we conducted further analysis using a fixed effects regression model to control for individual-level heterogeneity over time. This allowed us to more rigorously assess whether the subgroup differences we observed in the descriptive analysis are statistically significant. The model we used is as follows:

$$Expression_{it} = \lambda_t \times Time_t + \mu_i + \epsilon_{it}$$

where $Expression_{it}$ represents the expression of political or behavioural identity by author i at time t . λ_t captures the effect of time (by year and month) on the expression, μ_i represents individual-level fixed effects, and ϵ_{it} is the error term.

This approach allows us to control for individual differences and isolate the true temporal trends in academic expression, ensuring that observed patterns are not artefacts of sample composition changes. By **partialling out individual fixed effects**, we ensure that the trends we identify reflect true changes over time rather than differences between individuals.

The results of this approach are illustrated in Appendix Figures A26, A27, and A28, which provide examples of within-individual trends for various subgroups (gender, field, and US vs. non-US academics). These figures, reproduced here for reference, demonstrate that the level differences in these trends remain consistent with our main findings. For example, Appendix Figure A26 shows that male academics exhibit more toxic expression than female academics, while there is no significant difference in emotionality or reasoning. Appendix Figure A27 highlights that humanities scholars are more egocentric, toxic, and display higher emotionality compared to their counterparts in social sciences and STEM fields. Similarly, Appendix Figure A28 illustrates that US-based academics show greater support for techno-optimism and cultural liberalism, especially after 2020. These examples confirm that the trends observed in our main analysis are robust even when accounting for within-individual variation.

Figure A26: Academic Expression Over Time by Gender, Within-user estimates

Figure A27: Academic Expression Over Time by Fields, Within-user estimates

Figure A28: Academic Expression Over Time by whether academic's institution is US based, Within-user estimates

These consistent within-user trends improve validity of the subgroup differences we report and allay concerns about potential confounders related to sample composition.

Furthermore, to ensure that the observed subgroup differences in stances are not confounded by variations in tone (e.g., emotionality, egocentrism, toxicity), we ran additional regressions controlling for these variables. The results, shown in Appendix Figures A15 and A16 (reproduced previously in point #4), confirm that subgroup differences (e.g., between STEM and humanities academics on climate action) remain statistically significant even after controlling for tone. For instance, the coefficient for being in STEM is statistically distinguishable from the coefficient for being in social sciences or humanities at the 95% confidence level when the dependent variable is stance on climate action.

On the 3.5% of Relevant Tweets

We acknowledge that the 3.5% share of politically salient tweets is relatively small, as the referee noted. However, as detailed in our previous response (#6), political expression remains a significant dimension of academic discourse. Notably, 75% of academics in our sample have posted at least one non-neutral tweet on topics such as climate action, cultural liberalism, or economic collectivism, signalling that political expression is a nearly universal feature of academic engagement, even if it constitutes a smaller portion of total tweets. Additionally, when compared to other categories—such as the 16.7% of tweets related to research and 6.74% that mention political figures—the 3.5% figure represents 20.9% of the volume of tweets about research. This underscores that political discourse, though proportionally smaller, plays a crucial role in academic conversations on Twitter. We have clarified this point in the main text to reinforce the significance of political discourse in our analysis.

12. You pointed out that the current measurements are based on posted topical tweets and asked if the likelihood of an academic tweeting about a particular topic in the first place should be incorporated into the analysis. You noted that under the current method, differences in activeness regarding political tweets might not be distinguished.

Quote:

“Moreover, all the measurements are based on the posted topical tweets. But how likely is it that a user tweets that topic in the first place? Should we incorporate this likelihood into the calculation of the various measures? Under the current quantifications, it cannot distinguish users with different levels of activeness in posting political tweets, as it calculates the fraction among topical tweets.”

Response:

We appreciate this insightful comment regarding the likelihood of an academic tweeting about a topic as a factor in the analysis. The manuscript includes over-time plots (Appendix Figure A22) that illustrate the average number of tweets per topic for each month from 2016 to 2022. This figure presents the average number of tweets per author by month, for each topic that has a stance (i.e., pro-, anti-, or neutral-). The category “Any Political” aggregates stances across the seven topics, reflecting the overall political stance of academics over time. These averages are based on a sub-sample of topical tweets selected for stance detection, helping to clarify the volume of political expression and the extent to which individuals engage with politically salient topics. This ensures that our results are not overly influenced by a few very vocal and influential accounts that post frequently.

Figure A22: Topical tweets over time, average (2016-2022)

Our focus has been on ensuring that the comparisons made across subgroups—such as field or gender—are robust and reflect genuine differences in expression. In addition to the provided figures, we highlight that our approach is designed to normalise stance detection metrics across topical tweets, minimising the influence of individual activity levels. By normalising across topics, we ensure that even if an academic tweets more or less frequently about certain issues, the stance expressed within each topical category is given equal weight.

Additionally, as mentioned in our previous point, we conducted a within-user analysis that isolates individual author-level fixed effects, allowing us to control for individual differences in overall activity levels and topical engagement. This approach accounts for the likelihood of tweeting on a particular topic by focusing on temporal trends within each individual's posting behaviour. By partialling out these individual-level fixed effects, we ensure that our analysis reflects true within-user variation over time, rather than being skewed by differences in activeness or topic preferences across users.

This method provides a more accurate picture of how trends in academic expression evolve, independent of individual tweeting frequency or topical engagement.

We have updated the Appendix section “*Robust Trends - Partialling Out Individual Fixed Effects*” that discusses this method, making clear that individual level fixed effects are able to capture individual level variation in topical engagement. Here is the updated excerpt:

“This approach is particularly advantageous for several reasons. First, it acknowledges the heterogeneity inherent in academic behaviors, recognizing that individual authors may exhibit varying levels of expression based on personal, disciplinary, or topical engagement factors. By controlling for individual author-level fixed effects, we isolate the temporal trends in political expression while accounting for these variations. Second, by controlling for these author-specific effects, we can more accurately isolate the influence of time on Expression, yielding insights that are more reflective of true temporal trends rather than artifacts of the composition of the author cohort.”

We believe these clarifications and the inclusion of new visual data provide a clearer understanding of how we address the likelihood of topical engagement, while maintaining the robustness of the stance detection and comparison across different groups of academics.

13. You also had a few minor observations and typos.

Quote:

“L229: Results about Twitter reach and expertise are not discussed in the main text.”

Response: We updated the relevant areas (L146-154 and L184-189) to include these results. Here are the relevant excerpts.

L146-164 reads

“Academics with both higher social media reach (i.e., more followers) who are non-experts in the subject matter, as judged by their own research output, employ language that appears to be least supportive of climate action. Their revealed stance in support of climate action is notably less pronounced compared to the stance revealed by academics with high reach who have revealed expertise on climate change matters. Additionally, low-reach subject matter experts display significantly higher support for climate action than their high-reach, non-expert counterparts.”

L184-189 reads

“Academics with higher social media reach who lack demonstrated expertise in the specific socio-economic topics exhibit less progressive stances on cultural liberalism and economic collectivism. However, those with both high reach and relevant expertise tend to be more progressive. This distinction between reach and expertise highlights the potential divergence in expressed academic stances, where visibility on social media may not always correlate with subject matter depth.”

Quote: “L718: two field categories mentioned but 3 categories are used in the main text.”

Response: This was a typo. This is now fixed.

Quote: “Table 1: why are the N different under the Tone and Style of Expressions group?”

Response: This was a typo. This is now fixed.

Quote:

“There are numerous minor typo errors and inconsistencies in the text:

Line 36: “there is are ...”

Line 44: “Too date” -> “To date”

Line 139: thing -> think

Line 555-556: “I”, “my”, “Future drafts will” Also footnote on page 16.

L250: “continuous continuous”

L476: “Open AI”

Response: Thanks. These are now fixed.

Responses to reviewer 2: Communication/Communication/Information on social media, Social Media data analysis, science communication & science engagement online

Thanks a lot for your detailed comments and suggestions. In the following responses, we attempted to identify the main points of suggestions you raised in your report and provide an overview of how we attempted to adjust the manuscript to reflect your comments.

1. You commended on the use of OpenAlex for location, affiliation, and ranking information. You raised the concern that there is the potential of missing information in the OpenAlex dataset. You also raised concerns about the need to validate the OpenAlex information. You pointed us to <https://link.springer.com/article/10.1007/s11192-023-04923-y> as a potential reference that illustrates potential challenges around the data.

Quote:

“You used OpenAlex for location, affiliation, rankings, but you don't mention the potential missing information that is in the OpenAlex dataset nor do you speak to validating the information you found (e.g.

<https://link.springer.com/article/10.1007/s11192-023-04923-y>”

Response. Thank you for your thoughtful comment regarding the use of OpenAlex (OA) data. We have carefully considered your feedback and would like to address each of your concerns.

Potential Missing Information and Validation: In our analysis, we ensured that there were no missing values for affiliation, location, or ranking for any authors included in the final dataset. Specifically, we checked the final sample and confirmed that there were no missing (NA) values at the author level for these variables. As mentioned in the Methods section under "Location, Institutional Affiliation, and Rankings," we did not use OA directly for rankings. Instead, we used OA to obtain affiliations, which we then matched with the QS World University Rankings. This approach allowed us to achieve full coverage for all authors in analyses requiring these variables.

Unit of analysis relevant for the degree of missingness The study that you cited focuses on potential missing information at the *journal article level* within Open Alex. Our analysis however, is at the *level of the individual academic*. For most academics this implies we have multiple data points in any given year to infer their institutional affiliation. This reduces the overarching concern of missing institutional affiliation information. Further, the reference you cited also highlights, in their Figure 5, that the extent of missing information *at the article level* has drastically declined over time. In fact, the reference you provide stops in 2012 in terms of the temporal breakdown of the missingness of affiliation information at the article level. Our focus is on data from 2016-2022, a period where OA's coverage is most comprehensive.

Relevance and Coverage of OpenAlex It is worth giving a bit more context on Open Alex. OA is built on the foundation of the Microsoft Academic Graph (MAG) and CrossRef, making it the largest open-access bibliometric repository available, second only to Google Scholar, which is not open-access. OA is widely recognized for its extensive coverage, including retracted papers, which are challenging to find in other databases. For example, OA has been shown to cover about 90% and 75% of all articles found in Scopus and Web of Science (WoS), respectively, and often includes significantly more works (Jiao et al., 2023, Culbert et al., 2024). Furthermore, OA nearly doubles (and in some cases more than triples) the number of works found by Scopus in various source categories. This broad coverage is one of the reasons why bibliometric services like Overton have switched to using OA as their primary data source (Chawla, 2022).

Strengths and Limitations of OpenAlex While OA generally outperforms other databases in terms of coverage, some limitations do exist, particularly regarding author names, references, citations, and institutional data, as highlighted in recent studies (e.g., Culbert et al., 2024; Akbaritabar et al., 2023). However, our reliance on the 2016-2022 data, which has the most comprehensive coverage within OA, mitigates many of these concerns. Additionally, by focusing on author-level affiliation data, we avoid many of the inconsistencies associated with article-level data.

In summary, we are confident that our use of OA data is both appropriate and robust for the purposes of our study. The decisions we made in data selection and processing ensure that the potential issues highlighted by the reviewer are addressed effectively. We have cited relevant studies in our updated Methods section to provide further context and validation of our approach. We hope this response clarifies our methodology and addresses your concerns.

References:

- Akbaritabar, A., Theile, T. and Zagheni, E., 2023. Global flows and rates of international migration of scholars (No. WP-2023-018). Max Planck Institute for Demographic Research, Rostock, Germany.
- Chawla, D.S., 2022. Massive open index of scholarly papers launches. Nature.
- Culbert, J., Hobert, A., Jahn, N., Haupka, N., Schmidt, M., Donner, P. and Mayr, P., 2024. Reference coverage analysis of openalex compared to web of science and Scopus. arXiv preprint arXiv:2401.16359.
- Jiao, C., Li, K. and Fang, Z., 2023. How are exclusively data journals indexed in major scholarly databases? An examination of four databases. *Scientific Data*, 10(1), p.737.

2. You asked about the switch from GPT 4 to GPT 3.5 Turbo for stance detection and narrative extraction. You asked whether this made any difference to if we had used GPT 4 throughout. You point to our validation exercise that yielded an F1 around 84 to 92 and ask whether this would have been higher with GPT 4. You asked whether we validated this choice on a small subset.

Quote:

“You switched from GPT 4 to GPT 3.5 Turbo for stance detection and narrative extraction. I'm wondering what difference, if any, you might find had you used GPT 4 for this part of the work? You mention that you validated to F1 around 84 to 92, would this have been higher with GPT 4? Did you test a small subset to see if 3.5 and 4 gave similar results?”

Response Thank you for your insightful question regarding our use of GPT-3.5 Turbo for stance detection and narrative extraction, particularly in comparison to GPT-4. We recognize the importance of validating our model choice, especially given potential differences in performance between GPT-3.5 Turbo and GPT-4. We followed your suggestion and carried out a range of further validation.

First, we validated our stance detection method using the ACL SemEval-2016 Task 6 dataset, as detailed in the updated stance detection validation appendix. We compared GPT-3.5 Turbo and GPT-4 on this dataset and found their performance to be very similar. The average F1 scores for both models were consistently high, with GPT-4 only slightly outperforming GPT-3.5 Turbo on the Legalisation of Abortion topic (84.36 vs. 79.52). For other topics, including Feminism, Hillary Clinton, and Donald Trump, GPT-3.5 Turbo either matched or exceeded GPT-4's performance, with F1 scores ranging from 84.18 to 92.44.

Further, we expanded our validation by comparing GPT-3.5 Turbo with GPT-4, GPT-4o-mini, and GPT-4o across the specific topics in our study. We analysed 8,400 tweets labelled by all four models over 10 iterations, totaling 336,000 predictions. After filtering out unrelated labels, we found strong consistency across models, with agreement rates and F1 scores improving as model quality increased. GPT-4o, the most advanced model, achieved the highest agreement rates and F1 scores, confirming the robustness of our stance detection approach. For reference, we reproduce the results presented in Appendix A2 here.

Table A2: Comparison of Agreement and F1 Scores Across GPT Models

Comparison	Agreement (Modal)	Agreement (Iterations)	F1 (Modal)	F1 (Iterations)
GPT-3.5-turbo vs GPT-4o	0.781	0.772	0.806	0.795
GPT-3.5-turbo vs GPT-4	0.750	0.738	0.772	0.756
GPT-3.5-turbo vs GPT-4o-mini	0.684	0.681	0.696	0.691

Notes: Table presents a comparison of agreement rates and average F1 scores (F_{avg}) between different GPT models for stance detection. **Agreement (Modal)** refers to the proportion of identical stance predictions when considering the modal stance across 10 iterations per tweet. **Agreement (Iterations)** measures this proportion for each individual iteration treated as a unique instance. **F1 (Modal)** represents the average F1 score when one model's modal stance is used as the "true" label to evaluate another model's performance. Similarly, **F1 (Iterations)** does the same but treats each iteration as a separate instance. Higher values in these metrics indicate stronger consistency and accuracy in stance classification.

Given the minimal performance differences and the significantly lower cost of GPT-3.5 Turbo (around 20 times cheaper), we are confident that it provides an optimal balance of accuracy and cost-efficiency for our tasks. The updated appendix includes these validation results, further supporting the reliability of our method.

3. You also had a range of smaller comments that we present here and iterate on.

Quote: *“On line 245, you say (e.g., lower impact factor)... >> I believe this should be “higher impact factor” --- should impact factor be defined anywhere for audience?”*

Response: There was an error in the Egocentrism panel of Figure 4. After revising it, it is consistent with the text. We also defined the impact factor in the relevant footnote.

Quote: *“In the abstract you say “over 100,000 scholars”, but on page 3 you say 99,274 scholars.”*

Response: “Over” was a typo. Converted to “nearly” 100,000 scholars.

Responses to reviewer 3: Ethical issue in research involving social media data sources

Remarks to the author:

"It was agreed with the editor that I would only review the reflection of research ethics in the paper, particularly related to the use of online data. That is a very modest task in this case, as there is absolutely no attention for the ethics of scraping Twitter and academics' profiles without informing them, let alone asking for consent. If the authors hold the view that this is legitimate for reasons of public interest, they could at least state and explain this (see e.g., Stommel & De Rijk, 2021).

Stommel, W., & Rijk, L. D. (2021). Ethical approval: None sought. How discourse analysts report ethical issues around publicly available online data. Research Ethics, 17(3), 275-297.

“:

Response:

Thank you for your insightful comments on the ethical considerations of using publicly available online data in our research. We fully acknowledge the importance of adhering to rigorous ethical standards, especially when dealing with online data, as highlighted in Stommel & De Rijk (2021). We provide below a discussion of the various considerations.

1. Data Collection and Platform Compliance:

Our study did not involve scraping data from Twitter or OpenAlex; instead, we used their respective APIs, adhering to the terms of services of both platforms, which are designed to provide access to public data for academic research under controlled conditions. We followed data collection practices that respected user privacy throughout the process.

2. Public vs. Private Data Distinction:

We recognize that the public availability of data on platforms like Twitter does not automatically negate the ethical considerations of using that data in research. However, our study focused exclusively on data that users have willingly made publicly accessible. In line with Stommel & De Rijk (2021), we considered the public nature of Twitter posts—analogueous to conversations in a public square—where users are generally aware that their content is visible to the broader public. This context informed our decision that explicit informed consent was not required, as participants have already consented to the public sharing of their content under Twitter's terms of service.¹

3. Anonymization and Retrievability:

¹ “By publicly posting content, you are directing us to disclose that information as broadly as possible, including through our APIs, and directing those accessing the information through our APIs to do the same.” Twitter. https://x.com/en/privacy/previous/version_17

To further protect the privacy of individuals, all data in our study was anonymized and aggregated, ensuring that no personal identities could be traced back to specific data points. We understand the concern regarding the retrievability of quotes, as noted by Stommel & De Rijk (2021).

Although our study did not involve direct quotations or verbatim excerpts from Twitter posts, we took several precautions to ensure that individual users could not be identified, even through indirect means. We removed all personally identifiable information, including names, usernames, Twitter IDs, tweet IDs, and raw tweet content, thereby preventing any direct linkage to individuals. Additionally, the data was aggregated to a level where individual tweets could not be traced back to specific users, summarising content across time periods rather than quoting directly. Where references to users were necessary, we employed anonymized codes without any linkage to original identities, further safeguarding user privacy.

Furthermore, all data was securely stored in encrypted formats, both in transit and at rest, with access strictly controlled and limited to essential research personnel. These measures ensured that the data was used responsibly, in line with ethical guidelines and the need to protect user privacy.

4. Reflection on User Expectations:

We also considered the expectations and intentions of Twitter users. Although these expectations can vary, we operated under the assumption, supported by prior research, that users of public platforms like Twitter understand their posts are accessible to anyone, including researchers. Nevertheless, we took care to anonymize and aggregate the data to minimise any potential harm or breach of user expectations.

5. Ethical Approval and Research Integrity:

Our study received IRB approval from Imperial College London, which confirmed that no significant ethical issues were identified in the research protocol. The ethical review process ensured that our methods complied with the Data Protection Act 2018 and GDPR, further safeguarding participant privacy and confidentiality.

Moreover, we ensured that all data storage and processing were conducted securely, using encrypted storage solutions like Google BigQuery. This compliance with institutional and regulatory standards further reinforces our commitment to ethical research practices.

6. Importance of Study and Justification for Data Use:

In line with the suggestion by Stommel & De Rijk (2021) to clarify the importance and benefits of the research, we believe our study contributes significantly to understanding how academics engage with politically salient topics on social media. This understanding is crucial for public discourse and policymaking, especially in an era where social media increasingly influences public opinion. We believe that the societal benefits of this research justify the use of publicly available online data, while our rigorous ethical considerations mitigate any potential risks to individuals.

7. Transparency and Future Research:

We are committed to transparency in our research process. A replication package, including the aggregated data and analysis code, will be made available to the academic community. This approach aligns with the call for more detailed reporting of ethical considerations, as emphasised by Stommel & De Rijk (2021).

In addition, we have included an "Ethics and Inclusion Statement" in the revised manuscript, which outlines our adherence to ethical guidelines, anonymization protocols, and our efforts to ensure inclusion and transparency in the study.

We believe that our approach to using publicly available online data is ethically sound, adhering to both the letter and the spirit of ethical research guidelines. We appreciate the opportunity to further clarify these points and have strengthened our manuscript to reflect these considerations, ensuring that our research is both ethically responsible and valuable to the broader academic community.

Reference:

- Stommel, W. and Rijk, L.D., 2021. Ethical approval: None sought. How discourse analysts report ethical issues around publicly available online data. *Research Ethics*, 17(3), pp.275-297.

Responses to reviewer 4: Political attitude, Communication/Information on social media, Social Media data analysis

Thanks a lot for your detailed comments and suggestions. In the following responses, we attempted to identify the main points of suggestions you raised in your report and provide an overview of how we attempted to adjust the manuscript to reflect your comments.

1. You commended the breadth and scale of the analysis and pointed out that this comes with some advantages and disadvantages. Specifically, you highlighted that the comparison of means between two populations induce us to lose a lot of the detail and nuance.

Quote:

“I think a good research report raises more questions than provides answers, and this manuscript clearly does that. The authors seek to characterize expression via social media of academics at large, and to highlight ways in which those who work as academics may differ in their online communication--in form and content--from those who are not. The effort here is at a very broad scale, which directly relates to both its chief advantages and disadvantages. The breadth of the analysis, examining a large number of expressions affiliated with academics tweeting, provides us with some good broad comparisons in terms of who is tweeting, how they are tweeting, their reach, and how this may relate to expertise.”

Response: Thanks a lot for this point. You do highlight an important challenge with any form of applied research that leverages high dimensional data: the need for dimensionality reduction to make the underlying data accessible. Regression techniques and comparisons of means is one of the most common methods for such dimensionality reduction. We took specific care and attention to work with data that pertains to a balanced panel of academics on social media: that is, academics that consistently are active on the platform. This ensures that the temporal variation in the salient political expression can be studied over time and that differences are not confounding compositional effects of the underlying estimating sample.

We have highlighted this approach in more detail in the paper (Appendix section “*Robust Trends - Partialling Out Individual Fixed Effects*”) and added some further appendix tables that highlight the compositional differences this implies along with a detailed justification due to the desirable feature of allowing longitudinal comparisons.

2. You suggest that we double down on the role of “public intellectuals” in the social media age. You suggested that we spend more effort to confirm that many “public intellectuals” are not particularly academically influential in their own respective academic fields.

Quote:

“I think there is more value in confirming that many “public intellectuals” in the social media age are not especially influential in their own fields. When asked to name an astronomer or

astrobiologist, many in the public would default to Carl Sagan, for example. For a physicist, almost certainly Albert Einstein, though they may also be aware of Richard Feynman thanks to his public efforts, and there can be little doubt these folks were also influential to their field. Particularly in the case of junior scholars, the public communication online may also be a proxy for their informal connections in the field, and may lead to more academic opportunities thanks to "name recognition."

But there are significant challenges to drawing broadly from this analysis. The authors do a good job, I think, of highlighting some of these. They are very clear, for example, that the expressiveness on a topic like climate change may have more to do with comfort or desire to publicly communicate on the topic rather than convictions. And they note that those academics with the greatest public reach may not be representative of the top experts in the field."

Response:

Thank you for the valuable feedback regarding the role of "public intellectuals" in the social media age. We have taken your suggestion on board and made several additions to clarify and emphasise the distinction between social media visibility and academic influence.

First, we added a footnote in the results section "*Distribution of Academic Influence on Twitter*" to highlight the weak correlations between academic recognition (e.g., citation counts) and social media influence (followers, likes). Specifically, the correlations between citations and likes (-0.02), citations and followers (0.09), and citations and content creation (-0.02) demonstrate that many prominent public intellectuals on social media are not necessarily leading scholars in their fields. The footnote reads:

"The weak correlation between academic recognition and social media influence underscores that many prominent public intellectuals online gain visibility through public engagement rather than scholarly achievements, often holding lower academic credentials while commanding significant public attention, thus widening the gap between social media influencers and established academic experts."

This directly supports the point raised in your review, confirming that social media prominence does not equate to traditional academic authority.

Additionally, we have expanded the discussion section to reinforce this point further. The revised text now states:

"The over-representation of certain views—especially from high-reach, low-expertise academics—combined with the under-representation of experts could result in miscommunication of scientific consensus. This dynamic suggests that many 'public intellectuals' gain visibility more through public engagement than through traditional academic influence, reinforcing the idea that online prominence does not necessarily reflect expertise in their respective fields. As such, those with the greatest public reach may not represent the top scholars, potentially distorting public perceptions."

These changes ensure that we have explicitly addressed your concerns by highlighting the divergence between online influence and academic standing, making it clear that visibility on

social media often stems from public engagement rather than scholarly impact. This provides further evidence of the need for caution when interpreting the influence of "public intellectuals" in shaping public discourse.

3. You also suggested that we raise more awareness on the fact that Twitter represents a specific part of the information/attention/social media ecosystem globally and that we should also mention potential changes to algorithms that may have been latent.

Quote:

*"The article slowly eases us into the sample, which is drawn from Twitter. While the case can be made that Twitter is of primary importance as a social media platform, and the authors do so to a certain degree by noting its use by journalists in particular, it is clearly not *all* social media use. Moreover, the forms of expression on Twitter are shaped heavily by the ways in which the algorithm (here to indicate both social and technological factors) surface and reinforce certain types of expression and evidence. This can be as simple as (early) length constraints, but a number of folks (e.g., Pfitzner, Garas, & Schweitzer) have shown that emotional divergence predicts sharing of Tweets. It seems likely that academics may be successful on Twitter specifically to the degree to which they may code-switch and adapt to the discourse community specific to Twitter. I think it would be wrong to assume that the same kinds of dynamics are necessarily present on other platforms.*

And, of course, it would likely be wrong to assume that the constraints of Twitter/X have remained the same over time. We have seen a number of evolutionary forces shift the constraints, along with very explicit attempts to change discourse on the site. Some of these changes have resulted in academics abandoning the platform altogether. There has been some broad criticism of over-emphasizing Twitter in research because of the availability of the corpus. I don't think we have to go that far--there is a case to be made for its influence--but I do think it's important to bear in mind that platforms matter, and the expressions that are found on a social media platform are necessarily shaped by a negotiation between users and the structures of the platform."

Response:

Thank you for raising this important point regarding the specific role of Twitter within the broader social media ecosystem and the potential effects of algorithmic changes on the dynamics of academic expression. We fully agree that Twitter represents only one platform in a diverse media landscape and that the structure of expression on Twitter is shaped by its algorithms and platform constraints. We would like to clarify several key points:

Twitter's Role in the Media Ecosystem: While we recognize that Twitter is not representative of the entire social media ecosystem, we emphasise its particular relevance as a platform for original content and as a "multiplier" due to its popularity among journalists, politicians, and policymakers. Twitter's influence extends beyond academia, with its use as a dissemination tool, particularly for high-impact topics like climate change and socio-economic policy. This focus on Twitter is thus grounded in its outsized role in shaping public discourse, which is well documented.

Algorithmic Amplification and Expression Dynamics:

We also agree that algorithmic amplification or selection plays a relevant role in shaping expression dynamics. However, we want to clarify that our analysis does not focus on the extensive margin of expression—such as how often a post is retweeted or engaged with. Instead, we measure genuine expression, where each academic in our econometric exercise is given equal weight, regardless of engagement levels. Although we do not track engagement metrics, we acknowledge that algorithmic amplification could still increase the visibility of certain topics, which may, in turn, encourage more political expression among academics. To address this, we have added a clarification in the discussion section about how algorithms may shape broader discourse. Here is the relevant addition in the discussion section:

“While our focus is on genuine expressions rather than engagement metrics like retweets or likes, algorithmic amplification could still influence the visibility and prevalence of certain topics, potentially encouraging more political expression among academics and shaping broader discourse.”

You correctly note that algorithmic factors on Twitter—such as emotional divergence—can influence the visibility and sharing of content. This is consistent with research suggesting that emotionally charged content is more likely to be shared. In response, we have addressed this concern in our paper by conducting an analysis of whether “how” expressions (tone, emotionality, egocentrism, and toxicity) might confound the observed patterns in the “what” of academic expression. Specifically, we performed regressions controlling for expression style (emotionality, egocentrism, and toxicity) and found that these factors do not significantly alter subgroup differences in stance.

This suggests that while Twitter’s algorithm may amplify emotional content, our findings of political stance differences across subgroups are robust and not driven by tone or emotional expression. This analysis is detailed in our response to a different referee and included in Table A7, and Appendix Figures A15 and A16, which show that controlling for these “how” variables does not alter the substantive patterns in our data. We reproduce the analysis here for convenience:

Raw Correlations Between Stances and “How” Expressions

First, we examined the raw correlations between each stance (the “what”) and three key measures of expression style (the “how”)—emotionality, egocentrism, and toxicity. The results are presented in the table below:

Table A6: Raw Correlations Between Stances and Emotionality, Egocentrism, and Toxicity

Stance	Emotionality	Egocentrism	Toxicity
Climate Action	-0.008	-0.027	-0.033
Techno-Optimism	-0.017	-0.041	-0.055
Behavioural Adjustment	-0.008	0.000	-0.014
Cultural Liberalism	0.021	0.057	0.085
Economic Collectivism	0.004	0.015	0.000

Note: This table presents the raw correlations between different political stances and three key variables related to how academics express themselves: **Emotionality**, **Egocentrism**, and **Toxicity**. These variables represent the tone and style of expression, with emotionality capturing the ratio of affective to cognitive words, egocentrism measuring the prevalence of self-referential language, and toxicity capturing the likelihood of harmful or aggressive language. The stances—Climate Action, Techno-Optimism, Behavioural Adjustment, Cultural Liberalism, and Economic Collectivism—are defined as the net stance per author on each topic. The raw correlations are all near zero, suggesting minimal direct relationship between stance and expression style, with the highest correlation being 0.085 for Cultural Liberalism and Toxicity. This indicates that “how” things are said does not heavily influence the political stance expressed by academics.

These correlations show that the relationship between stance and expression style is minimal in most cases. For example, the correlation between emotionality and stance is very close to zero, with the largest observed correlation being 0.085 between cultural liberalism and toxicity. This suggests that, in general, expression style does not have a strong raw correlation with stance.

Regression Analysis to Control for “How” Expressions While Examining Subgroup Differences: In the second step, we explored whether the observed differences across subgroups could be confounded by expression style. To do this, we ran a series of regressions for each stance, comparing subgroup differences (e.g., gender, field, rankings, and country) while controlling for one of the three “how” expressions at a time (emotionality, egocentrism, or toxicity). Specifically, we estimate the following model:

$$\text{Stance}_{it} = \alpha + \beta_1 * \text{GroupVar}_{it} + \beta_2 * \text{HowExpression}_{it} + \varepsilon_{it}$$

Where GroupVar_{it} represents a subgroup indicator (e.g., male, STEM, US), and $\text{HowExpression}_{it}$ represents one of the three expression styles (emotionality, egocentrism, or toxicity). This allows us to assess whether controlling for the “how” variables alters the subgroup coefficient β_1 . If β_1 remains stable across the three models, it indicates that tone and style of expression do not confound subgroup differences.

Climate-related stances: We begin by presenting the subgroup differences for climate-related stances (Climate Action, Techno-Optimism, and Behavioural Adjustment), as originally shown in the paper (Figure 2). We then display the updated results after controlling for emotionality, egocentrism, and toxicity.

Figure: Subgroup Comparisons for Climate-Related Stances (Figure 2 Reproduced)

Figure: Subgroup Comparisons for Climate-Related Stances, Controlling for Emotionality, Egocentrism, and Toxicity (Appendix Figure A15 reproduced)

As seen in the new figure, the coefficients for subgroup differences remain consistent across the three regressions, each controlling for one of the expression styles. This suggests that the "how" expressions—whether emotionality, egocentrism, or toxicity—do not confound the subgroup differences for these climate-related stances.

Socio-economic stances: Next, we replicate this analysis for the stances on Cultural Liberalism and Economic Collectivism. Again, we start by presenting the subgroup differences from the original analysis (Figure 3) and then show the updated results with controls for emotionality, egocentrism, and toxicity.

Figure: Subgroup Comparisons for Economic and Cultural Stances (Figure 3 Reproduced)

Figure: Subgroup Comparisons for Economic and Cultural Stances, Controlling for Emotionality, Egocentrism, and Toxicity (Appendix Figure A16)

reproduced)

Again, we observe that the coefficients for subgroup indicators remain stable across all three regressions, indicating that controlling for the “how” expressions does not significantly alter the subgroup effects. Thus, it is unlikely that emotionality, egocentrism, or toxicity confound the observed differences across subgroups for these stances.

Based on our results, we find that expression style (the “how”)—whether emotionality, egocentrism, or toxicity—does not confound the observed subgroup differences in stance (the “what”). The coefficients for subgroup differences remain consistent regardless of whether we control for these expression styles, indicating that the variations we observe are primarily driven by the content of the statements rather than how they are expressed.

Potential Changes in Twitter's Algorithm Over Time: While concerns about evolving platform dynamics, especially under Twitter's recent ownership changes, are valid, we believe this has limited impact on our study. Our analysis covers the period from 2016 to 2022, before the major platform changes initiated by Elon Musk in 2023 and 2024. Although there has been a noticeable shift of users, including academics, to alternative platforms like Mastodon, Bluesky, and Threads, these migrations are primarily on the extensive margin. Due to network effects, which reinforce the value of remaining on a platform with a large and established user base, the intensive margin of migration (i.e., active engagement on alternative platforms) is likely lower. This means that while some users may have created accounts elsewhere, Twitter continues to serve as the primary platform for most of their activity.

Additionally, our use of a balanced sample—focusing on academics consistently active over time—further mitigates concerns about sporadic engagement or algorithm-driven distortions. We also perform robustness checks, controlling for individual-level fixed effects, ensuring that unobserved factors do not confound the results.

Balanced vs. Unbalanced Sample: Lastly, you also suggested that we introduce some more information about the populations and samples that are studied earlier in the paper. In response to your suggestion, we have made some changes in the text and also added some further qualification that helps improve the understanding of the estimating sample that we are working with. One particular salient dimension is the distinction between a balanced and an unbalanced sample. Our focus is on longitudinal comparisons within individual users over time. This ensures that we can interpret genuine differences in trends, rather than differences that may be driven by sporadic entry and exit of individual users in the estimating sample.

To ensure the reader can understand better the sample differences we have added a new comparison table in the appendix between the balanced and unbalanced sample in the Appendix as Table A9. This is reproduced here for reference:

Table: Balancedness Test: Comparison of Balanced and Unbalanced Populations (Table A9 reproduced)

Panel (a): Balanced Population						
Variable	Mean	Median	SD	Min	Max	N
Publication Metrics						
Nbr. Citations	1370.02	213	4523.99	1	238736	99274
Impact Factor (2Y)	16.93	9	38.56	0.02	4205	99274
Nbr. Works	57.58	24	114.38	1	7711	99274
Demographics						
Humanities	0.005	0	0.07	0	1	99274
STEM	0.291	0	0.45	0	1	99274
Social Sciences	0.098	0	0.30	0	1	99274
Gender: Male	0.60	1	0.49	0	1	99274
Twitter Metrics						
Nbr. Likes	3210.85	1673	5736.33	0	293797	99274
Nbr. Followers	1112.70	533	9145.94	0	1087504	99274
Nbr. Accounts Followed	749	529	1116.55	0	191923	99274
Nbr. Retweets	843.72	276	2117.91	0	181268	99274
Nbr. Posts	1575.24	1032	2052.13	4	117088	99274
Panel (b): Unbalanced Population						
Variable	Mean	Median	SD	Min	Max	N
Publication Metrics						
Nbr. Citations	1083.30	139	3969.42	1	331147	219273
Impact Factor (2Y)	15.49	9	30.96	0.02	2539	219273
Nbr. Works	48.26	18	110.51	1	8465	219273
Demographics						
Humanities	0.002	0	0.04	0	1	219273
STEM	0.262	0	0.44	0	1	219273
Social Sciences	0.071	0	0.26	0	1	219273
Gender: Male	0.59	1	0.49	0	1	219273
Twitter Metrics						
Nbr. Likes	5909.80	847	18705.81	0	2017914	219150
Nbr. Followers	2464.89	264	272177.95	0	126938095	219150
Nbr. Accounts Followed	713.74	328	2176.71	0	382058.5	219150
Nbr. Retweets	732.73	141	5519.14	0	1513703	219150
Nbr. Posts	3639.87	293	12652.17	1	734460	219150

Our balanced sample consists of approximately 99,274 academics who tweeted at least once in the first six months of 2016 and at least once in the last six months of 2022. This ensures a consistent presence across the entire study period. We compared this balanced sample with the unbalanced sample, which includes all individuals who tweeted at least once at any point during the period from 2016 to 2022.

The comparison reveals that the balanced and unbalanced samples are very similar in terms of non-Twitter metrics such as publication metrics (e.g., number of citations, number of works published), gender distribution, and discipline distribution. This similarity suggests that our balanced sample is representative of the broader academic population in these dimensions, at least on average.

However, and not surprisingly, the unbalanced sample reveals notable differences in the underlying Twitter metrics. This is entirely expected: we observe a disproportionately higher standard deviation in metrics such as the number of tweets, likes, and followers. This indicates that the unbalanced sample is noisier, with more extreme values that could introduce unnecessary variability into the analysis. By focusing on the balanced sample, we reduce this noise, resulting in a more reliable and stable dataset that is necessary to be able to ensure that the trends we are documenting are not masking compositional sample changes.

We have added this comparison to the Methods section of the manuscript, highlighting that while the balanced and unbalanced samples are comparable in non-Twitter metrics, the balanced sample is intentionally chosen to minimise noise and enhance the robustness of our results. We believe that this approach provides a more accurate reflection of academic behaviour on Twitter over time, without the distortions that could arise from extreme cases in the unbalanced sample.

4. You also raised a point on the role of the one-to-one correspondence between a Twitter user base and the academic user base.

Quote:

"Finally, the analysis makes the necessary assumption (I believe) that there is a one-to-one correspondence between a Twitter user and an academic. While that is true for some--I tweet under a single account--I would guess that a very large number of academics distribute their tweets through academic and non-academic accounts. There is a general messiness here between academics, their public communications, their academic tweets, and tweeting in general. Are these distinctions important? Given the broad thrust of the argument here, perhaps not. We might assume that the differences from the mean--both to the general population and in terms of the difference between twitter impact and traditional impact--are large enough that it is not vital. But what we gain in general brush strokes we lose in precision."

Response:

Thank you for your insightful comment regarding the assumption of a one-to-one correspondence between a Twitter user and an academic. We acknowledge that this assumption may not always hold, as some academics may manage multiple accounts for different purposes (e.g., professional vs. personal accounts) or use pseudonyms on social media, which can complicate the accuracy of our matching process.

To address this concern, we have considered relevant work, such as the paper by Mongeon et al. (2022), which discusses the complexities of matching scholars to their Twitter accounts using open data sources like OpenAlex and Crossref Event Data. As highlighted in that paper, a single academic can have multiple identifiers (e.g., OpenAlex author IDs) and may not always tweet under a single identifiable account, leading to potential discrepancies in the one-to-one assumption. While the matching algorithm employed in our study (and others) prioritises precision over recall, this can lead to missed matches or inaccuracies, particularly for academics with multiple accounts or who don't follow consistent naming conventions across platforms.

While this introduces some limitations to the precision of the dataset, the large sample size and the consistency of the observed trends across different subgroups suggest that the broader patterns identified are likely to remain robust despite these potential mismatches. Our discussion now explicitly acknowledges this limitation and its potential impact on our results. Here is the relevant excerpt:

"Additionally, our assumption of a one-to-one correspondence between Twitter users and academics may not always hold, as some manage multiple accounts or use pseudonyms. However, the large sample size and consistent trends across

subgroups suggest that the key patterns remain reliable despite these potential mismatches.”

5. You commended our methodological approach and the use of generative AI tools for sentiment analysis, while noting the challenges posed by sentiment analysis tools in general. You also highlighted the potential for our framework to inspire further exploration in this area.

Quote:

“Those limitations notwithstanding, I applaud the methodological precision and clear description of that method provided by the authors--as well as the clarity of exposition throughout. It is unfortunately rare that research reports in this area are thorough enough to be reproducible and this bucks that standard. It also provides an excellent framework for further exploration of these relationships. It's difficult not to see some of the current methods of sentiment analysis as blunt instruments, but this work raises questions that should be highly generative moving forward.”

Response: Thank you for your positive feedback on the clarity and reproducibility of our methodology. One of our primary goals was to leverage generative AI models, such as GPT-3.5 and GPT-4, to improve the accuracy and scalability of key tasks like topic detection, stance detection, and gender classification. To further strengthen the reliability of our results, we incorporated additional validation steps across all aspects of the analysis, using multiple models and datasets.

For stance detection, we performed cross-validation with several models (GPT-3.5, GPT-4, GPT-4o), achieving strong agreement rates and high F1 scores, which confirmed the robustness of our approach. In topic detection, using established datasets such as SemEval-2016, we consistently demonstrated high F1 scores, showing the effectiveness of our method. For gender classification, we tested the accuracy of GPT-3.5 on a comprehensive dataset of over 147,000 unique names, and achieved very high precision and recall.

We hope that this work not only solidifies the findings of our study but also provides a useful framework for the broader research community to explore these relationships further using advanced AI techniques.

5. You suggested we move some of the discussion around the sample and the methodology a bit earlier in the paper, without compromising the methodological section towards the back of the manuscript.

Quote:

“It's rare that I have so few recommendations for changes. I think at least touching a bit more on the limitations I've noted above would be helpful. At the very least, orienting us more toward the corpus quite early in the discussion would be helpful. This isn't an analysis of social media use--it's an analysis of Twitter use. There is a strong case to be made for why Twitter is a good thing to use here, but this should be clearly spelled out in the introduction, the abstract, and maybe even the title. (As someone with the name of defunct platforms in

my own titles, I recognize the danger in this, but clarity should prevail.)

And along the same lines: I recognize the format in which the research report is presented is standard for the journal, though leaving off methods to the end is a bit discombobulating for this reader. At least some of the grit of the method needs to be reported early on--what are the populations/samples, what are the measures, etc., in broad terms earlier on so as to reduce the ambiguity. It's a much better read if bounce around the manuscript and come back up for the results, but I think being a bit more precise about what is being measured would help to understand the conceptual contribution early on."

Response:

Thank you for your thoughtful feedback. We agree with your suggestion to clarify the role of Twitter in the study and provide more methodological context earlier in the paper. In response, we've made the following adjustments:

We replaced the term 'social media' with 'Twitter' in the specific line of the abstract that describes the core of our study: *"This paper describes patterns in academics' expression online found in a unique global dataset covering nearly 100,000 scholars linking their Twitter content to academic records."* This ensures clarity from the outset that our analysis focuses specifically on Twitter as a platform. The introduction now outlines the rationale for using Twitter, emphasising its outsized influence as a platform for content dissemination, particularly among journalists, policymakers, and academics.

To address your point about introducing methodological grit early on, we have added more detail earlier in the results section on the data sources, population size, and the tools used, providing readers with essential context without compromising the fuller methods discussion later in the manuscript.

Additionally, we have streamlined the introduction and moved portions of the discussion related to the results into the dedicated discussion section. We have also addressed the limitations you and other reviewers pointed out in the discussion. These changes were made to ensure a clearer exposition, as suggested. While the changes are modest, they collectively enhance readability and align with the guidance provided by the editor, other referees, and your suggestions.

Responses to the editor

First of all, we wanted to express our gratitude for the detailed editorial guidance in navigating the reviewer comments. We discuss each of your main points in turn and explain how we addressed your and the reviewer comments in turn in a brief overview with the more detailed responses being provided for each reviewer.

1. Quote: “(1) Reviewer #1 raises a series of important technical concerns about the validity of your analyses: the accuracy of LLM-based measures and the results, for example, is of special importance. Please present further validation of your measures and additional cross-checks (possibly with existing studies) in order to address Reviewer #1's concerns.”

Response.

Thanks a lot for the detailed guidance. We have reported on a broad range of additional cross checks and cross-referenced our methods with existing studies, as outlined below. These should address the points raised by Reviewer 1. We present these in more elaborate form and detail in our point-by-point response to Reviewer 1. Here, we provide a brief description for convenience.

- Stance Detection Validation

We carried out further validation exercises that can speak to the suitability of the stance detection approach that was taken. This involved validation with GPT 3.5, GPT-4. We have added a revised Appendix Table A1 that highlights the suitability of our approach.

- Topic specificity

We have highlighted that the techniques used are suitable to detect topics. This yielded a further Appendix Table A2 that highlights a high degree of model agreement across four models: GPT-3.5-turbo, GPT-4, GPT-4o, and GPT-4o-mini. This is strong evidence of the broad robustness of our stance detection approach.

- Topic Detection Validation

For topic detection – as opposed to stance detection –, we validated our methodology using the SemEval-2016 dataset. We generated topic-specific dictionaries using GPT-4 and applied them to the SemEval dataset. The analysis suggests high recall and balanced F1 scores across topics, demonstrating the effectiveness of the topic detection method we employed. The results are presented in Appendix Table A3.

- Gender Classification Validation

We have carried out an additional verification of the suitability for gender classification based on first names. We utilised a comprehensive dataset of 147,269 unique names from official sources. The GPT-3.5-turbo model was used to predict the gender associated with each name. The results showed very high precision (0.9866), recall (0.9871), and F1 scores (0.9868). The results are presented in Appendix Table A4 of the updated manuscript, reproduced here for ease of reference.

- **Cross-Checks with Existing Studies:**

We added comparing our methods to those in Alturayef et al. (2023) and Marageh et al. (2023), highlighting how dynamic keyword extraction improves accuracy over static models. Our approach addresses limitations of static dictionaries noted in prior studies like Alturayef et al. (2023), emphasizing the importance of context-aware keyword extraction.

We also referenced Chen et al. (2023) to explain how our stance detection uses a similar error-filtering technique to refine outputs and enhance precision. Additionally, we noted that our model achieves average F scores between 79.52% and 92.44%, outperforming traditional models from SemEval-2016 (e.g., RNNs and CNNs with transfer learning, which ranged from 56.28% to 67.82% (Mohammad et al., 2016; Zarella and Marsh, 2016)). These older methods required extensive feature engineering, whereas our approach dynamically adapts without manual tuning, while using a versatile method, illustrating the advantages of LLMs in handling nuanced political discourse on platforms like Twitter.

We believe that these additional validations across stance detection, topic detection, and gender inference illustrate the overall broad reliability and accuracy of the LLM-based models that we employed in our study.

We also contextualize our findings within existing research. Williams and Ceci (2023) highlight ideological bias on social media, where progressive voices dominate, consistent with Langbert's (2018) analysis of political affiliations in academia, suggesting amplification of these biases on platforms like Twitter. Crockett (2017) explains how digital platforms lower the barriers for expressing outrage, intensifying moral and polarized discourse—supporting our observation of emotionally charged content on topics like climate action gaining more visibility. Similarly, Ke, Ahn, and Sugimoto (2017) find that a small, highly active subset of social and computer scientists disproportionately shapes conversations on Twitter, paralleling our findings. Ferrara (2015) echoes concerns about algorithmic amplification, where non-expert academics with large followings may distort the perception of scientific consensus.

- **Reproducing Key Results Using Different Techniques**

To further improve the validity of our findings, we conducted additional robustness checks. Controlling for tone, we found the patterns remained consistent across the cross-section. We also emphasized the importance of our previously reported appendix, which controlled for individual-level fixed effects (unobserved heterogeneity), showing no change in results. These updates have been highlighted in the manuscript to address concerns raised by Reviewer #1 and Reviewer #4.

References:

- Alturayef, N., Luqman, H. and Ahmed, M., 2023. A systematic review of machine learning techniques for stance detection and its applications. *Neural Computing and Applications*, 35(7), pp.5113-5144.
- Chen, H., Wang, X., Chen, H., Song, Z., Jia, J. and Zhu, W., 2023. Grounding-Prompter: Prompting LLM with Multimodal Information for Temporal Sentence Grounding in Long Videos. *arXiv preprint arXiv:2312.17117*.

- Crockett, M.J., 2017. Moral outrage in the digital age. *Nature human behaviour*, 1(11), pp.769-771.
- Ferrara, E., 2015. " Manipulation and abuse on social media" by Emilio Ferrara with Ching-man Au Yeung as coordinator. *ACM SIGWEB Newsletter*, 2015(Spring), pp.1-9.
- Ke, Q., Ahn, Y.Y. and Sugimoto, C.R., 2017. A systematic identification and analysis of scientists on Twitter. *PLoS one*, 12(4), p.e0175368.
- Langbert, M., 2018. Homogenous: The political affiliations of elite liberal arts college faculty. *Academic questions*, 31(2).
- Maragheh, R.Y., Fang, C., Irugu, C.C., Parikh, P., Cho, J., Xu, J., Sukumar, S., Patel, M., Korpeoglu, E., Kumar, S. and Achan, K., 2023, December. LLM-take: theme-aware keyword extraction using large language models. In *2023 IEEE International Conference on Big Data (BigData)* (pp. 4318-4324). IEEE.
- Mohammad, S., Kiritchenko, S., Sobhani, P., Zhu, X. and Cherry, C., 2016, June. Semeval-2016 task 6: Detecting stance in tweets. In *Proceedings of the 10th international workshop on semantic evaluation (SemEval-2016)* (pp. 31-41).
- Williams, W.M. and Ceci, S.J., 2023. How politically motivated social media and lack of political diversity corrupt science. In *Ideological and Political Bias in Psychology: Nature, Scope, and Solutions* (pp. 357-375). Cham: Springer International Publishing.
- Zarrella, G. and Marsh, A., 2016. Mitre at semeval-2016 task 6: Transfer learning for stance detection. *arXiv preprint arXiv:1606.03784*.

2. Quote: "(2) As Reviewer #3 notes, currently there is no attention to ethics nor justification about your approach in the manuscript. In revision, please thoroughly discuss the ethical rationale for your study, and the steps you took to minimise harms. Please also transparently describe the nature of the ethics review and approval that took place for your study (as shared with us)."

Response:

We appreciate your feedback on the importance of thoroughly discussing the ethical rationale for our study and the steps taken to minimise potential harms. In our response to reviewer 3 we have expanded in much more detail, but below we provide a summary for your convenience. We provide a succinct summary of the arguments in the below 5 bullet points.

1. **Public Nature of Social Media Content:** Our study is based on social media content that is publicly accessible and produced with the intention of being shared in a public domain. Social media platforms are designed for users to engage in public discourse, and thus, it is reasonable to expect that such content could be used for research purposes. The data we analysed is already in the public domain, and our study does not involve private or sensitive information.
2. **Importance of Research on Social Media Discourse:** Social media has become a significant source of news and information for a large segment of the population. Understanding the structure of discourse among individuals who actively choose to participate in these platforms is vital for comprehending how information is

disseminated, consumed, and potentially manipulated. Our research contributes to this understanding by analysing patterns and trends in social media interactions, which is critical in an era where these platforms increasingly shape public opinion.

3. **Lack of Established Ethical Practices Among Social Media Content Producers:** Currently, there is limited understanding of how social media content producers themselves engage in ethical practices or adhere to codes of conduct. This gap in knowledge further justifies our study, as it highlights the need to explore how social media content producers operate in the realm of social media. Our research, therefore, provides valuable insights into this largely unregulated area that has been subject to significant interest given the inherent tension between freedom of expression and individual liberties.
4. **Impact of Algorithms and Anonymization:** We acknowledge that algorithms can influence the visibility and spread of social media content, potentially impacting the discourse in ways that are not immediately apparent. Our study takes this into account by focusing on descriptive patterns that are anonymized to protect individual identities. Furthermore, we are ignoring the actual *reach dimension* of the content, but rather, want to gain maximal understanding of the breadth of topic engagement and expression across individuals whereby each individual is given the same weight. The data analysis is conducted in a way that ensures no harm to the individuals involved, and no identifiable personal information is used in our research outputs.
5. **Ethics Review and Approval:** Our study was subjected to a rigorous ethics review process and received approval from the Institutional Review Board (IRB) at Imperial College. We have attached the full IRB review to this response for your reference. The ethical considerations discussed with the IRB, including the public nature of the data, anonymization techniques, and the overall importance of the research, align with the arguments presented here. This ensures that our study adheres to the highest ethical standards.

Additionally, we have now included an "Ethics and Inclusion Statement" in the revised manuscript, summarising our ethical considerations, anonymization protocols, data handling procedures, and our commitment to transparency.

We hope this addresses your concerns and provides a clear rationale for the ethical approach taken in our study. Thank you for your valuable feedback, which has helped us enhance the transparency and rigour of our ethical considerations.

3. Quote: "(3) *In revision, please add more extensive discussion on limitations, including key points raised by reviewers such as the limitations of the underlying datasets (Reviewers 2 and 4), the choice of Twitter as a data source, and concerns about sample representativeness and potential confounders (Reviewer 1).*"

Response:

We appreciate the constructive feedback regarding the need for a more extensive discussion of the study's limitations, particularly in relation to the concerns raised by Reviewers 1, 2, and 4. We have provided detailed responses for each of the reviewers' comments in their respective own response letters. We provide here a summary on how we incorporated these points in the revised manuscript.

- **Choice of Twitter as a Data Source:**

We acknowledge that Twitter is not representative of the entire population and that its user base may not reflect the diversity of the broader public. However, our choice of Twitter as a data source is deliberate and grounded in its role as a platform where key influencers—such as journalists, politicians, and academics—actively engage. These groups are often seen as thought leaders or intellectual elites, and their interactions on Twitter can have significant downstream effects on public discourse, media agendas, and potentially even policy-making. These are now clearly mentioned in abstract, introduction and discussion, as suggested by the reviewers.

- **Role of Twitter in Shaping Broader Media Landscape:**

One premise that positions the relevance of our paper is the potential for social media platforms like Twitter to influence traditional media and, by extension, the broader public sphere. Ample (anecdotal) evidence suggests that such a pipeline may exist. Traditional media outlets increasingly compete for information and attention, and what trends on social media can significantly shape the narratives that appear in mainstream media. This creates a feedback loop where content that gains traction on Twitter may influence broader news agendas and, ultimately, public policy discussions. The limitations of Twitter as a data source thus align with our research aim: to highlight the disproportionate influence that certain groups and trends on social media can exert on wider society.

- **Sample Representativeness and Potential Confounders:**

We recognize the concerns about sample representativeness and the potential influence of confounders. Given Twitter's user demographics and the nature of its platform, the sample may not capture the full spectrum of public opinion. However, this limitation is also a key focus of our study. The very fact that Twitter represents a distinct audience—one that includes many voices that may be perceived as influential—makes it a valuable subject of study, despite its lack of representativeness. We also acknowledge the potential role of confounders, such as the influence of algorithms or external events. Regarding the algorithms, in our response, we highlighted that this is why we think it is particularly prudent to work with a balanced panel of academics that engage in online discourse as this allows us to make longitudinal comparisons. Further, the sampling strategy implies that we are not giving more weight to (very) frequent participants in online discourse. These factors, while complicating the analysis, do not diminish the value of understanding the specific dynamics at play within this particular segment of the public discourse.

Thank you again for your insightful comments. We believe that acknowledging and addressing these limitations strengthens our manuscript, improves its transparency and provides a more nuanced understanding of the complex dynamics at play in social media-driven discourse.

4. Quote: “(4) *In your revision, please implement additional analyses and discussion to address Reviewer 1’s concerns about the depth of the analysis.*”

Response:

Thanks a lot. On the substantive additional analysis that were highlighted by the reviewer. We provide a summary of these analyses here along with the key observations.

- **Aggregation bias concerns**

The reviewer asked us to provide a view of the components that go into the indices that make up Economic collectivism and Cultural liberalism. The specific concern relates to the possibility that the aggregation into an index may conflate individual trends into a broader topic-related trend. We opted for the aggregation into an index for ease of presentation to reduce the dimensionality of the empirical observations into specific themes. Yet, we agree that this may raise the question about whether the underlying latent trends are an artefact of the aggregation. To alleviate these concerns, we conducted a detailed analysis to examine the trends within each composite measure and its respective components. This has been added in the discussion and in the appendix. Throughout we observe that the individual time series reveal very similar time trends. The only exception is the issue of attitudes towards Abortion Rights that saw a dynamic change in 2022. We have commented on it in a footnote where we refer the reader to the respective appendix.

- **Static versus dynamic affiliation**

The reviewer asked us to consider the sensitivity of the findings to using a time-varying coding of the country of affiliation of an academic and/or their institutional affiliation. In response, we have clarified our methodology to incorporate *time-varying institutional affiliations* using data from OpenAlex and yearly university rankings from QS World University Rankings. Specifically, we now extract the institution name from the author metadata associated with each publication by academics between 2016 and 2022, which allows us to capture changes in an academic’s affiliation and university ranking over time. This is not perfect but this allows us to capture changes in an academic’s affiliation over time, providing a more precise mapping of their geolocation and institutional context at the time each tweet was posted.

This impacts two analyses. For the analysis of country of affiliation—US versus non-US—we note that 87% of academics are consistently based at US or non-US institutions. However, when considering university rankings with our new 4-category system (1-100, 101-500, 501-1500, 1500+), we find that 50.3% of academics remain in the same category throughout the sample period. This higher mobility within university rankings is driven by both academics moving between institutions of different rankings and annual changes in the rankings themselves, which increases noise, particularly in the lower-ranked categories. Despite this, the analysis remains valuable as it captures the dynamic nature of academic institutions and the associated movement of academics, providing insights into patterns of academic expression over time.

Our main findings that academics from the US and those from top 100 institutions differ significantly from others still hold true. The refined analysis, however, provides additional

insights into the variability and non-linearity within the lower-ranked categories. These adjustments enhance the overall robustness and validity of our results.

Impact on main patterns and findings

To assess the impact of this adjustment, we repeated key analyses—specifically Figures 2-4 (and their appendix equivalents 4-6)—comparing results derived from static affiliation data (as of January 2023) with those using time-varying location data.. The overall results are very similar.

- **Comparability of balanced and unbalanced sample**

Reviewer 1 asked us to comment on the comparability of the balanced sample of academic users of Twitter with the unbalanced sample.

The reason to focus on a sample that we observe consistently over time is because, in addition to cross-sectional comparisons of academics across topics along salient dimensions such as gender or the country of academic affiliation, we want to make longitudinal comparisons. This is where sample selection that is done explicitly in a way to provide us with a longitudinal sample is relevant as this way we can ensure that the trends that are documented are not conflating compositional changes of the sample. We do expect that the balancing, on average, may actually attenuate the observed differences as we focus on a population of users that appears to be regular social media users.

To ensure the reader can understand better the sample differences we have added a new comparison table in the appendix between the balanced and unbalanced sample in the Appendix as Table A9.

Our balanced sample consists of approximately 99,274 academics who tweeted at least once in the first six months of 2016 and at least once in the last six months of 2022. This ensures a consistent presence across the entire study period. We compared this balanced sample with the unbalanced sample, which includes all individuals who tweeted at least once at any point during the period from 2016 to 2022.

The comparison reveals that the balanced and unbalanced samples are very similar in terms of non-Twitter metrics such as publication metrics (e.g., number of citations, number of works published), gender distribution, and discipline distribution. This similarity suggests that our balanced sample is representative of the broader academic population in these dimensions, at least on average.

- **Carry out formal statistical test on fit of power law**

Thank you for highlighting the need to rigorously substantiate the claim that the distributions of academic influence metrics follow a power-law. In response to your comment, we have now conducted a thorough statistical analysis following the methodology outlined by Clauset, Shalizi, and Newman (2009), as suggested.

This analysis involved fitting a power-law distribution to the data on content creation, engagement (likes received), public reach (followers), and citations among academics. The results, now included in the main body of our manuscript, confirm that these metrics do indeed follow a power-law distribution. Specifically, we calculated the power-law exponents (α) for each metric, with values ranging from 2.636 to 3.337, as detailed in Appendix Table A6. Additionally, the low Kolmogorov-Smirnov (KS) statistics obtained from our analysis further validate the goodness of fit for the power-law model across these metrics.

We have incorporated this additional validation into the manuscript, as described in the revised subsection on the "Distribution of Academic Influence on Twitter." This section now includes a brief overview of the statistical analysis and its results, with full details presented in Appendix Table A6. This ensures that our claims regarding the power-law nature of these distributions are well-supported by both visual and statistical evidence.

Response to reviewer 1: Text-mining, social media data analysis expert

Thank you for your thorough and insightful feedback. We have addressed each of your comments below in the order they were presented.

1. Quote: *“One feature of the work is its coverage of a large set of scholars and the selection of some important political topics. However, the analysis is limited to selected political topics only, which are only a tiny fraction (3.5%) of the tweets posted by the academics. Thus, the work might have limited immediate interest to the diverse audience across disciplines.”*

Response: Thanks a lot for raising this comment. This touches on two dimensions. First, the role that academics may play more broadly in societies as a potentially important vector of knowledge production and also increasingly as knowledge disseminators or translators. We do think that the latter role has grown in importance given the decline in professional journalism that has been widely documented. We further do perceive that academics are being listened to and that there is an often quite institutionalised role for academics.

Regarding the specific comment about the extent of political expression we offer some further clarifications.

Extent of Political Expression Among Academics: While it was observed that the analysis focuses on a selected set of topics around which we measure political expression, these topics constitute 3.5% of the total tweets by academics. This may not seem like a large number but it is worth emphasising the broader context of this finding. Our updated analysis, as reflected in Table 1 of the manuscript, shows that 75% of academics in our sample have engaged in political expression by making at least one non-neutral tweet on climate action, cultural liberalism, or economic collectivism during the study period. This significant level of engagement indicates that political discourse is not a negligible amount of academic activity on social media, rather than a marginal one. That is: politically salient expression is a near universal feature of academic expression online. Naturally, as the topic space may be expanded, the share increases. We focused on a subsample of topics and themes that social science research suggests as being particularly relevant during our sample period as they mark key socio-economic fault lines. We plan to highlight this point more clearly in the Introduction and Discussion sections to underscore the relevance and importance of our analysis.

Proportional Significance of Political Tweets: Turning to the number, as you noted, the 3.5% may initially seem like a small fraction. Yet, it represents a meaningful portion of the overall discourse when placed in context. To make this sharper we report on further analysis of academic social media expression. In a random subsample of academics, we have further analysed the social media content and classified the content using similar techniques. We observe that 16.70% of tweets are related to research, while 6.74% mention political figures. These two numbers provide a natural anchor: one fifth of the research related social media content can be attributed to political expression on the set of topics that we have

focused on. This comparison highlights that political expression is a substantial and non-negligible part of the academic conversation on Twitter. carried out further

2. Quotes: *“First, the work mentioned in the Introduction section that there is no work on the role of academics' online expressions in the perceptions of science. However, this question is far from being answered and rigorously analyzed in the current work. Instead, it presented a number of descriptive statistics and their variations along different dimensions, without any explorations/in-depth empirical analysis of the underlying causes and broader implications of these differences. Thus, the conceptual novelty of this work is also limited.”*

Response:

We appreciate the referee’s thoughtful feedback and understand the importance of highlighting the scope and limitations of the current study more clearly. We would like to emphasise that the focus of this paper is descriptive, aiming to present new patterns and raise pertinent questions about the online expression of academics rather than to establish causal relationships. By documenting these patterns across various dimensions—such as field, gender, and institutional ranking—we lay the groundwork for future studies that may engage in more targeted causal inquiries.

As such, this study seeks to provide the first large-scale empirical evidence of the variations in academic expression across disciplines and demographic factors, which is necessary before asking causal questions. We now acknowledge more explicitly in the Introduction and Discussion sections that the primary goal of this paper is descriptive, and causal analysis falls outside its current scope.

In the Discussion, we have expanded on the broader implications of these findings and posed new questions that emerge from our descriptive analysis. We believe that these questions will inspire further research into the social and academic factors that drive differences in online expression. We explicitly note that exploring the causes behind these observed trends, such as why egocentrism is on the rise, is a valuable next step for future research.

The following excerpt is added to the Discussion:

“Importantly, this study is descriptive, documenting trends without asserting causal relationships. We highlight differences in academic expression across subgroups but refrain from causal claims. Future research could investigate drivers behind these variations, such as institutional incentives or visibility, and explore how algorithms may amplify certain voices and shape public perception of science.”

The following is added to the Introduction:

“Our study provides a large-scale descriptive analysis of academics' political expression on Twitter across subgroups, laying the groundwork for future research to explore underlying causal factors, such as academics' motivations for political expression—whether driven by name recognition, ideological beliefs, or desire to disseminate knowledge.”

Additionally, we also contextualize our findings in the Discussion:

“Our findings echo concerns about ideological bias in social media, where progressive voices tend to dominate (Williams and Ceci, 2023). This aligns with Langbert’s (2018) analysis of political affiliations in academia, suggesting Twitter may amplify these biases. Studies like Crockett(2017) on digital moral outrage explain how online environments lower barriers for expressing outrage, increasing its frequency and intensity. This supports our findings that academic discourse on highly polarized topics, such as climate action, becomes more extreme and morally charged on social media. The incentives of platforms like Twitter, which reward emotionally charged content with greater visibility, may further exacerbate this polarization. Similarly, Ke, Ahn, and Sugimoto (2017) find that a small, highly active subset of social and computer scientists disproportionately shapes conversations on Twitter, paralleling our findings. Ferrara (2015) echoes concerns about algorithmic amplification, where non-expert academics with large followings may distort the perception of scientific consensus.”

References:

- Crockett, M.J., 2017. Moral outrage in the digital age. *Nature human behaviour*, 1(11), pp.769-771.
- Ferrara, E., 2015. " Manipulation and abuse on social media" by Emilio Ferrara with Ching-man Au Yeung as coordinator. *ACM SIGWEB Newsletter*, 2015(Spring), pp.1-9.
- Ke, Q., Ahn, Y.Y. and Sugimoto, C.R., 2017. A systematic identification and analysis of scientists on Twitter. *PLoS one*, 12(4), p.e0175368.
- Langbert, M., 2018. Homogenous: The political affiliations of elite liberal arts college faculty. *Academic questions*, 31(2).
- Williams, W.M. and Ceci, S.J., 2023. How politically motivated social media and lack of political diversity corrupt science. In *Ideological and Political Bias in Psychology: Nature, Scope, and Solutions* (pp. 357-375). Cham: Springer International Publishing.

3. Quote: *“Moreover, all the measurements are based on the posted topical tweets. But how likely is it that a user tweets that topic in the first place? Should we incorporate this likelihood into the calculation of the various measures? Under the current quantifications, it cannot distinguish users with different levels of activeness in posting political tweets, as it calculates the fraction among topical tweets.”*

Response:

We appreciate this insightful comment regarding the likelihood of an academic tweeting about a topic as a factor in the analysis. The manuscript includes over-time plots (Appendix Figure A21) that illustrate the average number of tweets per topic for each month from 2016 to 2022. This figure presents the average number of tweets per author by month, for each topic that has a stance (i.e., pro-, anti-, or neutral-). The category “Any Political” aggregates stances across the seven topics, reflecting the overall political stance of academics over time. These averages are based on a

sub-sample of topical tweets selected for stance detection, helping to clarify the volume of political expression and the extent to which individuals engage with politically salient topics. This ensures that our results are not overly influenced by a few very vocal and influential accounts that post frequently.

Figure A21: Topical tweets over time, average (2016-2022)

Our focus has been on ensuring that the comparisons made across subgroups—such as field or gender—are robust and reflect genuine differences in expression. In addition to the provided figures, we highlight that our approach is designed to normalise stance detection metrics across topical tweets, minimising the influence of individual activity levels. By normalising across topics, we ensure that even if an academic tweets more or less frequently about certain issues, the stance expressed within each topical category is given equal weight.

Additionally, as mentioned in our previous point, we conducted a within-user analysis that isolates individual author-level fixed effects, allowing us to control for individual differences in overall activity levels and topical engagement. This approach accounts for the likelihood of tweeting on a particular topic by focusing on temporal trends within each individual's posting behaviour. By partialling out these individual-level fixed effects, we ensure that our analysis reflects true within-user variation over time, rather than being skewed by differences in activeness or topic preferences across users. This method provides a more accurate picture of how trends in academic expression evolve, independent of individual tweeting frequency or topical engagement.

We have updated the Appendix section E.4 “*Robust Trends - Partialling Out Individual Fixed Effects*” that discusses this method, making clear that individual level fixed effects are able to capture individual level variation in topical engagement. Here is the updated excerpt:

“This approach is particularly advantageous for several reasons. First, it acknowledges the heterogeneity inherent in academic behaviors, recognizing that individual authors may exhibit varying levels of expression based on personal, disciplinary, or topical engagement factors. By controlling for individual author-level fixed effects, we isolate the temporal trends in political expression while accounting for these variations. Second, by controlling for these author-specific effects, we can more accurately isolate the influence of time on Expression, yielding insights that are more reflective of true temporal trends rather than artifacts of the composition of the author cohort.”

We believe these clarifications and the inclusion of new visual data provide a clearer understanding of how we address the likelihood of topical engagement, while maintaining the robustness of the stance detection and comparison across different groups of academics.

4. Quote: *“Second, sample representativeness remains unknown, which may confound the results. Although the study mentioned the sample selection process based on tweeting in the first and last 6 months in the focused period, there are insufficient details provided about the characteristics of this subset and the extent of representativeness of the broader academic population or even of the original 100k sample. This point is important, because scholars from certain fields may be more likely to have Twitter accounts, post more, and feature certain topics, all of which may affect the measurements and the generalizability of the findings.”*

Response:

The reason to focus on a sample that we observe consistently over time is because, in addition to cross-sectional comparisons of academics across topics along salient dimensions such as gender or the country of academic affiliation, we want to make longitudinal comparisons. This is where sample selection that is done explicitly in a way to provide us with a longitudinal sample is relevant as this way we can ensure that the trends that are documented are not conflating compositional changes of the sample. We do expect that the balancing, on average, may actually attenuate the observed differences as we focus on a population of users that appears to be regular social media users.

To ensure the reader can understand better the sample differences we have added a new comparison table in the appendix between the balanced and unbalanced sample in the Appendix as Table A9.

Our balanced sample consists of approximately 99,274 academics who tweeted at least once in the first six months of 2016 and at least once in the last six months of 2022. This ensures a consistent presence across the entire study period. We

compared this balanced sample with the unbalanced sample, which includes all individuals who tweeted at least once at any point during the period from 2016 to 2022.

The comparison reveals that the balanced and unbalanced samples are very similar in terms of non-Twitter metrics such as publication metrics (e.g., number of citations, number of works published), gender distribution, and discipline distribution. This similarity suggests that our balanced sample is representative of the broader academic population in these dimensions, at least on average.

However, and not surprisingly, the unbalanced sample reveals notable differences in the underlying Twitter metrics. This is entirely expected: we observe a disproportionately higher standard deviation in metrics such as the number of tweets, likes, and followers. This indicates that the unbalanced sample is noisier, with more extreme values that could introduce unnecessary variability into the analysis. By focusing on the balanced sample, we reduce this noise, resulting in a more reliable and stable dataset that is necessary to be able to ensure that the trends we are documenting are not masking compositional sample changes.

We have added this comparison to the Methods section of the manuscript, highlighting that while the balanced and unbalanced samples are comparable in non-Twitter metrics, the balanced sample is intentionally chosen to minimise noise and enhance the robustness of our results. We believe that this approach provides a more accurate reflection of academic behaviour on Twitter over time, without the distortions that could arise from extreme cases in the unbalanced sample.

Table: Balancedness Test: Comparison of Balanced and Unbalanced Populations (Table A9 reproduced)

Panel (a): Balanced Population						
Variable	Mean	Median	SD	Min	Max	N
Publication Metrics						
Nbr. Citations	1370.02	213	4523.99	1	238736	99274
Impact Factor (2Y)	16.93	9	38.56	0.02	4205	99274
Nbr. Works	57.58	24	114.38	1	7711	99274
Demographics						
Humanities	0.005	0	0.07	0	1	99274
STEM	0.291	0	0.45	0	1	99274
Social Sciences	0.098	0	0.30	0	1	99274
Gender: Male	0.60	1	0.49	0	1	99274
Twitter Metrics						
Nbr. Likes	3210.85	1673	5736.33	0	293797	99274
Nbr. Followers	1112.70	533	9145.94	0	1087504	99274
Nbr. Accounts Followed	749	529	1116.55	0	191923	99274
Nbr. Retweets	843.72	276	2117.91	0	181268	99274
Nbr. Posts	1575.24	1032	2052.13	4	117088	99274
Panel (b): Unbalanced Population						
Variable	Mean	Median	SD	Min	Max	N
Publication Metrics						
Nbr. Citations	1083.30	139	3969.42	1	331147	219273
Impact Factor (2Y)	15.49	9	30.96	0.02	2539	219273
Nbr. Works	48.26	18	110.51	1	8465	219273
Demographics						
Humanities	0.002	0	0.04	0	1	219273
STEM	0.262	0	0.44	0	1	219273
Social Sciences	0.071	0	0.26	0	1	219273
Gender: Male	0.59	1	0.49	0	1	219273
Twitter Metrics						
Nbr. Likes	5909.80	847	18705.81	0	2017914	219150
Nbr. Followers	2464.89	264	272177.95	0	126938095	219150
Nbr. Accounts Followed	713.74	328	2176.71	0	382058.5	219150
Nbr. Retweets	732.73	141	5519.14	0	1513703	219150
Nbr. Posts	3639.87	293	12652.17	1	734460	219150

5. Quote “Third, focusing on the interpretation and implications of the results, the paper lacks deep discussions about the domain significance and interpretation of the results. For example, the Discussion section is more of repeating the result section rather than a contextualized discussion. Another issue is how significant in terms of effect size is the stance difference between, e.g., humanities and STEM and others? Are they meaningful, considering that only 3.5% tweets are relevant (Table 3). Is there any confounder like the field composition of academics that explain the differences? Why is egocentrism on the rise? What are the factors that contribute to such a rise? Again, the study has not examined the underlying causes of the noted differences”

Response:

On Subgroup Differences and Methodological Robustness

To address the concern regarding the significance of subgroup differences (e.g., humanities vs. STEM), we conducted further analysis using a fixed effects regression model to control for individual-level heterogeneity over time. This allowed us to more rigorously assess whether the subgroup differences we observed in the descriptive analysis are statistically significant. The model we used is as follows:

$$Expression_{it} = \lambda_t \times Time_t + \mu_i + \epsilon_{it}$$

where $Expression_{it}$ represents the expression of political or behavioural identity by author i at time t . λ_t captures the effect of time (by year and month) on the expression, μ_i represents individual-level fixed effects, and ϵ_{it} is the error term.

This approach allows us to control for individual differences and isolate the true temporal trends in academic expression, ensuring that observed patterns are not artefacts of sample composition changes. By **partialling out individual fixed effects**, we ensure that the trends we identify reflect true changes over time rather than differences between individuals.

The results of this approach are illustrated in Appendix Figures A25, A26, and A27, which provide examples of within-individual trends for various subgroups (gender, field, and US vs. non-US academics). These figures, reproduced here for reference, demonstrate that the level differences in these trends remain consistent with our main findings. For example, Appendix Figure A25 shows that male academics exhibit more toxic expression than female academics, while there is no significant difference in emotionality or reasoning. Appendix Figure A26 highlights that humanities scholars are more egocentric, toxic, and display higher emotionality compared to their counterparts in social sciences and STEM fields. Similarly, Appendix Figure A27 illustrates that US-based academics show greater support for techno-optimism and cultural liberalism, especially after 2020. These examples confirm that the trends observed in our main analysis are robust even when accounting for within-individual variation.

Figure A25: Academic Expression Over Time by Gender, Within-user estimates

Figure A26: Academic Expression Over Time by Fields, Within-user estimates

Figure A27: Academic Expression Over Time by whether academic's institution is US based, Within-user estimates

These consistent within-user trends improve validity of the subgroup differences we report and allay concerns about potential confounders related to sample composition.

Furthermore, to ensure that the observed subgroup differences in stances are not confounded by variations in tone (e.g., emotionality, egocentrism, toxicity), we ran additional regressions controlling for these variables. The results, shown in Appendix Figures A15 and A16 (these two figures will be presented in detail in response #9 below), confirm that

subgroup differences (e.g., between STEM and humanities academics on climate action) remain statistically significant even after controlling for tone. For instance, the coefficient for being in STEM is statistically distinguishable from the coefficient for being in social sciences or humanities at the 95% confidence level when the dependent variable is stance on climate action.

On the 3.5% of Relevant Tweets

We acknowledge that the 3.5% share of politically salient tweets is relatively small, as the referee noted. However, as detailed in our previous response (#1), political expression remains a significant dimension of academic discourse. Notably, 75% of academics in our sample have posted at least one non-neutral tweet on topics such as climate action, cultural liberalism, or economic collectivism, signalling that political expression is a nearly universal feature of academic engagement, even if it constitutes a smaller portion of total tweets. Additionally, when compared to other categories—such as the 16.7% of tweets related to research and 6.74% that mention political figures—the 3.5% figure represents 20.9% of the volume of tweets about research. This underscores that political discourse, though proportionally smaller, plays a crucial role in academic conversations on Twitter. We have clarified this point in the main text to reinforce the significance of political discourse in our analysis.

6. Quote: *“There are numerous methodological concerns that remain to be addressed. 1. The study extensively leveraged LLM for measurements (stance detection, gender inference, etc.). However, the accuracy of LLM results needs to be checked. Although the stance detection performance was tested on the SemEval tasks, they are different from the actual focused tasks (climate action, etc.). Similarly, the accuracy of first name-based gender inference using GPT is unknown, while there are many existing studies on gender in science that can be used for cross-checks.”*

Response

We appreciate your suggestion to thoroughly validate our use of GPT-based models for stance detection, topic detection, and gender inference. To ensure the accuracy and reliability of these measurements, we conducted a series of validations across different datasets and tasks.

Stance Detection Validation

As you point out, our initial validation of the GPT-3.5-turbo model for stance detection used the ACL SemEval-2016 Task 6 dataset, which includes tweets labelled for stance (pro, anti, neutral) across a diverse set of topics. The results showed strong performance, achieving average F1 scores ranging from 84 to 92, closely aligning with human annotations.

To further test the generalizability of our approach, we compared the performance of GPT-3.5-turbo with GPT-4 on the same SemEval dataset. The results were consistent across both models, reinforcing our confidence in the external validity of GPT-3.5-turbo and its reliability for stance detection. The results are presented in Appendix Table A1 of the updated manuscript, reproduced here for ease of reference.

Table A1: Evaluation Metrics for Stance Detection

Task	Target	GPT 3.5 Turbo (F_{avg})	GPT 4 (F_{avg})
A	Feminism	92.44	81.89
A	Hillary Clinton	89.57	87.53
A	Abortion	79.52	84.36
B	Donald Trump	84.18	80.00

Notes: Table shows results of validation of stance detection step. We obtain human labels for stance detection task from ACM SemEval-2016 Task 6 (Mohammad et al. 2016). Humans labelled tweets are pro-, anti- and neutral-, on topics ranging from Abortion Rights to Donald Trump. The stance detection’s effectiveness was validated against 40,317 hand-coded labels from 137 humans, yielding F-scores of 79.52 to 92.44 for GPT-3.5 Turbo and 80.00 to 87.53 for GPT-4, which are considered very high for classification tasks.

Topic specificity

You also raised the concern about the suitability of the classifiers for different types of topics. To address this, we also conducted an additional analysis using a new subset of tweets to expand this validation to the specific topics in our study. This analysis involved labelling 400 random tweets per topic per stance from our main analysis sample, totaling 8,400 tweets. The results showed strong consistency across models, particularly as model quality increased, confirming the robustness of our stance detection approach. Specifically, these were labelled over 10 iterations by four models: GPT-3.5-turbo, GPT-4, GPT-4o, and GPT-4o-mini, leading to 336,000 predictions.

After filtering out tweets labelled as unrelated by any model, we retained 63,210 predictions per model (75.25% of the original set). We measured the agreement rate and F1 scores across these models, finding strong consistency, particularly as model quality increased. For instance, comparisons involving GPT-4o, the most advanced model, yielded the highest agreement rates and F1 scores, confirming the robustness of our stance detection approach. The results are presented in Appendix Table A2 of the updated manuscript, reproduced here for ease of reference.

Table A2: Comparison of Agreement and F1 Scores Across GPT Models

Comparison	Agreement (Modal)	Agreement (Iterations)	F1 (Modal)	F1 (Iterations)
GPT-3.5-turbo vs GPT-4o	0.781	0.772	0.806	0.795
GPT-3.5-turbo vs GPT-4	0.750	0.738	0.772	0.756
GPT-3.5-turbo vs GPT-4o-mini	0.684	0.681	0.696	0.691

Notes: Table presents a comparison of agreement rates and average F1 scores (F_{avg}) between different GPT models for stance detection. **Agreement (Modal)** refers to the proportion of identical stance predictions when considering the modal stance across 10 iterations per tweet. **Agreement (Iterations)** measures this proportion for each individual iteration treated as a unique instance. **F1 (Modal)** represents the average F1 score when one model’s modal stance is used as the "true" label to evaluate another model’s performance. Similarly, **F1 (Iterations)** does the same but treats each iteration as a separate instance. Higher values in these metrics indicate stronger consistency and accuracy in stance classification.

Topic Detection Validation

For topic detection, we validated our methodology using the SemEval-2016 dataset. We generated topic-specific dictionaries using GPT-4 and applied them to the SemEval dataset. The results showed high recall and balanced F1 scores across

topics, demonstrating the effectiveness of our topic detection method. The results are presented in Appendix Table A3 of the updated manuscript, reproduced here for ease of reference. We will discuss these in more detail in to point #10 below.

Table A3: Evaluation Metrics for Topic Detection Validation

Topic	Precision	Recall	F1-Score
Legalization of Abortion	86.51	66.00	74.87
Hillary Clinton	75.49	60.79	67.35
Feminist Movement	51.00	97.67	67.01
Donald Trump	79.41	65.59	71.84
Overall	65.47	74.32	69.62

Notes: Table shows precision, recall, and F1-Score metrics for topic detection. The results highlight the effectiveness of the topic dictionaries generated using GPT-4 and the robustness of the methodology. Tweets were matched against the dictionaries, and true and false positives and negatives were calculated. For instance, a true positive (TP) occurs when a tweet labeled as "Legalization of Abortion" contains a term from the corresponding dictionary, while a true negative (TN) occurs when no terms from the dictionary match tweets labeled for other topics. The high recall demonstrates the comprehensive nature of the dictionaries, which capture the evolving lexicon over the years, while the precision shows the effectiveness of filtering out unrelated tweets. This approach ensures accurate topic detection across diverse tweet datasets by leveraging context-aware keyword dictionaries and refining selections to enhance accuracy.

Gender Classification Validation

You raised the concern about the accuracy of the gender classification. To validate gender inference based on first names, we utilised a comprehensive dataset of 147,269 unique names from official sources. The GPT-3.5-turbo model was used to predict the gender associated with each name. The results showed high precision (0.9866), recall (0.9871), and F1 scores (0.9868) when weighted by name frequency, demonstrating the robustness of our gender classification method. The results are presented in Appendix Table A4 of the updated manuscript, reproduced here for ease of reference.

Table A4: Evaluation Metrics for Gender Classification

Metric	Unweighted	Weighted by Count
Precision	0.8097	0.9866
Recall	0.8202	0.9871
F1 Score	0.8149	0.9868
Accuracy	0.8610	0.9863

Notes: Validation results for the gender classification method using a dataset of 147,269 unique names from authoritative sources, including US Social Security Card Applications (1880-2019), UK Baby Names (2011-2018), British Columbia's Baby Names (1918-2018), and Australia's Popular Baby Names (1944-2019). The GPT-3.5-turbo model classified names as Male, Female, or Unclear. The table presents both unweighted and weighted metrics: Precision, Recall, F1 Score, and Accuracy. Unweighted metrics show high overall performance, treating all names equally. Weighted metrics, which give more importance to frequently occurring names, yield even higher scores, particularly for names more likely expected to appear in our dataset. This demonstrates the method's strong accuracy and robustness, especially for common names.

We believe that these additional validations across stance detection, topic detection, and gender inference underscore the reliability and accuracy of the GPT-based

models employed in our study. By ensuring that these models perform well across different datasets and tasks, we provide a strong foundation for the study's findings.

7. Quote: “2. *This accuracy issue is particularly important, as many major measurements like Cultural Liberalism are based on several separate tasks (Abortion Rights, Racial Equality, etc., Table 3). Therefore, it needs to reassure that the individual errors are not added up in the composite measures.*”

Response

You raised a point regarding the importance of accuracy in our composite measures, particularly for Cultural Liberalism and Economic Collectivism. We understand your concern that individual errors in the component tasks could accumulate and affect the overall composite measures. We opted for the aggregation into an index for ease of presentation to reduce the dimensionality of the empirical observations into specific themes. Yet, we agree that this may raise the question about whether the underlying latent trends are an artefact of the aggregation. Of course, we have checked these subcomponents before, but for increased transparency, we have added these as new supplementary findings.

To alleviate these concerns, we conducted a detailed analysis to examine the trends within each composite measure and its respective components. We present these in the figure plotted below and comment on the key observations. :

Time series plots reveal similar trends breaking down the composite measures into their constituent parts; we found that most subcomponents exhibit broadly similar trends, suggesting that individual component variations do not significantly skew the overall measure.

The exception is the change in attitudes towards abortion Rights'. Yet, the notable increase in tweets related to Abortion Rights in 2022 did not substantially alter the overall trend for Cultural Liberalism, which continued to decline. This is because the number of tweets related to Abortion Rights is relatively small compared to those related to Immigration or Racial Equality issues. This point was made in Table 3 (Now Table 2), the number of tweets related to Abortion Rights is roughly one-tenth of those related to Immigration or Racial Equality. Therefore, the influence of this spike on the composite measure is proportionally small.

These findings provide reassurance that the composite measures are robust and not unduly influenced by idiosyncratic variation in the subcomponents. We have included these time series plots in the manuscript appendix and provide them below for convenience, to further validate our approach and ensure transparency in how these composite measures are constructed. We have added a footnote to this effect in the main paper with a link to the supplementary appendix where we explicitly reference these observations.

8. Quote: “3. *Small samples (3 English tweets per topic monthly for stance detection and 10 tweets per year for toxicity measurement) raise concerns about potential uncertainty that casts doubts on whether the between-group variations are meaningful.*”

Response

Thank you for raising the concern regarding the potential measurement error that may be introduced by our sampling approach for stance detection and toxicity measurement. We recognize the importance of ensuring that between-group variations in our analysis are meaningful and not artefacts of small sample sizes. Our sampling strategy was specifically designed to ensure maximal between-group comparability by preventing the results from being skewed by highly prolific accounts. Specifically, we focus on creating a balanced panel to ensure our results are not overly influenced by a few very vocal and influential accounts that post frequently.

We now add a further comment on the sampling strategy in the revised draft. There we now emphasise that the sampling approach—capping at three tweets per author

per topic per month—is a deliberate balancing criterion aimed at creating a balanced panel. By focusing on academic-time as the unit of analysis rather than the individual tweet, we prevent our dataset from being skewed by a small number of academics who might tweet excessively. Additionally, our stance detection metric is normalised across all tweets for a given topic, which means that no single tweet or user has an outsized influence on the stance measurements. This ensures that our comparisons between groups remain robust and meaningful.

Specifically, 99.38% of cases involve just one tweet per author per topic per month, with only 0.62% involving two tweets and an even smaller fraction (0.006%) involving three tweets. This means that increasing the cap would affect less than 0.006% of the observations, making any potential impact on our results negligible. We further add more clarification in a new footnote in the Methods section that provides a reference of the minimal impact of this sampling cap, reproduced here for reference:

“Although we cap the sample at three tweets per author per topic per month for stance detection, 99.38% of cases involve just one tweet, with only 0.62% involving two and 0.006% involving three. Thus, increasing the cap would affect less than 0.006% of the observations, making any impact on results negligible.”

We hope this clarification addresses your concerns, and we appreciate your feedback, which has led us to further elucidate this aspect of our methodology in the manuscript.

9. Quote: *“4. It is argued that the ‘how’ of what is said might offer a signal for opinions. But the measurement of emotionality-reasoning ratio is independent of the ‘what’ aspect. This may be a confounding factor behind the observed variations of emotionality-reasoning.”*

Response

Thank you for this insightful comment. We appreciate your suggestion regarding the potential role of “how” things are said—such as tone, emotionality, egocentrism, or toxicity—possibly influencing the signal conveyed by the “what” aspect of the message. This concern highlights the possible confounding effect of the “how” on the observed differences in opinions across subgroups, and it is worth exploring further. To address this concern, we performed two key analyses:

Raw Correlations Between Stances and “How” Expressions

First, we examined the raw correlations between each stance (the “what”) and three key measures of expression style (the “how”)—emotionality, egocentrism, and toxicity. The results are presented in the table below:

Table A7: Raw Correlations Between Stances and Emotionality, Egocentrism, and Toxicity

Stance	Emotionality	Egocentrism	Toxicity
Climate Action	-0.0081 p = 0.0017 [-0.0131, -0.0030]	-0.0272 p < 0.0001 [-0.0323, -0.0222]	-0.0326 p < 0.0001 [-0.0377, -0.0276]
Techno-Optimism	-0.0166 p < 0.0001 [-0.0217, -0.0116]	-0.0407 p < 0.0001 [-0.0458, -0.0357]	-0.0553 p < 0.0001 [-0.0604, -0.0503]
Behavioural Adjustment	-0.0075 p = 0.0035 [-0.0125, -0.0025]	0.0003 p = 0.9128 [-0.0047, 0.0053]	-0.0135 p < 0.0001 [-0.0186, -0.0085]
Cultural Liberalism	0.0209 p < 0.0001 [0.0159, 0.0259]	0.0568 p < 0.0001 [0.0518, 0.0618]	0.0854 p < 0.0001 [0.0804, 0.0904]
Economic Collectivism	0.0035 p = 0.1727 [-0.0015, 0.0085]	0.0149 p < 0.0001 [0.0099, 0.0200]	0.0002 p = 0.9267 [-0.0048, 0.0053]

Note: This table presents the raw correlations, *p*-values, and 95% confidence intervals (CIs) between different political stances and three key variables related to how academics express themselves: **Emotionality**, **Egocentrism**, and **Toxicity**. These variables represent the tone and style of expression, with emotionality capturing the ratio of affective to cognitive words, egocentrism measuring the prevalence of self-referential language, and toxicity capturing the likelihood of harmful or aggressive language. The stances—Climate Action, Techno-Optimism, Behavioural Adjustment, Cultural Liberalism, and Economic Collectivism—are defined as the net stance per author on each topic. The raw correlations are all near zero, suggesting minimal direct relationship between stance and expression style, with the highest correlation being 0.085 (*p* < 0.0001, 95% CI [0.0804, 0.0904]) for Cultural Liberalism and Toxicity. This indicates that "how" things are said does not heavily influence the political stance expressed by academics.

These correlations show that the relationship between stance and expression style is minimal in most cases. For example, the correlation between emotionality and stance is very close to zero, with the largest observed correlation being 0.085 between cultural liberalism and toxicity. This suggests that, in general, expression style does not have a strong raw correlation with stance.

Regression Analysis to Control for “How” Expressions While Examining Subgroup Differences

In the second step, we explored whether the observed differences across subgroups could be confounded by expression style. To do this, we ran a series of regressions for each stance, comparing subgroup differences (e.g., gender, field, rankings, and country) while controlling for one of the three “how” expressions at a time (emotionality, egocentrism, or toxicity). Specifically, we estimate the following model:

$$\text{Stance}_{it} = \alpha + \beta_1 * \text{GroupVar}_{it} + \beta_2 * \text{HowExpression}_{it} + \varepsilon_{it}$$

Where *GroupVar_{it}* represents a subgroup indicator (e.g., male, STEM, US), and *HowExpression_{it}* represents one of the three expression styles (emotionality, egocentrism, or toxicity). This allows us to assess whether controlling for the “how” variables alters the subgroup coefficient β_1 . If β_1 remains stable across the three models, it indicates that tone and style of expression do not confound subgroup differences.

Climate-related stances: We begin by presenting the subgroup differences for climate-related stances (Climate Action, Techno-Optimism, and Behavioural

Adjustment), as originally shown in the paper (Figure 2). We then display the updated results after controlling for emotionality, egocentrism, and toxicity.

Figure: Subgroup Comparisons for Climate-Related Stances (Figure 2 Reproduced)

Figure: Subgroup Comparisons for Climate-Related Stances, Controlling for Emotionality, Egocentrism, and Toxicity (Appendix Figure A15 reproduced)

As seen in the new figure, the coefficients for subgroup differences remain consistent across the three regressions, each controlling for one of the expression styles. This suggests that the "how" expressions—whether emotionality, egocentrism, or toxicity—do not confound the subgroup differences for these climate-related stances.

Socio-economic stances: Next, we replicate this analysis for the stances on Cultural Liberalism and Economic Collectivism. Again, we start by presenting the subgroup differences from the original analysis (Figure 3) and then show the updated results with controls for emotionality, egocentrism, and toxicity.

Figure: Subgroup Comparisons for Economic and Cultural Stances (Figure 3 Reproduced)

Figure: Subgroup Comparisons for Economic and Cultural Stances, Controlling for Emotionality, Egocentrism, and Toxicity (Appendix Figure A16 reproduced)

Again, we observe that the coefficients for subgroup indicators remain stable across all three regressions, indicating that controlling for the “how” expressions does not significantly alter the subgroup effects. Thus, it is unlikely that emotionality, egocentrism, or toxicity confound the observed differences across subgroups for these stances.

Based on our results, we find that expression style (the “how”)—whether emotionality, egocentrism, or toxicity—does not confound the observed subgroup differences in stance (the “what”). The coefficients for subgroup differences remain consistent regardless of whether we control for these expression styles, indicating that the variations we observe are primarily driven by the content of the statements rather than how they are expressed.

We have included the raw correlation table in Appendix Table A7, and the two additional figures in Appendix Figures A15 and A16. These results are discussed in detail in Appendix Subsection "Robustness of Subgroup Differences in Political Stances After Controlling for Tone and Style Variables." We hope this additional analysis clarifies and strengthens the interpretation of our findings.

10. Quote: *"5. How accurate is the generated list of keyphrases? While keywords based filtering has a high precision, it suffers from a low recall. Also, it would be informative to provide the list of keyphrases."*

Response

Thanks a lot for your feedback and suggestions here. We have taken this suggestion on board to provide some more evidence on the suitability of the key phrase approach for topic detection, which we have documented in the newly added appendix section "Validation of Topic Detection." Specifically, we utilised the SemEval-2016 Task 6 dataset as a benchmark, which provides a set of tweets labelled for specific topics, including Feminist Movement, Hillary Clinton, Legalisation of Abortion, and Donald Trump. This dataset serves as a reliable ground truth for evaluating the effectiveness of our GPT-4-generated topic dictionaries.

Our approach involved generating topic-specific dictionaries through GPT-4, leveraging a diverse set of prompts that accounted for various ngrams, temporal contexts (2016-2022), and both general and Twitter-specific vernaculars. This comprehensive method ensured that our dictionaries captured the evolving and dynamic lexicon pertinent to each topic, which is particularly crucial on platforms like Twitter where language usage rapidly changes.

We then applied these dictionaries to the SemEval dataset to evaluate their performance. For each topic, we calculated standard evaluation metrics, including true positives (TP), true negatives (TN), false positives (FP), and false negatives (FN). The results, as presented in Table A3: "Evaluation Metrics for Topic Detection Validation," **provided here for ease of reference**, indicate that our methodology achieves high recall across all topics, demonstrating the effectiveness of our comprehensive dictionary generation process in capturing a wide range of relevant tweets.

Table A3: Evaluation Metrics for Topic Detection Validation

Topic	Precision	Recall	F1-Score
Legalization of Abortion	86.51	66.00	74.87
Hillary Clinton	75.49	60.79	67.35
Feminist Movement	51.00	97.67	67.01
Donald Trump	79.41	65.59	71.84
Overall	65.47	74.32	69.62

Notes: Table shows precision, recall, and F1-Score metrics for topic detection. The results highlight the effectiveness of the topic dictionaries generated using GPT-4 and the robustness of the methodology. Tweets were matched against the dictionaries, and true and false positives and negatives were calculated. For instance, a true positive (TP) occurs when a tweet labeled as "Legalization of Abortion" contains a term from the corresponding dictionary, while a true negative (TN) occurs when no terms from the dictionary match tweets labeled for other topics. The high recall demonstrates the comprehensive nature of the dictionaries, which capture the evolving lexicon over the years, while the precision shows the effectiveness of filtering out unrelated tweets. This approach ensures accurate topic detection across diverse tweet datasets by leveraging context-aware keyword dictionaries and refining selections to enhance accuracy.

For example, the "Legalisation of Abortion" topic achieved a precision of 86.51% and a recall of 66.00%, resulting in an F1-Score of 74.87. Similarly, the "Donald Trump" topic achieved a precision of 79.41% and a recall of 65.59%, with an F1-Score of 71.84. These results underscore the robustness of our approach, particularly in ensuring high recall, which is critical for capturing the broad spectrum of relevant discourse. The union of terms across different years and ngram combinations allowed us to maintain a high recall while ensuring the precision remained sufficiently strong to filter out unrelated content effectively.

While the specific topics in the SemEval dataset differ from those used in our main study (with the exception of "Legalisation of Abortion"), the consistent performance across multiple topics enhances our confidence in the external validity of our methodology when applied to other topics in our research. This validation also supports the scalability and cost-efficiency of our approach, which allows for the accurate detection of topics in large tweet datasets without the need to parse each individual tweet through the language model.

Additionally, we have added an appendix table showing 20 randomly sampled ngrams per topic (Appendix Table A8, **reproduced here for reference**), and we also provide the full list of terms for each topic in the replication package as a topic-by-term list. This ensures transparency and allows for further scrutiny of the generated dictionaries.

Table: Random Sample of Terms for Topics Detected using GPT-4 (Appendix Table A8 reproduced)

Topic	Terms
Abortion Rights	roe v. wade 2021 debate, #abortionaccess, #abortionlaw, my body my, #prolife, abortion restriction laws, stop the bans, justice for abortion rights, georgia abortion ban, repeal hyde amendment, abortion access in 2017, #stopthebans, partial birth, texas abortion bill, telemedicine for abortion, abortion, abortion ban laws, #stand-withpp, gestational age limits, pro life anti abortion
Climate Action	#netzero, climate change and agriculture, climate change mitigation, united nations climate, carbon footprint, climate change 2013, earth temperatures, protect our planet, scientific consensus on climate, global climate change, climate change convention, stop global warming now, #cop21 paris, paris climate agreement 2015, greenhouse gas emission reduction, climate action summit 2019, carbon capture storage, trump climate change, climate change adaptation, global average temperature
Immigration	border security and enforcement, stricter immigration laws, bipartisan framework for comprehensive immigration, customs and border, scrapping the diversity lottery, #deportationforce, asylum seeking policies, travel ban supreme court, #refugeeasylum, secure border initiative, legal status pathway, us immigration policy, fight against deportation, ban on muslim countries, support immigration reform, family detention centers, 2017 sanctuary cities controversy, comprehensive immigration reform 2022, abolish ice movement, migrant detention
Racism or Race Relations	ethnic minorities, race-related hate crimes, reparations for african americans, #charlestonshooting, racism in sports 2008, stand against racism 2022, hate-crime, racial identity, #racismisavirus, racial inequality awareness, protests against racial injustice, black lives matter, racial inequality in 2007, blacklivesmatter, racial profiling in police, address racial disparities, racial equality now, color should not matter, #whitesupremacy, say no to racism
Redistribution of Income	wealth tax proposals, redistribution of income, taxation for redistribution, economic inequality issues, wealth redistribution policies, income redistribution efforts, redistribution debate, income redistribution policies, redistribution, progressive income redistribution, income gap widening, tax on wealth, #progressivetax for fairness, income redistribution and economy, progressive tax system, income redistribution mechanisms, economic redistribution strategies, basic income, #redistribution, top1percent
Tax Policy	cryptocurrency tax rules, #irs, digital tax policy, tax rate adjustments, obama tax policy, tax code simplification, tax cuts, #obamatax, tax bracket changes, small business tax, tax policy under obama, tax policy review, trump tax cuts, income tax rates, tax legislation, tax avoidance, impact of taxes, tax policy debates, #taxes, property tax
Welfare State	public housing assistance programs, welfare state critique, #socialsecurity, public pension scheme reform, welfare budget, income redistribution policy, social expend, public healthcare, social welfare policies, food stamps, welfare rights, welfare state issues, food stamp program 2013, reform the welfare state, benefit system, unemployment benefits policy 2018, welfare reform policies, economic impact welfare state, #publicwelfare, welfare state development

By incorporating this validation step, we have demonstrated that our topic detection method not only balances precision and recall effectively but also provides a reliable foundation for subsequent stance detection and analysis. The detailed metrics provided in Table A2 should reassure that the generated keyphrases are both accurate and comprehensive, addressing the initial concern about the potential limitations of keyword-based filtering.

11. Quote: “6. User geolocation is based on their most recent affiliation recorded in OpenAlex. But wouldn't it make more sense to use the “current” affiliation when the tweet was posted?”

Response:

We appreciate this insightful suggestion regarding the use of time-varying affiliation data for geolocation, rather than relying on the most recent affiliation recorded in OpenAlex. We did have a look at this in the preparation of the manuscript. We agree that this adjustment provides a more accurate reflection of the academic's location at the time of each tweet, which could influence the analysis of their political expression and related behaviours.

In response, we have updated our methodology to incorporate *time-varying institutional affiliations* using data from OpenAlex and yearly university rankings using data from QS World University Rankings. Specifically, we now extract the institution name from the author metadata associated with each publication by academics between 2016 and 2022. This approach allows us to provide a more precise mapping of their geolocation and university ranking at the year each tweet was posted. The methodological adjustments are outlined in detail in the updated Methods/Data section of our manuscript:

“We accessed all publications by academics within this period through OpenAlex and extracted the institution name from the author metadata for each publication. By relying on article-level affiliation data, we ensure that our analysis reflects changes in an author's affiliation over time, rather than a single, static affiliation. We use yearly aggregation based on the modal affiliation of each academic to maintain consistency.”

Moreover, to handle cases where an academic did not publish in certain years, we generated a complete set of institution-year pairs by carrying forward the nearest available ranking and backfilling as necessary to ensure continuity

“In cases where an academic did not publish in certain years, we generated a complete set of institution-year pairs, filling any gaps by carrying forward the nearest available ranking and backfilling as necessary to ensure continuity.”

This refinement is particularly significant when analysing results based on categorical data, such as whether an academic is affiliated with a top 100 university, ranked 101-500, 501-1500, or 1500+, and whether they are based in the US or elsewhere. We chose to introduce the 4-category system for university rankings, rather than simply using a top 100 or not, by keeping the top 100 category intact and disaggregating the 100+ category into three distinct groups. This allows us to explore whether a monotonic, linear, or non-linear pattern holds across these different tiers, providing deeper insights into how institutional ranking is associated with academic behaviour and expression. This disaggregation is particularly useful as it allows us to examine variations within the lower-ranked institutions, which are often more diverse and may exhibit different patterns of behaviour.

We recognize that this adjustment may impact results, especially for academics who transition between these categories during the study period. To quantify the stability and movement across these categories, we first analysed the proportion of academics remaining within each ranking category from 2016 to 2022.

For the country of affiliation—US versus non-US—we note that the vast majority, 87% of academics, are consistently based at US or non-US institutions. However, for university rankings, with our new 4-category system (1-100, 101-500, 501-1500, 1500+), we find that 50.3% of academics remain in the same category from the beginning to the end of the sample period. This change reflects a higher degree of mobility within the university ranking categories, driven by both academics moving between universities of different rankings and the rankings themselves changing every year. This increases noise, particularly in the lower-ranked categories, where institutions are more likely to fluctuate. Although the academic rank of an institution may vary—especially outside the very top—this analysis is still valuable as it allows us to capture the dynamic nature of academic institutions and the associated movement of academics across institutions, which in turn helps us understand patterns in academic expression over time.

Impact on main patterns and findings

To assess the impact of this adjustment, we repeated key analyses—specifically Figures 2-4 (and their appendix equivalents 4-6)—using the new time-varying location and ranking data. The overall results for the US versus non-US classification remain consistent with our previous analysis that used time-invariant geolocation. In general, we observe a monotonic pattern from 1-100 to 101-500 to 501-1500, with the 1500+ category being particularly noisy and showing some degree of nonlinearity. This additional result further enriches our analysis, as it highlights the complexity of academic behaviour in relation to institutional rank, particularly in the lower tiers.

Our main claim that top 100 institutions differ significantly from others generally holds. This refined analysis not only confirms our original findings but also provides new insights into the variability and non-linearity present in the lower-ranked categories. These variations align with the idea that these categories encompass a diverse range of academic institutions, thereby enhancing the robustness and validity of our findings.

We have included excerpts from Figures 2-4 in the referee report to illustrate these comparisons, demonstrating how political expression, tone, and style of expression vary with time-varying affiliation data. This adjustment ensures that our analysis more accurately reflects the dynamic nature of academic affiliations and rankings, thereby enhancing the robustness and validity of our findings.

12. Quote: “The readability of the manuscript needs to be improved substantially, considering the intended general readership of the journal. For instance, the Introduction section may be too long, where the result discussion part could be moved to the Discussion section and the road map may not be necessary. The methods and results sections are dense. The results should be structured in a way that logically builds from the data and methods, allowing the reader to easily trace how the key findings were derived. Many terms like economic collectivism are better explained immediately after being mentioned. The results description parts should clearly indicate which panel in a figure is being referred to. By strengthening the coherence between the data, methods, and findings, the paper would be significantly easier for readers to follow and understand the full scope of the research.”

Response:

Thanks a lot for this suggestion. We have streamlined the introduction significantly, moving parts of the results discussion into the appropriate discussion section, and adjusted the flow of the methods and results sections to better align with the data construction. We have also incorporated your suggestions regarding the figures. While we do not provide a line-by-line breakdown of the changes, we have incorporated these adjustments in our letter to the editor to reflect the overall improvements in the paper’s structure and clarity.

13. Quote: “Minor comments:

L147: To claim the distributions follow a power-law, there is a rigorous statistical test procedure to follow [Clauset, A., Shalizi, C. R., & Newman, M. E. (2009). Power-law distributions in empirical data. SIAM review, 51(4), 661-703.]”

Response:

Thank you for highlighting the need to rigorously substantiate the claim that the distributions of academic influence metrics follow a power-law. In response to your comment, we have now conducted a thorough statistical analysis following the methodology outlined by Clauset, Shalizi, and Newman (2009), as suggested.

This analysis involved fitting a power-law distribution to the data on content creation, engagement (likes received), public reach (followers), and citations among academics. The results, now included in the main body of our manuscript, confirm that these metrics do indeed follow a power-law distribution. Specifically, we calculated the power-law exponents (α) for each metric, with values ranging from 2.636 (95% CI [2.625, 2.648]) to 3.337 (95% CI [3.323, 3.352]), as detailed in Appendix Table A6, **reproduced here for reference**. Additionally, the low Kolmogorov-Smirnov (KS) statistics obtained from our analysis further validate the goodness of fit for the power-law model across these metrics.

We have incorporated this additional validation into the manuscript, as described in the revised subsection on the "Distribution of Academic Influence on Twitter." This section now includes a brief overview of the statistical analysis and its results, with full details presented in Appendix Table A6. This ensures that our claims regarding the power-law nature of these distributions are well-supported by both visual and statistical evidence.

Table: Power-Law Fit Parameters for Academic Influence Metrics (Table A6 reproduced)

Metric	Power-Law Exponent (α)	Minimum Value (x_{\min})	KS Statistic	Log-Likelihood Ratio
Content Creation (Posts)	3.337	2650	0.007	-1.112 (p = 0.399)
Engagement (Likes Received)	3.146	15491.5	0.015	-3.660 (p = 0.112)
Public Reach (Followers)	2.636	1332	0.011	0.403 (p = 0.001)
Citations Count	2.929	15808	0.033	-12.281 (p = 0.003)

Notes: This table summarizes the results of fitting a power-law distribution to various metrics of academic influence on Twitter: content creation (posts), engagement (likes received), public reach (followers), and citations count. The analysis follows the method by Clauset et al. (2009), which estimates the power-law exponent (α) and the minimum value (x_{\min}) above which the data follows a power-law distribution. The KS statistic measures the goodness of fit, where lower values indicate a better fit to the power-law model. The log-likelihood ratio compares the fit of the power-law model to that of an alternative distribution, with the p-value indicating the statistical significance of the comparison. The results confirm that these metrics exhibit power-law behavior, indicating significant inequality across these academic influence measures.

14. Quote: “L229: Results about Twitter reach and expertise are not discussed in the main text.”

Response:

We updated the relevant area (L139-146) and few more (L167-173, L313-319, and L325-329) to include these results and discussion. Here are some relevant excerpts:

L139-146 (Results) reads

“Academics with higher social media reach but lacking expertise in the subject matter are less supportive of climate action compared to high-reach experts ($t(13, 504) = -18.17, p < 0.001, \text{Cohen's } d = 0.22, 95\% \text{ CI } [0.20, 0.24]$). Low-reach experts display significantly higher support than high-reach non-experts ($t(8, 147) = -19.83, p < 0.001, \text{Cohen's } d = 0.32, 95\% \text{ CI } [0.30, 0.34]$). Low-credibility, high-reach academics are 8.62 times more expressive about climate action than the average U.S. user, whereas low-reach subject matter experts are 19.2 times more expressive. This raises concerns that science influencers may skew perceptions on topics where they lack expertise.”

L167-173 (Results) reads

“Academics with higher social media reach but non-expertise exhibit less progressive stances on cultural liberalism compared to experts ($t(18, 499) = 5.15, p < 0.001, \text{Cohen's } d = 0.05, 95\% \text{ CI } [0.03, 0.07]$) and on economic collectivism ($t(14, 743) = 5.59, p < 0.001, \text{Cohen's } d = 0.06, 95\% \text{ CI } [0.04, 0.08]$). However, they are slightly more progressive than the broader academic population on cultural liberalism ($t(149, 470) = 3.34, p = 0.0009, \text{Cohen's } d = -0.02, 95\% \text{ CI } [-0.03, -0.01]$) and economic collectivism ($t(143, 033) = 3.53, p = 0.0004, \text{Cohen's } d = -0.02, 95\% \text{ CI } [-0.03, -0.01]$).”

L313-319 (Discussion) reads

“Notably, prominent academic influencers on social media are often not leading scholars, indicating a divergence between academic recognition and social media visibility, potentially distorting perceptions. Academics with high public reach but limited subject expertise show weaker support for climate action compared to other academics, aligning more with average social media users. This suggests a division of labor in science communication but could also reflect individual incentives tied to popularity”

L325-329 (Discussion) reads

“We observe that academics with high Twitter visibility but lack of domain expertise show the least support for proactive climate actions and progressive cultural values. Such dynamics may be notably shaped by the algorithms of social media platforms and could have a disproportionate influence on public narratives and policy decisions, potentially creating a skewed representation of the state of consensus opinion in academia.”

15. Quote: *“L718: two field categories mentioned but 3 categories are used in the main text.”*

Response:

Thank you. This was a typo. This is now fixed.

16. Quote: *“Table 1: why are the N different under the Tone and Style of Expressions group?”*

Response: Thank you. This was a typo. This is now fixed.

17. Quote: *“There are numerous minor typo errors and inconsistencies in the text:*

Line 36: “there is are ...”

Line 44: “Too date” -> “To date”

Line 139: thing -> think

Line 555-556: “I”, “my”, “Future drafts will” Also footnote on page 16.

L250: “continuous continuous”

L476: “Open AI””

Response:

Thank you. These are now fixed.

Responses to reviewer 2: Communication/Communication/Information on social media, Social Media data analysis, science communication & science engagement online

Thank you for your thorough and insightful feedback. We have addressed each of your comments below in the order they were presented.

1. Quote: *“In the abstract you say “over 100,000 scholars”, but on page 3 you say 99,274 scholars.”*

Response:

Thank you for noting this. “Over” was a typo. It is now corrected to “nearly” 100,000 scholars.

2. Quote: *“On line 245, you say (e.g., lower impact factor)... >> I believe this should be “higher impact factor” --- should impact factor be defined anywhere for audience?”*

Response:

There was an error in the Egocentrism panel of Figure 4. After revising it, it is consistent with the text. We now also defined the impact factor in the relevant footnote as follows:

“The impact factor, also known as the 2-year mean citedness, is calculated by using the citations from the year prior to the current year (numerator) and the citation-receiving publications from the two years prior to that (denominator).”

3. Quote: *“You used OpenAlex for location, affiliation, rankings, but you don't mention the potential missing information that is in the OpenAlex dataset nor do you speak to validating the information you found (e.g. <https://link.springer.com/article/10.1007/s11192-023-04923-y>)”*

Response.

Thank you for your thoughtful comment regarding the use of OpenAlex (OA) data. We have carefully considered your feedback and would like to address each of your concerns.

Potential Missing Information and Validation: In our analysis, we ensured that there were no missing values for affiliation, location, or ranking for any authors included in the final dataset. Specifically, we checked the final sample and confirmed that there were no missing (NA) values at the author level for these variables. As mentioned in the Methods section 4.7 under “Location, Institutional Affiliation, and Rankings,” we did not use OA directly for rankings. Instead, we used OA to obtain affiliations, which we then matched with the QS World University Rankings. This approach allowed us to achieve full coverage for all authors in analyses requiring these variables.

Unit of analysis relevant for the degree of missingness The study that you cited focuses on potential missing information at the *journal article level* within Open Alex. Our analysis however, is at the *level of the individual academic*. For most academics this implies we have multiple data points in any given year to infer their institutional affiliation. This reduces the overarching concern of missing institutional affiliation information. Further, the reference you cited also highlights, in their Figure 5, that the extent of missing information *at the article level* has drastically declined over time. In fact, the reference you provide stops in 2012 in terms of the temporal breakdown of the missingness of affiliation information at the article level. Our focus is on data from 2016-2022, a period where OA's coverage is most comprehensive.

Relevance and Coverage of OpenAlex It is worth giving a bit more context on Open Alex. OA is built on the foundation of the Microsoft Academic Graph (MAG) and CrossRef, making it the largest open-access bibliometric repository available, second only to Google Scholar, which is not open-access. OA is widely recognized for its extensive coverage, including retracted papers, which are challenging to find in other databases. For example, OA has been shown to cover about 90% and 75% of all articles found in Scopus and Web of Science (WoS), respectively, and often includes significantly more works (Jiao et al., 2023, Culbert et al., 2024). Furthermore, OA nearly doubles (and in some cases more than triples) the number of works found by Scopus in various source categories. This broad coverage is one of the reasons why bibliometric services like Overton have switched to using OA as their primary data source (Chawla, 2022).

Strengths and Limitations of OpenAlex While OA generally outperforms other databases in terms of coverage, some limitations do exist, particularly regarding author names, references, citations, and institutional data, as highlighted in recent studies (e.g., Culbert et al., 2024; Akbaritabar et al., 2023). However, our reliance on the 2016-2022 data, which has the most comprehensive coverage within OA, mitigates many of these concerns. Additionally, by focusing on author-level affiliation data, we avoid many of the inconsistencies associated with article-level data.

In summary, we are confident that our use of OA data is both appropriate and robust for the purposes of our study. The decisions we made in data selection and processing ensure that the potential issues highlighted by the reviewer are addressed effectively. We have cited relevant studies in our updated Methods section to provide further context and validation of our approach. We hope this response clarifies our methodology and addresses your concerns.

References:

- Akbaritabar, A., Theile, T. and Zagheni, E., 2023. Global flows and rates of international migration of scholars (No. WP-2023-018). Max Planck Institute for Demographic Research, Rostock, Germany.
- Chawla, D.S., 2022. Massive open index of scholarly papers launches. Nature.
- Culbert, J., Hobert, A., Jahn, N., Haupka, N., Schmidt, M., Donner, P. and Mayr, P., 2024. Reference coverage analysis of openalex compared to web of science and Scopus. arXiv preprint arXiv:2401.16359.

- Jiao, C., Li, K. and Fang, Z., 2023. How are exclusively data journals indexed in major scholarly databases? An examination of four databases. *Scientific Data*, 10(1), p.737.

4. Quote: “You switched from GPT 4 to GPT 3.5 Turbo for stance detection and narrative extraction. I’m wondering what difference, if any, you might find had you used GPT 4 for this part of the work? You mention that you validated to F1 around 84 to 92, would this have been higher with GPT 4? Did you test a small subset to see if 3.5 and 4 gave similar results?”

Response

Thank you for your insightful question regarding our use of GPT-3.5 Turbo for stance detection and narrative extraction, particularly in comparison to GPT-4. We recognize the importance of validating our model choice, especially given potential differences in performance between GPT-3.5 Turbo and GPT-4. We followed your suggestion and carried out a range of further validation.

First, we validated our stance detection method using the ACL SemEval-2016 Task 6 dataset, as detailed in the updated stance detection validation appendix. We compared GPT-3.5 Turbo and GPT-4 on this dataset and found their performance to be very similar. The average F1 scores for both models were consistently high, with GPT-4 only slightly outperforming GPT-3.5 Turbo on the Legalisation of Abortion topic (84.36 vs. 79.52). For other topics, including Feminism, Hillary Clinton, and Donald Trump, GPT-3.5 Turbo either matched or exceeded GPT-4’s performance, with F1 scores ranging from 84.18 to 92.44.

Further, we expanded our validation by comparing GPT-3.5 Turbo with GPT-4, GPT-4o-mini, and GPT-4o across the specific topics in our study. We analysed 8,400 tweets labelled by all four models over 10 iterations, totaling 336,000 predictions. After filtering out unrelated labels, we found strong consistency across models, with agreement rates and F1 scores improving as model quality increased. GPT-4o, the most advanced model, achieved the highest agreement rates and F1 scores, confirming the robustness of our stance detection approach. For reference, we reproduce the results presented in Appendix A2 here.

Table A2: Comparison of Agreement and F1 Scores Across GPT Models

Comparison	Agreement (Modal)	Agreement (Iterations)	F1 (Modal)	F1 (Iterations)
GPT-3.5-turbo vs GPT-4o	0.781	0.772	0.806	0.795
GPT-3.5-turbo vs GPT-4	0.750	0.738	0.772	0.756
GPT-3.5-turbo vs GPT-4o-mini	0.684	0.681	0.696	0.691

Notes: Table presents a comparison of agreement rates and average F1 scores (F_{avg}) between different GPT models for stance detection. **Agreement (Modal)** refers to the proportion of identical stance predictions when considering the modal stance across 10 iterations per tweet. **Agreement (Iterations)** measures this proportion for each individual iteration treated as a unique instance. **F1 (Modal)** represents the average F1 score when one model’s modal stance is used as the “true” label to evaluate another model’s performance. Similarly, **F1 (Iterations)** does the same but treats each iteration as a separate instance. Higher values in these metrics indicate stronger consistency and accuracy in stance classification.

Given the minimal performance differences and the significantly lower cost of GPT-3.5 Turbo (around 20 times cheaper), we are confident that it provides an optimal balance of accuracy

and cost-efficiency for our tasks. The updated appendix includes these validation results, further supporting the reliability of our method.

Responses to reviewer 3: Ethical issue in research involving social media data sources

Remarks to the author:

"It was agreed with the editor that I would only review the reflection of research ethics in the paper, particularly related to the use of online data. That is a very modest task in this case, as there is absolutely no attention for the ethics of scraping Twitter and academics' profiles without informing them, let alone asking for consent. If the authors hold the view that this is legitimate for reasons of public interest, they could at least state and explain this (see e.g., Stommel & De Rijk, 2021).

Stommel, W., & Rijk, L. D. (2021). Ethical approval: None sought. How discourse analysts report ethical issues around publicly available online data. Research Ethics, 17(3), 275-297.

Response:

Thank you for your insightful comments on the ethical considerations of using publicly available online data in our research. We fully acknowledge the importance of adhering to rigorous ethical standards, especially when dealing with online data, as highlighted in Stommel & De Rijk (2021). We provide below a discussion of the various considerations.

1. Data Collection and Platform Compliance:

Our study did not involve scraping data from Twitter or OpenAlex; instead, we used their respective APIs, adhering to the terms of services of both platforms, which are designed to provide access to public data for academic research under controlled conditions. We followed data collection practices that respected user privacy throughout the process.

2. Public vs. Private Data Distinction:

We recognize that the public availability of data on platforms like Twitter does not automatically negate the ethical considerations of using that data in research. However, our study focused exclusively on data that users have willingly made publicly accessible. In line with Stommel & De Rijk (2021), we considered the public nature of Twitter posts—analogueous to conversations in a public square—where users are generally aware that their content is visible to the broader public. This context informed our decision that explicit informed consent was not required, as participants have already consented to the public sharing of their content under Twitter's terms of service.¹

3. Anonymization and Retrievability:

¹ "By publicly posting content, you are directing us to disclose that information as broadly as possible, including through our APIs, and directing those accessing the information through our APIs to do the same." Twitter. https://x.com/en/privacy/previous/version_17

To further protect the privacy of individuals, all data in our study was anonymized and aggregated, ensuring that no personal identities could be traced back to specific data points. We understand the concern regarding the retrievability of quotes, as noted by Stommel & De Rijk (2021).

Although our study did not involve direct quotations or verbatim excerpts from Twitter posts, we took several precautions to ensure that individual users could not be identified, even through indirect means. We removed all personally identifiable information, including names, usernames, Twitter IDs, tweet IDs, and raw tweet content, thereby preventing any direct linkage to individuals. Additionally, the data was aggregated to a level where individual tweets could not be traced back to specific users, summarising content across time periods rather than quoting directly. Where references to users were necessary, we employed anonymized codes without any linkage to original identities, further safeguarding user privacy.

Furthermore, all data was securely stored in encrypted formats, both in transit and at rest, with access strictly controlled and limited to essential research personnel. These measures ensured that the data was used responsibly, in line with ethical guidelines and the need to protect user privacy.

4. Reflection on User Expectations:

We also considered the expectations and intentions of Twitter users. Although these expectations can vary, we operated under the assumption, supported by prior research, that users of public platforms like Twitter understand their posts are accessible to anyone, including researchers. Nevertheless, we took care to anonymize and aggregate the data to minimise any potential harm or breach of user expectations.

5. Ethical Approval and Research Integrity:

Our study received IRB approval from Imperial College London, which confirmed that no significant ethical issues were identified in the research protocol. The ethical review process ensured that our methods complied with the Data Protection Act 2018 and GDPR, further safeguarding participant privacy and confidentiality.

Moreover, we ensured that all data storage and processing were conducted securely, using encrypted storage solutions like Google BigQuery. This compliance with institutional and regulatory standards further reinforces our commitment to ethical research practices.

6. Importance of Study and Justification for Data Use:

In line with the suggestion by Stommel & De Rijk (2021) to clarify the importance and benefits of the research, we believe our study contributes significantly to understanding how academics engage with politically salient topics on social media. This understanding is crucial for public discourse and policymaking, especially in an era where social media increasingly influences public opinion. We believe that the societal benefits of this research justify the use of publicly available online data, while our rigorous ethical considerations mitigate any potential risks to individuals.

7. Transparency and Future Research:

We are committed to transparency in our research process. A replication package, including the aggregated data and analysis code, will be made available to the academic community. This approach aligns with the call for more detailed reporting of ethical considerations, as emphasised by Stommel & De Rijk (2021).

In addition, we have included an "Ethics and Inclusion Statement" in the revised manuscript, which outlines our adherence to ethical guidelines, anonymization protocols, and our efforts to ensure inclusion and transparency in the study.

We believe that our approach to using publicly available online data is ethically sound, adhering to both the letter and the spirit of ethical research guidelines. We appreciate the opportunity to further clarify these points and have strengthened our manuscript to reflect these considerations, ensuring that our research is both ethically responsible and valuable to the broader academic community.

Reference:

- Stommel, W. and Rijk, L.D., 2021. Ethical approval: None sought. How discourse analysts report ethical issues around publicly available online data. *Research Ethics*, 17(3), pp.275-297.

Responses to reviewer 4: Political attitude, Communication/Information on social media, Social Media data analysis

Thank you for your thorough and insightful feedback. We have addressed each of your comments below in the order they were presented.

1. Quote: *“I think a good research report raises more questions than provides answers, and this manuscript clearly does that. The authors seek to characterize expression via social media of academics at large, and to highlight ways in which those who work as academics may differ in their online communication--in form and content--from those who are not. The effort here is at a very broad scale, which directly relates to both its chief advantages and disadvantages. The breadth of the analysis, examining a large number of expressions affiliated with academics tweeting, provides us with some good broad comparisons in terms of who is tweeting, how they are tweeting, their reach, and how this may relate to expertise.”*

Response: Thanks a lot for this point. You do highlight an important challenge with any form of applied research that leverages high dimensional data: the need for dimensionality reduction to make the underlying data accessible. Regression techniques and comparisons of means is one of the most common methods for such dimensionality reduction. We took specific care and attention to work with data that pertains to a balanced panel of academics on social media: that is, academics that consistently are active on the platform. This ensures that the temporal variation in the salient political expression can be studied over time and that differences are not confounding compositional effects of the underlying estimating sample.

We have highlighted this approach in more detail in the paper (Appendix section E.4 “*Robust Trends - Partialling Out Individual Fixed Effects*”) and added some further appendix tables that highlight the compositional differences this implies along with a detailed justification due to the desirable feature of allowing longitudinal comparisons.

2. Quote: *“While these distinctions are helpful, I think we lose a great deal of detail in comparing means of the two populations. It is certainly indicative of where more research could head, but that broad view confirms what, I suspect, many would assume is the case: scientists (and other folks who have longer-than-average educational careers) are more likely to hold scientific expertise in high esteem, and therefore more likely to hold the scientific consensus on climate change to be the best one to adopt, for example.*

I think there is more value in confirming that many “public intellectuals” in the social media age are not especially influential in their own fields. When asked to name an astronomer or astrobiologist, many in the public would default to Carl Sagan, for example. For a physicist, almost certainly Albert Einstein, though they may also be aware of Richard Feynman thanks to his public efforts, and there can be little doubt these folks were also influential to their field. Particularly in the case of junior scholars, the public communication online may also be a proxy for their informal connections in the field, and may lead to more academic opportunities thanks to “name recognition.”

But there are significant challenges to drawing broadly from this analysis. The authors do a good job, I think, of highlighting some of these. They are very clear, for example, that the expressiveness on a topic like climate change may have more to do with comfort or desire to publicly communicate on the topic rather than convictions. And they note that those academics with the greatest public reach may not be representative of the top experts in the field.”

Response:

Thank you for the valuable feedback regarding the role of “public intellectuals” in the social media age. We have taken your suggestion on board and made several additions to clarify and emphasise the distinction between social media visibility and academic influence.

First, we added a footnote in the results section 2.1 *“Distribution of Academic Influence on Twitter”* to highlight the weak correlations between academic recognition (e.g., citation counts) and social media influence (followers, likes). Specifically, the correlations between citations and likes (-0.024), citations and followers (0.090), and citations and content creation (-0.021) demonstrate that many prominent public intellectuals on social media are not necessarily leading scholars in their fields. The footnote reads:

“The weak correlation underscores that many prominent public intellectuals online gain visibility through public engagement rather than scholarly achievements, often holding lower academic credentials while commanding significant public attention, thus widening the gap between social media influencers and established academic experts.” (L100)

This directly supports the point raised in your review, confirming that social media prominence does not equate to traditional academic authority.

Additionally, we have expanded the discussion section to reinforce this point further. The revised text now states:

"The over-representation of certain views—especially from high-reach, low-expertise academics—combined with under-representation of experts could result in miscommunication of scientific consensus. This suggests that many 'public intellectuals' gain visibility more through public engagement than traditional academic influence, reinforcing that online prominence does not necessarily reflect expertise. Thus, those with the greatest public reach may not represent top scholars, potentially distorting public perceptions." (L386-L391)

These changes ensure that we have explicitly addressed your concerns by highlighting the divergence between online influence and academic standing, making it clear that visibility on social media often stems from public engagement rather than scholarly impact. This provides further evidence of the need for caution when interpreting the influence of "public intellectuals" in shaping public discourse.

3. Quote: *"The article slowly eases us into the sample, which is drawn from Twitter. While the case can be made that Twitter is of primary importance as a social media platform, and the authors do so to a certain degree by noting its use by journalists in particular, it is clearly not *all* social media use. Moreover, the forms of expression on Twitter are shaped heavily by the ways in which the algorithm (here to indicate both social and technological factors) surface and reinforce certain types of expression and evidence. This can be as simple as (early) length constraints, but a number of folks (e.g., Pfizner, Garas, & Schweitzer) have shown that emotional divergence predicts sharing of Tweets. It seems likely that academics may be successful on Twitter specifically to the degree to which they may code-switch and adapt to the discourse community specific to Twitter. I think it would be wrong to assume that the same kinds of dynamics are necessarily present on other platforms.*

And, of course, it would likely be wrong to assume that the constraints of Twitter/X have remained the same over time. We have seen a number of evolutionary forces shift the constraints, along with very explicit attempts to change discourse on the site. Some of these changes have resulted in academics abandoning the platform altogether. There has been some broad criticism of over-emphasizing Twitter in research because of the availability of the corpus. I don't think we have to go that far—there is a case to be made for its influence—but I do think it's important to bear in mind that platforms matter, and the expressions that are found on a social media platform are necessarily shaped by a negotiation between users and the structures of the platform."

Response:

Thank you for raising this important point regarding the specific role of Twitter within the broader social media ecosystem and the potential effects of algorithmic changes on the dynamics of academic expression. We fully agree that Twitter represents only one platform in a diverse media landscape and that the structure of expression on Twitter is shaped by its algorithms and platform constraints. We would like to clarify several key points:

Twitter's Role in the Media Ecosystem. While we recognize that Twitter is not representative of the entire social media ecosystem, we emphasize its particular relevance as a platform for original content and as a "multiplier" due to its popularity among journalists, politicians, and policymakers. Twitter's influence extends beyond academia, with its use as a dissemination tool, particularly for high-impact topics like climate change and socio-economic policy. This focus on Twitter is thus grounded in its outsized role in shaping public discourse, which is well documented.

Algorithmic Amplification and Expression Dynamics. We also agree that algorithmic amplification or selection plays a relevant role in shaping expression dynamics. However, we want to clarify that our analysis does not focus on the extensive margin of expression—such as how often a post is retweeted or engaged with. Instead, we measure genuine expression, where each academic in our econometric exercise is given equal weight, regardless of engagement levels. Although we do not track engagement metrics, we acknowledge that algorithmic amplification could still increase the visibility of certain topics, which may, in turn, encourage more political expression among academics. To address this, we have added a clarification in the discussion section about how algorithms may shape broader discourse. Here is the relevant addition in the discussion section:

“While our focus is on genuine expressions rather than engagement metrics like retweets or likes, algorithmic amplification could still influence the visibility and prevalence of certain topics, potentially encouraging more political expression among academics and shaping broader discourse.” (L365-368)

You correctly note that algorithmic factors on Twitter—such as emotional divergence—can influence the visibility and sharing of content. This is consistent with research suggesting that emotionally charged content is more likely to be shared. In response, we have addressed this concern in our paper by conducting an analysis of whether “how” expressions (tone, emotionality, egocentrism, and toxicity) might confound the observed patterns in the “what” of academic expression. Specifically, we performed regressions controlling for expression style (emotionality, egocentrism, and toxicity) and found that these factors do not significantly alter subgroup differences in stance.

This suggests that while Twitter’s algorithm may amplify emotional content, our findings of political stance differences across subgroups are robust and not driven by tone or emotional expression. This analysis is detailed in our response to a different referee and included in Table A7, and Appendix Figures A15 and A16, which show that controlling for these “how” variables does not alter the substantive patterns in our data. We reproduce the analysis here for convenience:

Raw Correlations Between Stances and “How” Expressions

First, we examined the raw correlations between each stance (the “what”) and three key measures of expression style (the “how”)—emotionality, egocentrism, and toxicity. The results are presented in the table below:

Table A7: Raw Correlations Between Stances and Emotionality, Egocentrism, and Toxicity

Stance	Emotionality	Egocentrism	Toxicity
Climate Action	-0.0081 p = 0.0017 [-0.0131, -0.0030]	-0.0272 p < 0.0001 [-0.0323, -0.0222]	-0.0326 p < 0.0001 [-0.0377, -0.0276]
Techno-Optimism	-0.0166 p < 0.0001 [-0.0217, -0.0116]	-0.0407 p < 0.0001 [-0.0458, -0.0357]	-0.0553 p < 0.0001 [-0.0604, -0.0503]
Behavioural Adjustment	-0.0075 p = 0.0035 [-0.0125, -0.0025]	0.0003 p = 0.9128 [-0.0047, 0.0053]	-0.0135 p < 0.0001 [-0.0186, -0.0085]
Cultural Liberalism	0.0209 p < 0.0001 [0.0159, 0.0259]	0.0568 p < 0.0001 [0.0518, 0.0618]	0.0854 p < 0.0001 [0.0804, 0.0904]
Economic Collectivism	0.0035 p = 0.1727 [-0.0015, 0.0085]	0.0149 p < 0.0001 [0.0099, 0.0200]	0.0002 p = 0.9267 [-0.0048, 0.0053]

Note: This table presents the raw correlations, p-values, and 95% confidence intervals (CIs) between different political stances and three key variables related to how academics express themselves: **Emotionality**, **Egocentrism**, and **Toxicity**. These variables represent the tone and style of expression, with emotionality capturing the ratio of affective to cognitive words, egocentrism measuring the prevalence of self-referential language, and toxicity capturing the likelihood of harmful or aggressive language. The stances—Climate Action, Techno-Optimism, Behavioural Adjustment, Cultural Liberalism, and Economic Collectivism—are defined as the net stance per author on each topic. The raw correlations are all near zero, suggesting minimal direct relationship between stance and expression style, with the highest correlation being 0.085 (*p* < 0.0001, 95% CI [0.0804, 0.0904]) for Cultural Liberalism and Toxicity. This indicates that "how" things are said does not heavily influence the political stance expressed by academics.

These correlations show that the relationship between stance and expression style is minimal in most cases. For example, the correlation between emotionality and stance is very close to zero, with the largest observed correlation being 0.085 between cultural liberalism and toxicity. This suggests that, in general, expression style does not have a strong raw correlation with stance.

Regression Analysis to Control for “How” Expressions While Examining Subgroup Differences: In the second step, we explored whether the observed differences across subgroups could be confounded by expression style. To do this, we ran a series of regressions for each stance, comparing subgroup differences (e.g., gender, field, rankings, and country) while controlling for one of the three “how” expressions at a time (emotionality, egocentrism, or toxicity). Specifically, we estimate the following model:

$$\text{Stance}_{it} = \alpha + \beta_1 * \text{GroupVar}_{it} + \beta_2 * \text{HowExpression}_{it} + \varepsilon_{it}$$

Where GroupVar_{it} represents a subgroup indicator (e.g., male, STEM, US), and $\text{HowExpression}_{it}$ represents one of the three expression styles (emotionality, egocentrism, or toxicity). This allows us to assess whether controlling for the “how” variables alters the subgroup coefficient β_1 . If β_1 remains stable across the three models, it indicates that tone and style of expression do not confound subgroup differences.

Climate-related stances: We begin by presenting the subgroup differences for climate-related stances (Climate Action, Techno-Optimism, and Behavioural Adjustment), as originally shown in the paper (Figure 2). We then display the updated results after controlling for emotionality, egocentrism, and toxicity.

Figure: Subgroup Comparisons for Climate-Related Stances (Figure 2 Reproduced)

Figure: Subgroup Comparisons for Climate-Related Stances, Controlling for Emotionality, Egocentrism, and Toxicity (Appendix Figure A15 reproduced)

As seen in the new figure, the coefficients for subgroup differences remain consistent across the three regressions, each controlling for one of the expression styles. This suggests that the "how" expressions—whether emotionality, egocentrism, or toxicity—do not confound the subgroup differences for these climate-related stances.

Socio-economic stances: Next, we replicate this analysis for the stances on Cultural Liberalism and Economic Collectivism. Again, we start by presenting the subgroup differences from the original analysis (Figure 3) and then show the updated results with controls for emotionality, egocentrism, and toxicity.

Figure: Subgroup Comparisons for Economic and Cultural Stances (Figure 3 Reproduced)

Figure: Subgroup Comparisons for Economic and Cultural Stances, Controlling for Emotionality, Egocentrism, and Toxicity (Appendix Figure A16)

reproduced)

Again, we observe that the coefficients for subgroup indicators remain stable across all three regressions, indicating that controlling for the “how” expressions does not significantly alter the subgroup effects. Thus, it is unlikely that emotionality, egocentrism, or toxicity confound the observed differences across subgroups for these stances.

Based on our results, we find that expression style (the “how”)—whether emotionality, egocentrism, or toxicity—does not confound the observed subgroup differences in stance (the “what”). The coefficients for subgroup differences remain consistent regardless of whether we control for these expression styles, indicating that the variations we observe are primarily driven by the content of the statements rather than how they are expressed.

Potential Changes in Twitter's Algorithm Over Time. While concerns about evolving platform dynamics, especially under Twitter's recent ownership changes, are valid, we believe this has limited impact on our study. Our analysis covers the period from 2016 to 2022, before the major platform changes initiated by Elon Musk in 2023 and 2024. Although there has been a noticeable shift of users, including academics, to alternative platforms like Mastodon, Bluesky, and Threads, these migrations are primarily on the extensive margin. Due to network effects, which reinforce the value of remaining on a platform with a large and established user base, the intensive margin of migration (i.e., active engagement on alternative platforms) is likely lower. This means that while some users may have created accounts elsewhere, Twitter continues to serve as the primary platform for most of their activity.

Additionally, our use of a balanced sample—focusing on academics consistently active over time—further mitigates concerns about sporadic engagement or algorithm-driven distortions. We also perform robustness checks, controlling for individual-level fixed effects, ensuring that unobserved factors do not confound the results.

Balanced vs. Unbalanced Sample. Lastly, you also suggested that we introduce some more information about the populations and samples that are studied earlier in the paper. In response to your suggestion, we have made some changes in the text and also added some further qualification that helps improve the understanding of the estimating sample that we are working with. One particular salient dimension is the distinction between a balanced and an unbalanced sample. Our focus is on longitudinal comparisons within individual users over time. This ensures that we can interpret genuine differences in trends, rather than differences that may be driven by sporadic entry and exit of individual users in the estimating sample.

To ensure the reader can understand better the sample differences we have added a new comparison table in the appendix between the balanced and unbalanced sample in the Appendix as Table A9. This is reproduced here for reference:

Table: Balancedness Test: Comparison of Balanced and Unbalanced Populations (Table A9 reproduced)

Panel (a): Balanced Population						
Variable	Mean	Median	SD	Min	Max	N
Publication Metrics						
Nbr. Citations	1370.02	213	4523.99	1	238736	99274
Impact Factor (2Y)	16.93	9	38.56	0.02	4205	99274
Nbr. Works	57.58	24	114.38	1	7711	99274
Demographics						
Humanities	0.005	0	0.07	0	1	99274
STEM	0.291	0	0.45	0	1	99274
Social Sciences	0.098	0	0.30	0	1	99274
Gender: Male	0.60	1	0.49	0	1	99274
Twitter Metrics						
Nbr. Likes	3210.85	1673	5736.33	0	293797	99274
Nbr. Followers	1112.70	533	9145.94	0	1087504	99274
Nbr. Accounts Followed	749	529	1116.55	0	191923	99274
Nbr. Retweets	843.72	276	2117.91	0	181268	99274
Nbr. Posts	1575.24	1032	2052.13	4	117088	99274
Panel (b): Unbalanced Population						
Variable	Mean	Median	SD	Min	Max	N
Publication Metrics						
Nbr. Citations	1083.30	139	3969.42	1	331147	219273
Impact Factor (2Y)	15.49	9	30.96	0.02	2539	219273
Nbr. Works	48.26	18	110.51	1	8465	219273
Demographics						
Humanities	0.002	0	0.04	0	1	219273
STEM	0.262	0	0.44	0	1	219273
Social Sciences	0.071	0	0.26	0	1	219273
Gender: Male	0.59	1	0.49	0	1	219273
Twitter Metrics						
Nbr. Likes	5909.80	847	18705.81	0	2017914	219150
Nbr. Followers	2464.89	264	272177.95	0	126938095	219150
Nbr. Accounts Followed	713.74	328	2176.71	0	382058.5	219150
Nbr. Retweets	732.73	141	5519.14	0	1513703	219150
Nbr. Posts	3639.87	293	12652.17	1	734460	219150

Our balanced sample consists of approximately 99,274 academics who tweeted at least once in the first six months of 2016 and at least once in the last six months of 2022. This ensures a consistent presence across the entire study period. We compared this balanced sample with the unbalanced sample, which includes all individuals who tweeted at least once at any point during the period from 2016 to 2022.

The comparison reveals that the balanced and unbalanced samples are very similar in terms of non-Twitter metrics such as publication metrics (e.g., number of citations, number of works published), gender distribution, and discipline distribution. This similarity suggests that our balanced sample is representative of the broader academic population in these dimensions, at least on average.

However, and not surprisingly, the unbalanced sample reveals notable differences in the underlying Twitter metrics. This is entirely expected: we observe a disproportionately higher standard deviation in metrics such as the number of tweets, likes, and followers. This indicates that the unbalanced sample is noisier, with more extreme values that could introduce unnecessary variability into the analysis. By focusing on the balanced sample, we reduce this noise, resulting in a more reliable and stable dataset that is necessary to be able to ensure that the trends we are documenting are not masking compositional sample changes.

We have added this comparison to the Methods section of the manuscript, highlighting that while the balanced and unbalanced samples are comparable in non-Twitter metrics, the balanced sample is intentionally chosen to minimise noise and enhance the robustness of our results. We believe that this approach provides a more accurate reflection of academic behaviour on Twitter over time, without the distortions that could arise from extreme cases in the unbalanced sample.

4. Quote: *"Finally, the analysis makes the necessary assumption (I believe) that there is a one-to-one correspondence between a Twitter user and an academic. While that is true for some--I tweet under a single account--I would guess that a very large number of academics distribute their tweets through academic and non-academic accounts. There is a general messiness here between academics, their public communications, their academic tweets, and tweeting in general. Are these distinctions important? Given the broad thrust of the argument here, perhaps not. We might assume that the differences from the mean--both to the general population and in terms of the difference between twitter impact and traditional impact--are large enough that it is not vital. But what we gain in general brush strokes we lose in precision."*

Response:

Thank you for your insightful comment regarding the assumption of a one-to-one correspondence between a Twitter user and an academic. We acknowledge that this assumption may not always hold, as some academics may manage multiple accounts for different purposes (e.g., professional vs. personal accounts) or use pseudonyms on social media, which can complicate the accuracy of our matching process.

To address this concern, we have considered relevant work, such as the paper by Mongeon et al. (2022), which discusses the complexities of matching scholars to their Twitter accounts using open data sources like OpenAlex and Crossref Event Data. As highlighted in that paper, a single academic can have multiple identifiers (e.g., OpenAlex author IDs) and may not always tweet under a single identifiable account, leading to potential discrepancies in the one-to-one assumption. While the matching algorithm employed in our study (and others) prioritises precision over recall, this can lead to missed matches or inaccuracies, particularly for academics with multiple accounts or who don't follow consistent naming conventions across platforms.

While this introduces some limitations to the precision of the dataset, the large sample size and the consistency of the observed trends across different subgroups suggest that the broader patterns identified are likely to remain robust despite these potential mismatches. Our discussion now explicitly acknowledges this limitation and its potential impact on our results. Here is the relevant excerpt:

"Additionally, our assumption of a one-to-one correspondence between Twitter users and academics may not always hold, as some manage multiple accounts or use pseudonyms. However, the large sample size and consistent trends across subgroups suggest that the key patterns remain reliable despite these potential mismatches." (L368-372)

5. Quote: *“Those limitations notwithstanding, I applaud the methodological precision and clear description of that method provided by the authors--as well as the clarity of exposition throughout. It is unfortunately rare that research reports in this area are thorough enough to be reproduceable and this bucks that standard. It also provides an excellent framework for further exploration of these relationships. It's difficult not to see some of the current methods of sentiment analysis as blunt instruments, but this work raises questions that should be highly generative moving forward.”*

Response: Thank you for your positive feedback on the clarity and reproducibility of our methodology. One of our primary goals was to leverage generative AI models, such as GPT-3.5 and GPT-4, to improve the accuracy and scalability of key tasks like topic detection, stance detection, and gender classification. To further strengthen the reliability of our results, we incorporated additional validation steps across all aspects of the analysis, using multiple models and datasets.

For stance detection, we performed cross-validation with several models (GPT-3.5, GPT-4, GPT-4o), achieving strong agreement rates and high F1 scores, which confirmed the robustness of our approach. In topic detection, using established datasets such as SemEval-2016, we consistently demonstrated high F1 scores, showing the effectiveness of our method. For gender classification, we tested the accuracy of GPT-3.5 on a comprehensive dataset of over 147,000 unique names, and achieved very high precision and recall.

We hope that this work not only solidifies the findings of our study but also provides a useful framework for the broader research community to explore these relationships further using advanced AI techniques.

6. Quote: *“It's rare that I have so few recommendations for changes. I think at least touching a bit more on the limitations I've noted above would be helpful. At the very least, orienting us more toward the corpus quite early in the discussion would be helpful. This isn't an analysis of social media use--it's an analysis of Twitter use. There is a strong case to be made for why Twitter is a good thing to use here, but this should be clearly spelled out in the introduction, the abstract, and maybe even the title. (As someone with the name of defunct platforms in my own titles, I recognize the danger in this, but clarity should prevail.)”*

And along the same lines: I recognize the format in which the research report is presented is standard for the journal, though leaving off methods to the end is a bit discombobulating for this reader. At least some of the grit of the method needs to be reported early on--what are the populations/samples, what are the measures, etc., in broad terms earlier on so as to reduce the ambiguity. It's a much better read if bounce around the manuscript and come back up for the results, but I think being a bit more precise about what is being measured would help to understand the conceptual contribution early on.”

Response:

Thank you for your thoughtful feedback. We agree with your suggestion to clarify the role of Twitter in the study and provide more methodological context earlier in the paper. In response, we've made the following adjustments:

We replaced the term 'social media' with 'Twitter' in the specific line of the abstract that describes the core of our study: *"This paper describes patterns in academics' expression online found in a unique global dataset covering nearly 100,000 scholars linking their Twitter content to academic records."* This ensures clarity from the outset that our analysis focuses specifically on Twitter as a platform. The introduction now outlines the rationale for using Twitter, emphasising its outsized influence as a platform for content dissemination, particularly among journalists, policymakers, and academics.

To address your point about introducing methodological grit early on, we have added more detail earlier in the results section on the data sources, population size, and the tools used, providing readers with essential context without compromising the fuller methods discussion later in the manuscript.

Additionally, we have streamlined the introduction and moved portions of the discussion related to the results into the dedicated discussion section. We have also addressed the limitations you and other reviewers pointed out in the discussion. These changes were made to ensure a clearer exposition, as suggested. While the changes are modest, they collectively enhance readability and align with the guidance provided by the editor, other referees, and your suggestions.

Subject: Final Submission of Manuscript: "Political Expression of Academics on Twitter" (NATHUMBEHAV-24051944C)

Dear [REDACTED]

We are pleased to submit the final materials for our manuscript titled *"Political Expression of Academics on Twitter"* (NATHUMBEHAV-24051944C).

Summary of Paper (<250 characters):

First preference: An analysis of nearly 100,000 academics on Twitter reveals strong progressive stances on climate and social issues, driven by a small, vocal subset. The study highlights potential gaps between academic voices and public discourse.

Alternative version: A large-scale study of academics' tweets shows strong support for climate action and liberal social policies, mostly driven by a small, active minority. Findings highlight disparities in who shapes academic discourse—and how it reaches the public.

Social Media Promotion:

To maximize the visibility of this work, we welcome the use of social media to promote its publication. On X/Twitter, the first author and the corresponding author, Prashant Garg, can be tagged at @Prashant_Garg_, and Thimo Fetzer at @fetzert. Institutional accounts include Imperial College London (@imperialcollege), University of Warwick (@warwickuni), and University of Bonn (@UniBonn).

We also encourage utilizing Bluesky, where Prashant Garg is at @prashantgarg.bsky.social, Thimo Fetzer at @trfetzer.com, and our respective institutional accounts: Imperial College London (@imperialcollegeldn.bsky.social) and University of Warwick (@uni-of-warwick.bsky.social).

Suggested hashtags include: #AcademicTwitter or #ClimateAction

Point-by-Point Reviewer Response:

There are no remaining reviewer comments requiring a response, as both reviewers expressed full support for publication:

- **Reviewer #1:** "All my comments have been addressed."
- **Reviewer #3:** "The reflections on ethical issues involved in the use of Twitter data are extensive and sincere... I would support publication."

Transparent Peer Review:

We agree to participate in the Transparent Peer Review initiative. We are happy for the reviewer comments, author rebuttal letters, and editorial decision letters to be published as a Supplementary item alongside the article.

Acknowledgment of Reviewer Contributions:

We fully support the acknowledgment of reviewer contributions. We appreciate the time, expertise, and constructive feedback provided during the peer review process.

We look forward to hearing from you regarding the next steps and remain available to address any further questions or requests.

Thank you for your guidance throughout this process.

Best regards,

Prashant Garg

Department of Economics and Public Policy

Imperial College London
prashant.garg@imperial.ac.uk

Thiemo Fetzer
Department of Economics
University of Warwick & University of Bonn
t.fetzer@warwick.ac.uk